# Evolution under dark conditions of particles from old and modern diesel vehicles, in a new environmental chamber characterized with fresh exhaust emissions

Boris Vansevenant[1,2,3]; Cédric Louis[1,2]; Corinne Ferronato[3]; Ludovic Fine[3]; Patrick Tassel[1]; Pascal Perret[1]; Evangelia Kostenidou[4]; Brice Temime-Roussel[4]; Barbara D'Anna[4]; Karine Sartelet[5]; Véronique Cerezo[1]; Yao Liu[1]*

[1]EASE, University Gustave Eiffel, Bron, 69500, France
[2]French Agency for Ecological Transition, ADEME, 49000, Angers
[3]IRCELYON, University Claude Bernard Lyon 1, Villeurbanne, 69100, France
[4]LCE, Aix-Marseille University, UMR 7376 CNRS, Marseille, 13331, France
[5]CEREA, Joint Laboratory Ecole des Ponts ParisTech/EdF R&D, University Paris-Est, Marne-la Vallée, France

*Correspondence to*: Yao Liu (yao.liu@univ-eiffel.fr)

**Abstract.** Atmospheric particles have several impacts on health and environment, especially in urban areas. Part of those particles is not fresh, and has undergone atmospheric chemical and physical processes. Due to a lack of representativeness in experimental conditions, and experimental artifacts such as particle wall losses in chambers, there are uncertainties on the effects of physical processes (condensation, nucleation and coagulation), and their role in particle evolution from modern vehicles. This study develops a new method to correct wall losses, accounting for size-dependence and experiment-to-experiment variations. It is applied to the evolution of fresh diesel exhaust particles to characterize the physical processes which they undergo. The correction method is based on the black carbon decay and a size-dependent coefficient to correct particle distributions. Six diesel passenger cars, Euro 3 to Euro 6 were driven on a chassis dynamometer with Artemis Urban cold start and Artemis Motorway cycles. Exhaust was injected in an 8 $m^3$ chamber with Teflon walls. The physical evolution of particles was characterized during 6 to 10 hours. Increase of particle mass in observed even without photochemical reactions, due to the presence of intermediate volatility organic compounds and semi-volatile organic compounds. These compounds were quantified at emission, and induce a particle mass increase up to 17 %.$h^{-1}$, mainly for the older vehicles (Euro 3 and Euro 4). Condensation is 4 times faster when the available particle surface if multiplied by 6.5. If initial particle number concentration is below [8-9]$\times 10^4$ #.$cm^{-3}$, a nucleation mode seems to be present, but not measured by the SMPS. The growth of nucleation mode particles results in an increase of measured [PN]. Above this threshold, particle number concentration decreases due to coagulation, up to -27 %.$h^{-1}$. In those conditions, the chamber and experimental set up are well suited to characterize and quantify the process of coagulation.

## 1. Introduction

Air pollution is a major concern due to its impacts on climate, environment and health. It has been classified as carcinogenic to humans by the International Agency for Research on Cancer (IARC, 2016). Its effects are greater in urban areas, where pollutants such as fine particles accumulate and present a serious health risk due to human

exposure. Road traffic is an important source of particles in urban environments (Rivas et al., 2020). Many particle-emitting vehicles such as old diesel cars are still present on roads of several countries (EEA, 2020).

Freshly emitted particles and gases undergo physical evolutions in the atmosphere, such as nucleation, coagulation

and condensation (Seinfeld and Pandis, 2016). Chemical reactions in the gas phase, enhanced by oxidative processes and photochemistry, form organic compounds with lower volatility (Seinfeld and Pandis, 2016). These compounds can undergo the physical processes of condensation and nucleation, leading to Secondary Organic Aerosols (SOA). SOA is estimated to contribute to 50-85 % of the Organic Aerosol (OA) burden (Zhu et al., 2017), and to 30-77 % of PM$_{2.5}$ during haze pollution in China (Huang et al., 2014). Numerous studies focus on particle

evolution and SOA formation in the presence of added oxidants and strong UV lights (Chen et al., 2019; Chu et al., 2016; Lambe et al., 2011; Lu et al., 2019; Sbai et al., 2020; Wang et al., 2018b). They aim at characterizing the photooxidation processes of organic compounds during several hours. This might not be thoroughly representative of all atmospheric conditions (Chen et al., 2019). Several situations exist in which particles evolve with limited light (winter rush hours, nighttime, tunnels), and exposure may occur close to traffic (Harrison et al.,

2018), shortly after emissions. In those situations, some physical evolutions took place, but not yet oxidation/photooxidation processes. Some studies focused on physical processes only (Giechaskiel et al., 2005; Harrison et al., 2018; Jeong et al., 2015; Kozawa et al., 2012; Morawska et al., 2008; Zhang et al., 2004; Zhang and Wexler, 2004). They are mainly based on the measurement of particles near freeways or from vehicles exhaust plume. These methods enable the study of emissions in real conditions, but are impacted by the presence of external

sources and high dilution due to the wind and distant measurement. The presence of external sources (cooking, building, heater, wood burning, biogenic sources…) creates a mixture with the traffic emission. It induces difficulties to evaluate the impact of traffic on atmospheric aerosol. Moreover, few of these studies were conducted in the last decade, resulting in a lack of knowledge on modern vehicles, motorizations and aftertreatment technologies. Some studies focusing on particle physical evolution use a constant volume sampler (CVS) (Louis

et al., 2017), but they are limited by the short residence time.

Moreover, the physical processes of condensation and nucleation depend in part on concentrations of gas-phase particle precursors, such as Semi-volatile and Intermediate-Volatility Organic Compounds (SVOCs and IVOCs, respectively). IVOCs can participate in physical evolutions and SOA formation (Sartelet et al., 2018; Xu et al., 2020), but their quantification is impacted by sampling methodology. The uncertainties arise from the effects that

dilution and temperature have on the gas-particle partitioning. They also come from the chemical identification to quantify total IVOCs (Xu et al., 2020; Zhao et al., 2016, 2015). This results in a lack of knowledge on their emissions and on the effect of motorization and aftertreatment technologies (Drozd et al., 2019; Zhao et al., 2015). This also impacts the understanding of their role in particle evolutions and the accuracy of atmospheric models (Sartelet et al., 2018; Zhao et al., 2018). More investigations should consequently be conducted to characterize the

IVOC and SVOC emissions from modern vehicles equipped with different aftertreatment technologies. The particle physical evolutions during several hours and under controlled experimental conditions should also be investigated. The studies focusing only on vehicle emissions are complementary to those performed near freeways with presence of other sources and with photochemistry. This would help better understand the actual contribution of road traffic to particle pollution.

Laboratory studies on particle evolutions are often performed in atmospheric chambers. They are a good mean of investigating the physical processes of nucleation, condensation and coagulation. The study of those processes

requires information on particle concentrations, size and distributions (Leskinen et al., 2015; Nah et al., 2017; Pierce et al., 2008; Wang et al., 2018a). However, these parameters are affected by leakage and wall deposition due to Brownian diffusion, gravitational settling, turbulence and electrostatic charge of the Teflon walls (Leskinen et al., 2015; Nah et al., 2017; Pierce et al., 2008; Seinfeld and Pandis, 2016; Wang et al., 2018a, 2014). Leakage and wall deposition of particles induce a decrease in particle mass over time. This prevents the analysis of the mass evolutions from condensation or evaporation of organic material onto or from pre-existing particles. Moreover, since wall deposition depends on particle diameter, it affects the distribution evolutions and therefore the ability to quantify the role of coagulation. It also affects the observation of nucleation mode particles. Particle data in chambers therefore need to be corrected for leakage and wall losses.

Some studies (Grieshop et al., 2009; Nah et al., 2017; Pathak et al., 2007) have used size-independent wall losses correction methods. It is usually done with the help of a tracer, such as black carbon, assumed to be reliable, and enabling the consideration of the experiment-to-experiment variations. However, particle wall losses are known to be size-dependent (Charan et al., 2018; La et al., 2016; Leskinen et al., 2015; Pierce et al., 2008; Wang et al., 2018a; Weitkamp et al., 2007). The use of the same correction coefficient to the whole distribution can mislead the observation of processes such as coagulation. Other studies have therefore used size-dependent correction methods (Nah et al., 2017; Pierce et al., 2008; Wang et al., 2018a). They enable a more precise analysis of the processes affecting the size distribution. However, those methods are usually based on the decay of ammonium sulphate particles (Wang et al., 2018a). They assume that their loss rates are the same as those of the studied particles. Furthermore, the loss rate studies cannot be conducted simultaneously with the actual experiments. They are therefore usually carried out before and after the campaign, thus preventing the experiment-to-experiment variations from being accounted for (Wang et al., 2018a).

Firstly, this study presents the characterization of a new 8 m$^3$ cubic chamber with Teflon walls, aimed at studying the physical particle evolutions from a wide range of on-road vehicles. Six diesel vehicles (2 without diesel particle filter (DPF) Euro 3 and Euro 4; 3 Euro 5 with additive DPF or catalyzed DPF; 1 Euro 6 with additive DPF and selective catalytic reducer (SCR)) were tested to develop the new leakage and size-dependent wall loss correction method. Five gasoline vehicles (3 with port fuel injection Euro 3, 4 and 5; 2 Euro 5 with direct injection) were also used. The new leakage and wall loss correction method is based on the complementarity of a tracer (black carbon) and a size-dependent equation to correct wall losses. It accounts for experiment-to-experiment variations.

Secondly, this study presents the emissions and evolutions of particles from the 6 diesel vehicles, Euro 3 to Euro 6, in urban and motorway conditions. Emission factors (EFs) of particle number (PN), particle mass (PM), black carbon (BC), non-methane hydrocarbons (NMHCs) and IVOCs are estimated. They are used to discuss the impact of aftertreatment technologies on emissions and particle evolutions. Moreover, the results of the diesel particle physical evolutions in the chamber under dark conditions are presented. Data are corrected for leakage and wall losses using the new correction method. The physical processes are characterized in the dark in order to better understand their share in particle evolution mechanisms. This study is complementary to those focusing on photochemical processes, to better evaluate the total contribution of road traffic to atmospheric particle pollution.

## 2.    Methods

### 2.1.    Chassis dynamometer

The vehicles are tested on a chassis dynamometer, used to reproduce different driving conditions. The 2 non-power wheels are fixed to the floor, while the 2 power wheels are placed on a rotating 48" roller. The speed can go up to 200 km.h$^{-1}$, and is given by the driving guide in front of the vehicle. The dynamometer can apply traction or resistance to the wheels to reproduce the effects of inertia, vehicle load, slope, rolling resistance and aerodynamic resistance. A ventilation system is placed in front of the vehicle to avoid over heating of the engine. The air is

blown with a flow proportional to the vehicle speed to cool the engine similarly as it would be on road.

### 2.2.    Artemis cycles

The vehicles are tested with the Artemis Urban Cold start (UC) cycle and the Artemis Motorway (MW) cycle (André, 2004). They represent specific driving conditions, typical of urban environments and highways, and are chosen to simulate emissions found within big cities (small streets and urban highways). The speed profiles are

given in Fig. A1 for the UC cycle and the MW cycle.

### 2.3.    Vehicles

Dynamometer tests are performed with diesel and gasoline passenger cars (PCs), whose characteristics are given in Table 1. Six diesel vehicles are used in this study: 2 vehicles without DPF, Euro 3 (D1) and Euro 4 (D2); 3 Euro 5 vehicles, 2 with an additive DPFs (D3 and D4) and 1 with a catalyzed DPF (D5); and 1 Euro 6 vehicle (D6)

equipped with an additive DPF and a SCR. Five gasoline vehicles are tested: 3 with port fuel injection (PFI), G1 to G3 with standards Euro 3, 4 and 5; 2 Euro 5 with direct injection (DI), G4 and G5.

**Table 1: Characteristics of the diesel and gasoline vehicles used in this study for the chamber characterization and the particle evolution.**

| Diesel vehicles | | | | | | |
|---|---|---|---|---|---|---|
| **Vehicle** | **D1** | **D2** | **D3 (add DPF)** | **D4 (add DPF)** | **D5 (cat DPF)** | **D6 (add DPF and SCR)** |
| **Standard** | Euro 3 | Euro 4 | Euro 5 | Euro 5 | Euro 5 | Euro 6 |
| **Mass (kg)** | 1290 | 1085 | 1080 | 1515 | 1200 | 1090 |
| **Mileage (km)** | 191612 | 95982 | 105823 | 103000 | 85617 | 43919 |
| **Aftertreatment technologies** | DOC[1] | DOC | DOC + add DPF[2] | DOC + add DPF | DOC + cat DPF | DOC + SCR[3] + add DPF |

| Gasoline vehicles | | | | | |
|---|---|---|---|---|---|
| **Vehicle** | **G1 (PFI)** | **G2 (PFI)** | **G3 (PFI)** | **G4 (DI)** | **G5 (DI)** |
| **Injection** | PFI[4] | PFI | PFI | DI[5] | DI |
| **Standard** | Euro 3 | Euro 4 | Euro 5 | Euro 5 | Euro 5 |
| **Mass (kg)** | 1185 | 1170 | 1030 | 1100 | 1285 |
| **Mileage (km)** | 195305 | 95096 | 27712 | 97089 | 92550 |
| **Aftertreatment technologies** | TWC[6] | TWC | TWC | TWC | TWC |

[1] DOC = Diesel Oxidation Catalyst        [4] PFI = Port Fuel Injection
[2] DPF = Diesel Particle Filter        [5] DI = Direct Injection
[3] SCR = Selective Catalyst Reducer        [6] TWC = Three-Way Catalyst

### 2.4. Exhaust gas sampling and injection into the chamber

Exhaust gas is sampled from the tailpipe by an ejector dilutor (Dekati L7, Sapelem 50 NL.min⁻¹, Sapelem 1 NL.min⁻¹). The dilutor works with air which is dehumidified, filtered using HEPA particle filters and activated carbon, and heated at 120 ° using a Fine Particle Sampler (FPS Dekati, 4000). Exhaust gas is diluted between 3 and 20 times, depending on air pressure and ejector dilutor type. The air and exhaust gas are injected into the chamber at the height of 90 cm, through a 2 to 4 m long stainless-steel line. The line is heated at 120 °C to avoid condensation and deposition on its inner surface. The heated line is cleaned after each injection with hot dry clean air, during a couple of hours.

### 2.5. Environmental chamber

The study of the exhaust particle evolution is performed inside the University Gustave Eiffel UGE 8 m³ cubic chamber. The structure of the chamber is made of aluminum, and the walls are made with a 70 µm thin Teflon film, used because chemically inert. However, it can be electrostatically charged and influence the wall deposition of gas and particles inside the chamber. The theoretical total volume is 8000 L, but it can vary ($\pm$ 300 L) due to deformation of the walls, depending on inner pressure. To avoid contamination from the outside, the chamber is always kept at a slight overpressure (~ 5 Pa) by injection of clean air. This also compensates the losses by leakage and instrument sampling. This injection induces dilution inside the chamber. The temperature, humidity and relative pressure are continuously monitored. Instruments sample the air of the chamber at the center, at 5 different heights: 20 cm, 60 cm, 100 cm, 140 and 180 cm.

### 2.6. Cleaning

Before the first experiment, the inner walls of the chamber were cleaned by hand with microfiber cloth. While this removes most particles and gaseous species on the walls, it also induces high electrostatic charge due to friction of the cloth on Teflon. Manual cleaning was therefore only done once. At the end of each experiment, cleaning is performed with injection of dry clean air, with flowrates varying between 150 and 300 L.min⁻¹. Cleaning lasts a minimum of 12 hours, thus renewing the air inside the chamber at least 13 times. Since dilution induces an exponential decrease of the pollutant concentrations, this gives a minimum theoretical percentage of clean air inside the chamber of 99 %.

### 2.7. Instruments

#### 2.7.1. Gas-phase analysis

Non-Methane Hydrocarbons (NMHCs) are measured either directly from the tailpipe with a Horiba Portable Emissions Measurement System (PEMS) or after a CVS by flame ionization detection with a Horiba analysis system.

IVOCs are collected from the heated line (prior injection into the chamber) by sampling diluted exhaust through stainless steel tubes filled with Tenax TA at a flow rate of 45 mL.min⁻¹. The samples are collected during the entire driving cycle. Collected samples are further analyzed by Automatic Thermal Desorption coupled with Gas Chromatography-Mass Spectrometry detector (ATD-GC-MS). The Markes Unity thermodesorber and a GC6890 gas chromatograph from Agilent fitted with the MS5973 mass spectrometer from Agilent are used. The thermal desorption system consists in a 2-stage desorption. During the first desorption step, the compounds are desorbed

by heating the Tenax TA under a stream of helium and are condensed on a trap filled with adsorbent and maintained at -5 °C. During the second desorption, the second trap is flash-heated to 300 °C for a rapid introduction of the compounds into the chromatographic column. The chromatographic column used is an Agilent HP1MS

30 m × 0.25 mm, 0.25 µm. The mass spectrometer operates in scanning mode at an electron ionization of 70 eV. Mass spectral data are acquired over a mass range of 33-450 amu. Qualitative identification of compounds is based on the match of the retention time and confirmed by matching their mass spectra with those of standards and from the NIST mass spectral library. Quantification is conducted by the external standard method. Known amount (1 µL) of standard solutions of volatile organic compounds (VOCs) and IVOCs are introduced into cleaned Tenax

TA tubes, using an automatic heated GC injector. The calibration tubes are analyzed in the same conditions as mentioned before. The chromatogram shows an unresolved complex mixture, mainly composed of co-eluted hydrocarbons which cannot be further separated by a single-dimensional GC. The alkanes (linear and branched) are quantified by SIR-based response factor of these compounds using the fragment of m/z=57. The response factor used for quantification of branched alkanes is the one of the compounds with the same number of carbon as

its parent chain. The fragment m/z=83 is used for the quantification of cyclohexane and of the other cyclic compounds. The fragment m/z=78 is used for the quantification of benzene, and m/z=91 for other aromatics. The fragment m/z=128 is used for the quantification of naphthalene. The compounds from the standard solutions taken for quantification of each identified compound are listed in Table 2. The use of response factors from different compounds than the identified ones can induce some error in their quantification. Over the whole range of IVOCs,

this error is estimated to be below 8 % (7.8 % and 0.4 % respectively for 2 urban cycles). For single SVOCs, the errors range from 90 to 98 %. Since all compounds above C22 are quantified using compounds with different carbon numbers (C20 for vehicles D1, D5, and C16 for vehicles D2, D3, D6), the impact on their quantification is important. They are however of the same order for all vehicles, and allow comparison between vehicles and driving conditions.

**Table 2: List of the compounds from the standard solutions whose response factors were chosen for quantification of the identified compounds, for vehicles D1, D4 and D5 as well as vehicles D2, D3 and D6. The identified alkanes are sorted by number of carbon atoms in their parent chain: C12 to C16, C17 to C20 and C>20.**

| Identified compounds | Compounds from standard solutions whose response factor were taken for quantification | |
| --- | --- | --- |
| | Vehicles D1 and D5 | Vehicles D2, D3 and D6 |
| n-alkanes C12 to C16<br>b-alkanes with parent chain C12 to C16 | dodecane<br>tridecane<br>tetradecane<br>pentadecane<br>hexadecane | dodecane<br>tridecane<br>tetradecane<br>pentadecane<br>hexadecane |
| n-alkanes C17 to C20<br>b-alkanes with parent chain C17 to C20 | heptadecane<br>octadecane<br>nonadecane<br>eicosane | hexadecane |
| n-alkanes C>20<br>b-alkanes with parent chain C>20 | eicosane | hexadecane |
| naphthalene | naphthalene | toluene |
| cycloalkanes in the IVOC retention time range | cyclohexane | R-cyclohexane |
| aromatics in the IVOC retention time range with C≤9 | branched benzene (C9) | branched benzene (C9) |
| aromatics in the IVOC retention time range with C≥10 | branched benzene (C10) | branched benzene (C10) |

Linear (n-) and branched (b-) alkanes are classified according to their number of carbon C as an indicator of their effective saturation concentration, according to Zhao et al. (2015). Alkanes with C<12 are considered as VOCs, alkanes from C12 to C22 are considered as IVOCs, and alkanes with C>22 are considered as SVOCs. Also according to Zhao et al. (2015), IVOCs are considered to be the sum of n- and b- alkanes (from C12 to C22), naphthalene, as well as aromatics and cycloalkanes in the same retention time bin as C12-C22 alkanes.

The $CO_2$ concentration is measured from the chamber using a MIR2M (Environment SA), which samples air at 1.5 L.min$^{-1}$. The temporal resolution is 1 second, and the analytical uncertainty is 0.001 %.

### 2.7.2. Particle-phase analysis

Particle concentrations and size distributions are measured with a Scanning Mobility Particle Sizer (SMPS, TSI), composed of an advanced aerosol neutralizer (3087), a Differential Mobility Analyzer (DMA, 3081) column, and a Condensation Particle Counter (CPC, 3775). The sampling flow is 0.3 L.min$^{-1}$. Classification is based on the particle electrical mobility, and the measurement range goes from 14 to 615 nm. The particle mass is computed, with the assumption that particles are spherical, with an arbitrary density of 1.2 (Barone et al., 2011; Totton et al., 2010). Data are given with a 5 minute time scale, with a 5 % analytic uncertainty.

The black carbon concentration is measured using an AE33-7 aethalometer from Magee Scientific. Air is sampled at 2 L.min$^{-1}$ on filter tape. The black carbon concentration is given by absorption measurements at 880 nm (Andreae and Gelencser, 2006). Data are given with a time-scale of 1 minute.

### 2.8. Ammonium sulphate experiments

In order to test the wall losses correction method with non-volatile inert seeds, experiments are performed with ammonium sulphate particles. Particles are generated from diluted solutions using an atomizer aerosol generator (TSI, 3079A) at a flow rate of 5 L.min$^{-1}$ during 20 min to 2.5 hours. After injections, particles reached concentrations ranging from 5500 to 50600 #.cm$^{-3}$. Characteristics of these experiments are given in Table 3.

**Table 3: Characteristics of the ammonium sulphate experiments.**

| Concentration of solution (mg.L$^{-1}$) | Injection duration (min) | Initial [PN][1] (#.cm$^{-3}$) | Initial [PM][2] (#.cm$^{-3}$) | Initial mode (nm) |
|---|---|---|---|---|
| 25.0 | 33 | 5589 | 0.5 | 40.0 |
| 50.0 | 60 | 8100 | 0.9 | 40.0 |
| 10.0 | 24 | 19197 | 2.3 | 41.4 |
| 50.0 | 120 | 37931 | 7.2 | 59.4 |
| 10.0 | 167 | 50622 | 17.2 | 68.5 |

[1] [PN] = Particle Number concentration        [2] [PM] = Particle Mass concentration

### 2.9. Sum up of the experiments

Table 4 summarizes the experimental conditions for the 6 diesel and the 5 gasoline vehicles. Most experiments are performed under similar conditions and are repeated at least twice. For the D4 and the G3 vehicles, different conditions are tested. The line was heated once at 80 °C instead of 120 °C, showing no specific influence on particle concentration in the chamber. Also, different ejector dilutions are tested, and show a great impact on the initial particle concentrations in the chamber. An ideal dilution of 8.4 is found, leading to total dilution in the

chamber between 65 and 130. It is considered ideal for the sake of this study since it enables to have diluted exhaust giving particles concentrations covering a wide range, depending on emissions. This variety in initial particle concentration is important to study the influence of initial concentration on particle evolution.

**Table 4: Sum up of the conditions for the diesel and gasoline experiments.**

| Vehicle | Standard | Line temp (°C) | Ejector dilution | Artemis Urban Cold Start | | Artemis Motorway | |
|---------|----------|----------------|------------------|--------------------------|-------------|------------------|-------------|
| | | | | Number of cycles | Total dilution | Number of cycles | Total dilution |
| **Diesel vehicles** | | | | | | | |
| **D1** | Euro 3 | 120 | 8.4 | 2 | 97 | 3 | 65 |
| **D2** | Euro 4 | 120 | 8.4 | 2 | 97 | 3 | 130 |
| **D3 (add DPF)** | Euro 5 | 120 | 8.4 | 2 | 97 | 2 | 65-130 |
| **D4 (add DPF)** | Euro 5 | 80-120 | 2.3-15.0 | 2 | 19-387 | 1 | 26 |
| **D5 (cat DPF)** | Euro 5 | 120 | 8.4 | 2 | 97 | 2 | 65 |
| **D6 (add DPF and SCR)** | Euro 6 | 120 | 8.4 | 1 | 97 | 1 | 65 |
| **Gasoline vehicles** | | | | | | | |
| **G1 (PFI)** | Euro 3 | 120 | 8.4 | 1 | 97 | 3 | 65 |
| **G2 (PFI)** | Euro 4 | 120 | 8.4 | | | 1 | 130 |
| **G3 (PFI)** | Euro 5 | 120 | 2.3-8.4 | 3 | 19 | 4 | 13-130 |
| **G4 (DI)** | Euro 5 | 120 | 8.4 | 2 | 97 | 2 | 130 |
| **G5 (DI)** | Euro 5 | 120 | 8.4 | 2 | 97 | 1 | 130 |

### 2.10. Chamber characterization

### 2.10.1. Mixing time

During injection inside the chamber, turbulence is induced due to the flow of exhaust gas, and the mixture is not homogeneous right after the end of injection. Moreover, to avoid the increase of particle deposition surface and turbulence (Crump et al., 1982; Nomura et al., 1997), there is no fan system to make the mixture homogeneous. To determine the necessary time to have a homogeneous mixture inside the chamber, $CO_2$ is injected, and the concentration is measured at 3 of the 5 sampling heights of the chamber: the bottom one (20 cm), the middle one (100 cm) and the top one (180 cm). The average concentration over the 5 sampling heights is also monitored during injection of $CO_2$.

### 2.10.2. Leakage

Leakage describes the loss of pollutants and air due to overpressure, through the corners and door of the chamber. The leak rate is quantified experimentally using 2 complementary methods. The first method, called "pressure method", uses injection of air inside the chamber. A constant flow of air is injected at a precise value. Simultaneously, pressure is monitored, and when it gets stable, it means that the flow or air exiting the chamber (e.g. leak flow) is the same as the one injected. This experiment is realized with different flow rates from 0.1 L.min$^{-1}$ up to 6.0 L.min$^{-1}$.

The second method uses measurement of the $CO_2$ concentration, and is referred to as the "$CO_2$ method". High concentrations of $CO_2$ are injected inside the chamber, and the concentration decay is observed at constant

pressure. This method was used by Papapostolou et al. (2011) with the CO decay. The $CO_2$ concentration decreases exponentially due to dilution by compensation of air injected into the chamber, and to leakage at a given relative pressure varying between 0.17 Pa and 18.42 Pa. The decay coefficient is converted into a leak flow, expressed in L.min$^{-1}$, thus allowing comparison with the "pressure-method" results. The leak flow ($F_{leak}$) can also be expressed as a leak rate ($Rate_{leak}$), given as the percentage of the total volume exiting the chamber in 1 hour, according to 255 Eq. (1).

$$Rate_{leak}(\%vol.h^{-1}) = \frac{100 \times 60 \times F_{leak}}{8000} \tag{1}$$

### 2.10.3. Wall losses

The leakage and size-dependent wall loss correction method presented in this study is based on 4 consecutive steps. Step 1 consists in correcting total [PM] using the decay of the black carbon concentration [BC]. During step 2, the particle distribution is corrected using a size-dependent wall loss coefficient. This coefficient is based 260 on the theory of Crump and Seinfeld (1981), with an arbitrary estimation of the turbulence. Step 3 consists in optimizing the turbulence parameter $k_e$ to fit corrected data with results of step 1. Finally, step 4 is the computation of the total particle number and mass corrected concentrations.

- **Step 1: [PM] correction using [BC]**

The first step consists in correcting the total particle mass, using BC as a tracer for primary particle emissions, as 265 done by Grieshop et al. (2009). As BC is an inert compound (Platt et al., 2013; Wang et al., 2018a), the decay of its concentration is due to leakage and wall deposition, with a loss rate $k_{BC}$. By assuming that particles are internally mixed, total particle mass has the same loss rate as BC (Grieshop et al., 2009; Hennigan et al., 2011; Platt et al., 2013). This assumption can induce some uncertainty if the loss rate is size-dependent (Wang et al., 2018a). Still, it appears to be a good estimation of the [PM] losses due to leakage and wall losses, and has been used in several 270 chamber studies (Grieshop et al., 2009; Hennigan et al., 2011; Platt et al., 2013). A corrected PM concentration can thus be obtained at all times of each experiment with Eq. (2).

$$[PM]_{corrected}^{step1}(t) = [PM]_{measured}(t) \times \exp(k_{BC} \times t) \tag{2}$$

For some experiments, the decay of [BC] cannot accurately be fitted by a 1$^{st}$ order exponential decay. In those cases, the term $\exp(k_{BC} t)$ is replaced by $[BC]_{t0}/[BC]_t$. This has no consequences on the corrected [PM], since the term $\exp(k_{BC} \times t)$ in Eq. (2) is equal to $[BC]_{t0}/[BC]_t$ assuming a 1$^{st}$ order exponential decay. Both corrections are 275 therefore equivalent, the only goal being to simulate the [BC] evolution as well as possible.

This corrected concentration $[PM]_{corrected}^{step1}(t)$ represents the evolution of particle mass due to nucleation or condensation/evaporation of organic material. Particles lost to the walls are assumed to be at equilibrium with suspended ones (Grieshop et al., 2009).

- **Step 2: Particle-size-dependent correction**

The second step consists in correcting particle mass and number concentration, accounting for the size-dependence of wall deposition (Charan et al., 2018; Leskinen et al., 2015; Pierce et al., 2008; Wang et al., 2018b, 2018a). The aerosol wall deposition rate due to turbulent diffusion, Brownian diffusion and gravitational sedimentation is given

in Eq. (3) (Corner and Pendlebury, 1951; Crump and Seinfeld, 1981). It is given for a cubic chamber of side length L, as a function of particle diameter $D_p$.

$$\beta_i(D_p) = \frac{1}{L} \times \left[ \frac{8\sqrt{k_e D(D_p)}}{\pi} + v(D_p) \times \coth\left( \frac{\pi v(D_p)}{4\sqrt{k_e D(D_p)}} \right) \right] \tag{3}$$

With:

- i the SMPS particle size channel of geometric midpoint diameter $D_p$
- $k_e$ the eddy diffusivity coefficient (sec$^{-1}$)
- $D(D_p) = \frac{k_{boltz} T C_c}{3\pi\mu D_p}$ the Brownian diffusivity (m$^2$.sec$^{-1}$)
- $v(D_p) = \frac{D_p^2 \rho_p g C_c}{18\mu}$ the terminal particle velocity (m.sec$^{-1}$)
- $C_c$ the Cunningham slip-correction factor $C_c = 1 + \frac{2\lambda}{D_p} \times \left[ 1.257 + 0.4 \times \exp\left( -\frac{1.1 \times D_p}{2\lambda} \right) \right]$

All terms of this equation, except for $k_e$, can be found using literature (Crump and Seinfeld, 1981; Seinfeld and Pandis, 2016) and experimental data. They are listed in Appendix B. The $k_e$ coefficient represents the turbulence inside the chamber, induced by the exhaust gas injection, the air flow to compensate leakage and sampling, and electrostatic forces near the walls (Charan et al., 2018; Crump and Seinfeld, 1981; Pierce et al., 2008; Seinfeld and Pandis, 2016; Wang et al., 2018a). As it cannot be measured (Charan et al., 2018), $k_e$ will in a first instance be given an arbitrary value in order to compute $\beta_i$ ($D_p$). The rate of leakage is taken as defined by Schnell et al. (2006) as the ratio of the flows entering (or exiting) the chamber $\dot{V}$ by the volume of the chamber V: $\alpha = \dot{V}/V$. It follows similar basis to the "pressure method" defined above. It assumes that at constant pressure, the flow of air injected into the chamber is equal to the sum of leakage and of what is sampled by the instruments. It is determined for each experiment, using the flow of air injected into the chamber $\dot{V}$ (L.sec$^{-1}$) and the total volume of the chamber V. The loss of particles due to leakage and wall deposition in each size bin can then be estimated for the arbitrary value of $k_e$. Assuming that the coefficients $\alpha$ and $\beta_i$ ($D_p$) are constant over the course of each experiment, the loss process is a first-order exponential decay (Leskinen et al., 2015; Nah et al., 2017; Pierce et al., 2008; Verheggen and Mozurkewich, 2006; Wang et al., 2018a, 2018b, 2014) given by Eq. (4).

$$[PM_i]_{lost}^{step2}(t)\Big|_{arbitrary\ k_e} = [PM_i]_{measured}(t_0) \times \left( 1 - e^{-(\alpha+\beta_i)\times t} \right) \tag{4}$$

By adding this to the measured distribution at time t ($[PM_i]_{measured}(t)$), a distribution corrected for leakage and wall deposition can be obtained. It represents the particle mass distribution evolution, solely due to photochemical and physical processes (Eq. (5)).

$$[PM_i]_{corrected}^{step2}(t)\Big|_{arbitrary\ k_e} = [PM_i]_{measured}(t) + [PM_i]_{lost}^{step2}(t) \tag{5}$$

The corresponding total mass concentration can be computed with Eq. (6). The factor 64 comes from the fact that multiple charge corrections are based on a 64-channel resolution (TSI, 2010).

$$[PM]_{corrected}^{step2}(t)\Big|_{arbitrary\ k_e} = \frac{\sum[PM_i]_{corrected}^{step2}(t)\Big|_{arbitrary\ k_e}}{64} \tag{6}$$

• **Step 3: Optimization using least square error algorithm**

So far, the evolution of the total particle mass corrected for leakage and wall deposition has 2 different expressions. They both represent the evolution of total particle mass in the chamber solely due to photochemical and physical processes of nucleation and condensation (coagulation can also occur, but has no effect on particle mass). However, the expression obtained during step 2 ($[PM]_{corrected}^{step2}(t)\Big|_{arbitrary\ k_e}$) is attached to high uncertainties as

it is found for an arbitrary value of $k_e$. Charan et al. (2018) realized an optimal fitting of experimental data to determine the parameter $k_e$. Following the same idea, an optimization of $k_e$ is performed to minimize the difference between the corrected PM concentrations obtained during step 1 (considered as the reference) and during step 2 (Eq. (7)).

$$\sum_{t=t_0}^{t=t_{final}} \left\{[PM]_{corrected}^{step1}(t) - [PM]_{corrected}^{step2}(t)\Big|_{optimized\ k_e}\right\}^2 \xrightarrow{k_e} minimum \tag{7}$$

This operation results in an optimized expression of $\beta_i$ ($D_p$), for each chamber experiment. It is used in the equations

of step 2 (Eq. (3), Eq. (4), Eq. (5), Eq. (6)), to obtain corrected distributions and total concentrations. The values of $\beta_i$ ($D_p$) are used to compute a mean wall loss coefficient $\beta_{ke}^{mean}$, depending on the $k_e$ value found after optimization. To account for the SMPS bins, each $\beta_i$ ($D_p$) is weighted with the relative size of the channel i ($D_p$), according to Eq. (8).

$$\beta_{k_e}^{mean} = \sum_{D_{p,min}}^{D_{p,max}} \left[\beta_i(D_p) \times \frac{D_{p,i+1} - D_{p,i}}{D_{p,max+1} - D_{p,min}}\right] \tag{8}$$

With:

- $D_{p,i}$ the diameter associated with the bin i, with $D_p$ ranging from 14.1 nm to 615.3 nm
- $D_{p,i+1}$-$D_{p,i}$ the size of the bin i, where $D_{p,max+1}$=637.8 nm
- $\sum(D_{p,i+1} - D_{p,i}) = 623.7\ nm$ the total measurement range considered in this study

The optimized coefficient $\alpha+\beta_{ke}^{mean}$ represents the rate of particle losses due to leakage and size-dependent wall deposition. It is an indicator of total losses, but it is never actually applied ($\alpha+\beta_i$ is the applied coefficient).

• **Step 4: Corrected number distribution and number concentration**

The size-dependent optimized loss coefficient $\alpha+\beta_i$ ($D_p$) can be applied to the number distribution to determine what is lost due to leakage and wall deposition (Eq. (9)).

$$[PN_i]_{lost}(t) = [PN_i]_{measured}(t_0) \times \left(1 - e^{-(\alpha+\beta_i)\times t}\right) \tag{9}$$

This lost distribution can be added to the measured one, to obtain a corrected number distribution (Eq. (10)). The corresponding corrected total number concentration can be computed (TSI, 2010) with Eq. (11).

$$[PN_i]_{corrected}(t) = [PN_i]_{measured}(t) + [PN_i]_{lost}(t) \tag{10}$$

$$[PN]_{corrected}(t) = \frac{\sum [PN_i]_{corrected}(t)}{64} \tag{11}$$

This method can also be used to determine the size-dependent loss rate $\beta_i$ ($D_p$) for seed only experiments, with ammonium sulphate particles. The evolution of those particles is not impacted by effects of condensation or evaporation. This means that the corrected total mass of ammonium sulphate particles should be a constant, equal to the concentration at time $t_0$, as expressed in Eq. (12). This corrected mass in used as the reference in Eq. (7) of step 3. The mass concentration decay is associated to a rate $k_{PM}^{amm\ sul}$, which represents the losses due to leakage and wall deposition only. It can therefore be compared to the rate $k_{BC}$ found for exhaust particle experiments.

$$[PM]_{corrected}^{step1}(t) = [PM]_{measured}^{seed\ only}(t_0) \tag{12}$$

## 3.    Results

### 3.1.    Chamber characterization

#### 3.1.1.    Mixing time

Mixing time is investigated using $CO_2$ measurements at 3 sampling heights as presented in Figure 1a. They show that just 1 min after the end of the $CO_2$ injection, the vertical gradient of $CO_2$ is important. The mean $CO_2$ concentration is $0.977 \pm 0.013$ %, but this value reaches 0.964 % at the top of the chamber and 0.990 % at the bottom. After 20 minutes, the mean concentration is $0.972 \pm 0.003$ %, with a minimum value of 0.969 % at the top and 0.975 % at the bottom. From that time, the concentration variability is less than 1 % between the minimum and maximum values, and the mixture can be considered homogeneous. After 1 hour, the mixture is homogeneous if the instrument uncertainties are accounted for.

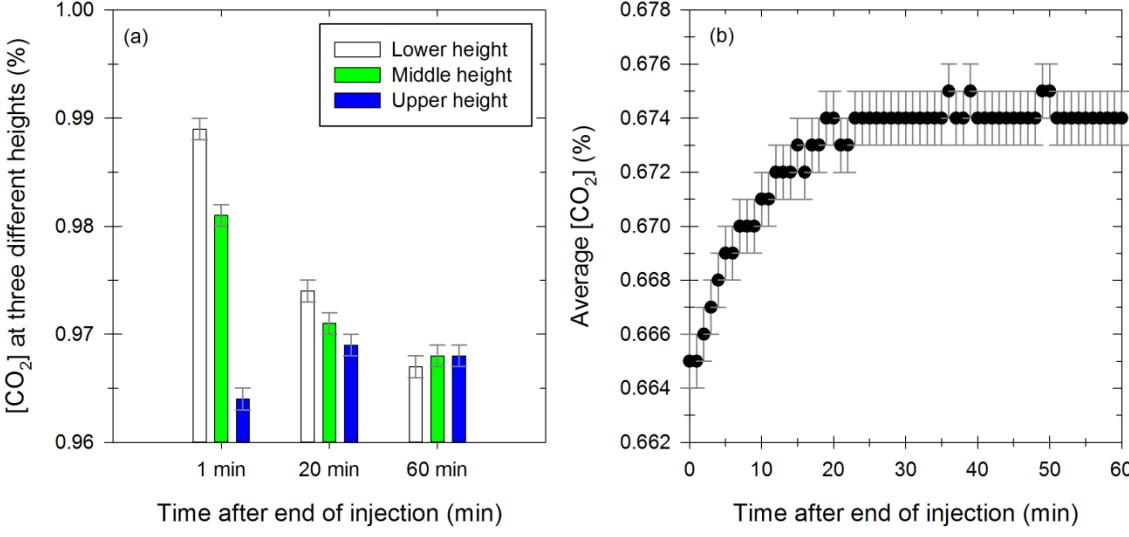

**Figure 1: Determination of the mixing time in the chamber, using injection of $CO_2$, with measurements of the concentration at 3 different heights 1 min, 20 min and 60 min after injection (a), and with evolution of the average concentration during 60 min after injection (b).**

The $CO_2$ concentration averaged over the 5 sampling heights is shown in Figure 1b. Results show that the mean concentration increases rapidly during the first 20 minutes following injection. Then it reaches its maximum and remains stable. At this stage, the mixture can be considered as homogeneous.

The results of the $CO_2$ tests indicate that the mixture inside the chamber can be considered as homogeneous 20 minutes after the end of injection. Therefore, in this study, all the chamber analysis will start 20 minutes after the

end of the exhaust gas injection.

### 3.1.2. Leakage

The leak flows obtained with the "pressure method" and the "$CO_2$ method" are presented in Figure 2 with the green and black curves respectively.

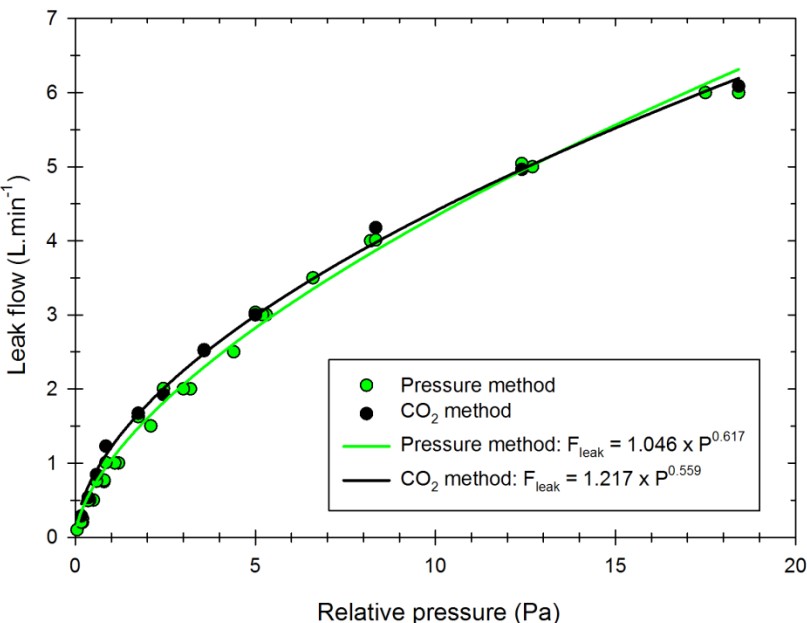

**Figure 2 : Values of the leak flow as a function of relative pressure, obtained with the "pressure method" (green) of the "$CO_2$ method" (black). The dots represent the experimental results, and the curves represent the mathematical simulation for both methods.**

Both methods give similar results. A numerical fit of the curves gives a mathematical expression (Eq. (13)) for the leak flow $F_{leak}$ (L.min$^{-1}$) as a function of the relative pressure $\Delta P$ (Pa) between the chamber and outside.

$$F_{leak}(L.min^{-1}) = (1.13 \pm 0.12) \times \Delta P^{0.59 \pm 0.04}$$
(13)

For values of relative pressure between 0.2 and 5 Pa, the leak flow is between 0.4 and 2.9 L.min$^{-1}$, corresponding to a leak rate between 0.3 and 2.2 %vol.h$^{-1}$. Platt et al. (2013) reported for the PSI chamber an average leak rate of 0.08 %vol.h$^{-1}$. This is lower than the one found in this study, probably due to fact that the chamber is operated under overpressure conditions. This leak rate can be used to estimate gas and particle phase leakage from the measured relative pressure. For this study, a leak rate is estimated for each experiment, without the dependency

on the relative pressure. Indeed, it can change between the end of injection and the remaining of the chamber experiment (the leakage due to high overpressure measured right after injection is not accounted for, but decreases rapidly). Another leak definition is given in the wall losses section, with similar basis as the "pressure method", and taking into account the instruments sampling flows.

### 3.1.3. Wall losses

The 4-step correction method is applied to more than 50 experiments using particles from passenger car exhausts (diesel or gasoline) and ammonium sulphate particles. The associated experimental protocols are described in the part Methods. The optimization processes give $k_e$ values between 0.001 and 27.32 sec$^{-1}$, with an average of 2.41 sec$^{-1}$. The corresponding wall loss coefficients are respectively $1.05 \times 10^{-4}$, $1.58 \times 10^{-2}$ and $4.69 \times 10^{-3}$ min$^{-1}$ for a particle of diameter 100 nm. These values are similar to those reported by Leskinen et al. (2015) and Babar et al. (2017), of $7.50 \times 10^{-4}$ and $3.96 \times 10^{-3}$ min$^{-1}$, respectively. Figure 3 shows the size distribution of the wall loss coefficient $\beta_i$ ($D_p$) as well as the corresponding mean wall loss coefficient $\beta_{ke}^{mean}$ for 3 values of $k_e$ in this range: $k_e$=0.36 sec$^{-1}$; $k_e$=2.57 sec$^{-1}$; $k_e$=16.70 sec$^{-1}$. These values are chosen to represent the lower part of the range, the average, and the upper part of the range, respectively. The mean wall loss coefficients are respectively $1.31 \times 10^{-3}$ min$^{-1}$; $8.44 \times 10^{-3}$ min$^{-1}$; and $2.13 \times 10^{-2}$ min$^{-1}$.

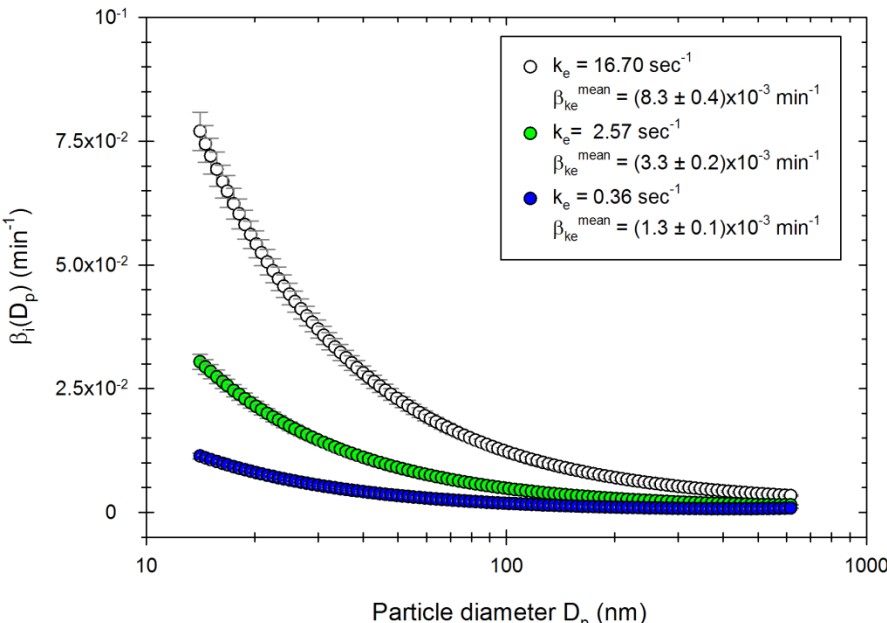

**Figure 3: Distribution of the wall loss coefficient $\beta_i$ ($D_p$) for 3 values of $k_e$ obtained experimentally: 16.70 sec$^{-1}$ (white); 2.57 sec$^{-1}$ (green) and 0.36 sec$^{-1}$ (blue).**

Figure 3 shows that for a given $k_e$, wall losses vary greatly as a function of the particle diameter in the range 15-600 nm. They are higher for smaller particles, due to higher Brownian diffusion. These results also show that the wall loss coefficients are very sensitive to the parameter $k_e$. Differences increase as the particle diameters decrease. The value of the eddy diffusivity $k_e$ depends on the turbulence induced by the injection into the chamber and the electrostaticity of the Teflon walls, which can change between 2 experiments. Since experimental conditions greatly influence the turbulence and the wall deposition (Nah et al., 2017; Wang et al., 2018a), it is important to account for experiment-to-experiment variations when correcting particle losses.

In about a third of the chamber experiments, total particle losses cannot be approximated by a simple exponential decay of [BC]. These experiments usually took place shortly after the chamber walls were manually cleaned. These actions resulted in an important electrostatic charge induced to the walls, thus increasing particle losses. For these "charged walls" experiments, the optimized values of $k_e$ range from 0.62 to 27.32 sec$^{-1}$, with an average of 7.98 sec$^{-1}$. These values are high but still lower than some values (36 and 269.4 sec$^{-1}$) found in literature (Crump and Seinfeld, 1981; Okuyama et al., 1986). The associated values of the wall loss coefficient $\beta_{ke}^{mean}$ and the total

loss coefficient $\alpha+\beta_{ke}^{mean}$ are shown in Figure 4 with the hatched boxes, for diesel (light grey) and gasoline (white). Values of $\beta_{ke}^{mean}$ are in the range $[3.82\text{-}10.67]\times10^{-3}$ $min^{-1}$ for diesel and $[1.66\text{-}4.26]\times10^{-3}$ $min^{-1}$ for gasoline. Respective mean values are $7.13\times10^{-3}$ $min^{-1}$ and $2.92\times10^{-3}$ $min^{-1}$. Values of $\alpha+\beta_{ke}^{mean}$ are in the range $[4.19\text{-}11.05]\times10^{-3}$ $min^{-1}$ for diesel and $[2.04\text{-}5.01]\times10^{-3}$ $min^{-1}$ for gasoline. Respective mean values are

$7.69\times10^{-3}$ $min^{-1}$ and $3.73\times10^{-3}$ $min^{-1}$. The high values of $k_e$ show the importance of accounting for size-dependent losses. Also, the average agreement between [PM] corrections of step 3 (see Appendix E) is $(94.7 \pm 2.6)$ %.

For the remaining experiments (69 %), total particle losses are well approximated by a simple exponential decay of [BC]. Values of $k_{BC}$ range from $1.11\times10^{-3}$ to $3.15\times10^{-3}$ $min^{-1}$ for vehicle exhaust experiments. Values of $k_{PM}^{amm\,sul}$ range from $5.27\times10^{-4}$ to $1.17\times10^{-3}$ $min^{-1}$ for ammonium sulphate experiments. For those experiments,

the average agreement between [PM] corrections obtained during step 3 (according to the calculation of Appendix E) is $(97.2 \pm 2.5)$ %. The losses found for these experiments are lower than for the "charged walls" experiments. This is because the wall electrostatic charges have been neutralized with deposition of particles during previous experiments. The associated values of $k_e$ for those "neutralized walls" experiments range from 0.04 to 3.23 $sec^{-1}$ (vehicle exhaust) and from 0.001 to 0.06 $sec^{-1}$ (ammonium sulphate). These values are in the range of what has

been found in previous studies. Charan et al. (2018) found values of $k_e$ between 0.015 and 8.06 $sec^{-1}$ in simulations of wall losses in Teflon environmental chambers. The values of the wall loss coefficient $\beta_{ke}^{mean}$ and of the total loss coefficient $\alpha+\beta_{ke}^{mean}$ associated to those "neutralized walls" experiments are shown in Figure 4 (dotted boxes), for diesel (light grey), gasoline (white) and ammonium sulphate (dark grey). Values of $\beta_{ke}^{mean}$ are in the range $[0.76\text{-}3.72]\times10^{-3}$ $min^{-1}$ for diesel, $[0.55\text{-}3.55]\times10^{-3}$ $min^{-1}$ for gasoline and $[0.25\text{-}0.63]\times10^{-3}$ $min^{-1}$ for ammonium

sulphate. Respective mean values are $1.82\times10^{-3}$ $min^{-1}$, $1.63\times10^{-3}$ $min^{-1}$ and $0.46\times10^{-3}$ $min^{-1}$. Values of $\alpha+\beta_{ke}^{mean}$ are in the range $[1.14\text{-}4.09]\times10^{-3}$ $min^{-1}$ for diesel, $[0.93\text{-}3.93]\times10^{-3}$ $min^{-1}$ for gasoline and $[0.63\text{-}1.01]\times10^{-3}$ $min^{-1}$ for ammonium sulphate. Respective mean values are $2.24\times10^{-3}$ $min^{-1}$, $2.16\times10^{-3}$ $min^{-1}$ and $0.83\times10^{-3}$ $min^{-1}$. The mean wall loss coefficients $\beta_{ke}^{mean}$ for the "neutralized walls" experiments are 3.9 (diesel) and 1.8 (gasoline) times lower than for the "charged walls" experiments. The mean $\alpha+\beta_{ke}^{mean}$ values for the "neutralized walls" experiments

are 3.4 (diesel) and 1.7 (gasoline) times lower than for the "charged walls" experiments. This is close to what was found by Wang et al. (2018a), with particle loss rates 3-4 times higher between "undisturbed" and "disturbed" experiments.

Over all the chamber experiments, the obtained values of $k_e$ range over 4 orders of magnitude. Among those experiments, the ones with ammonium sulphate particles are those with $k_e$ of the order of $10^{-3}$ to $10^{-2}$ $sec^{-1}$. These

values are low, which could be attributed to the nature of the particles. Moreover, the values of the order of magnitude $10^{1}$ $sec^{-1}$ were all found for experiments associated to highly electrostatically charged walls (e.g. with high wall losses). It could explain the high values of $k_e$. Finally, without considering those specific conditions (ammonium sulphate and highly charged wall experiments), the values of $k_e$ range over 2 orders of magnitude. In the simulations of particle wall deposition made by Charan et al. (2018), the values of $k_e$ used in the model also

range over 2 orders of magnitude (from 0.015 to 8.06 $sec^{-1}$). The values found in the standard conditions of this study are in a similar range (from 0.04 to 3.23 $sec^{-1}$). The diversity of $k_e$ values therefore seems reasonable.

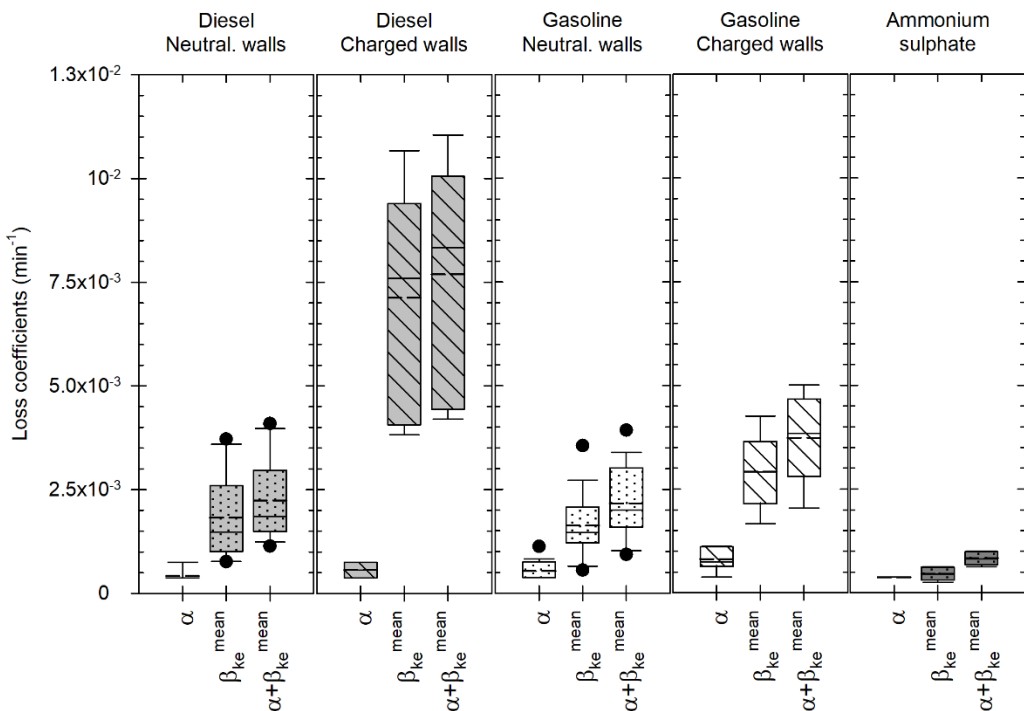

**Figure 4: Leakage coefficient α, wall loss coefficient $\beta_{ke}^{mean}$ (accounting for channel size) and total loss coefficient $\alpha+\beta_{ke}^{mean}$ for the chamber experiments performed with particles from diesel (light grey) and gasoline (white) engines, as well as ammonium sulphate particles (dark grey). Coefficients for the diesel and gasoline experiments are sorted in 2 categories: "neutralized walls" experiments for which [BC] evolution follows a 1st order exponential decay (dotted boxplots) or "charged walls" experiments for which [BC] evolutions follows a 2nd order exponential decay (hatched boxplots). The boxes represent the 25th and 75th percentile. The solid line in the center represents the median; the dashed line represents the mean. The whiskers are the 10th and 90th percentiles.**

Figure 4 shows that the leakage coefficients α are of the same order for all conditions. They only vary from $0.38\times10^{-3}$ min$^{-1}$ to $1.13\times10^{-3}$ min$^{-1}$. This is because the dilution air was injected with flows varying only between 3 and 9 L.min$^{-1}$. Moreover, leakage coefficients are much smaller than the wall loss coefficients for the diesel and gasoline experiments. They represent 19 % and 7 % of the total loss coefficients for diesel experiments with neutralized and charged walls respectively. For gasoline experiments, these values are 24 % and 22 % respectively. Wall losses therefore appear to be the main contribution of total particle losses in the chamber for exhaust experiments. For ammonium sulphate experiments, total loss coefficients are smaller than for exhaust experiments. Leakage therefore represents a bigger share, and accounts for 45 % of the total losses.

Figure 4 also shows that particles from diesel and gasoline engines cover a wide range of $\alpha+\beta_{ke}^{mean}$ coefficients, especially for some diesel experiments. This is mostly due to the fact that some of those experiments took place when the walls were electrostatically charged. The first experiments performed with charged walls (e.g. with the highest electrostatic charge) were with diesel engine particles. This explains the very high values reached by the diesel "charged walls" coefficients. Ammonium sulphate particles have lower loss coefficients than engine particles, and remain in a narrow range. This could be explained by their nature and chemical composition. Also, the flow at which they are introduced in the chamber (5.6 L.m$^{-1}$) is much lower than for vehicle exhaust experiments (20-60 L.min$^{-1}$). The turbulence and Brownian diffusion are therefore lower, and wall deposition decrease. For the ammonium sulphate experiments, the total loss coefficient (leak + wall losses) found with the 4-step correction method range from $8\times10^{-6}$ sec$^{-1}$ to $2\times10^{-5}$ sec$^{-1}$, for particles of diameter 100 nm. Nah et al. (2017) studied the wall loss coefficient ammonium sulphate of particles in a 12 m$^3$ chamber. They found values of β at 100 nm around $1-5\times10^{-5}$ sec$^{-1}$. The values found in this study are in pretty good agreement with what was found

by Nah et al. (2017). Also, the magnitude of the corrections was computed for each cycle, using the average ratio of corrected [PM] divided by measured [PM]. For the neutralized wall experiments, corrected [PM] is on average $(1.5 \pm 0.4)$ times higher than measured [PM]. For the charged wall experiments, corrected [PM] is on average $(2.8 \pm 1.5)$ times higher than measured [PM]. For the ammonium sulphate experiments, corrected [PM] is on average $(1.3 \pm 0.2)$ times higher than measured [PM]. Moreover, Platt et al. (2013) found particle half-life between 3.3 and 4 hours. This is equivalent to having [BC] decay coefficients of $3.5 \times 10^{-3}$ and $2.9 \times 10^{-3}$ min$^{-1}$ respectively. Over a 10-hour long experiment, this would give corrected [PM] 3.4 and 2.7 times higher than measured [PM] (respectively). The corrections applied in our study therefore appear to be in a reasonable range. These elements indicate that the correction method developed in this study gives good results, consistent with what is found for other chambers of comparable size. Figure 4 shows that the nature of the particles is likely to impact the loss coefficients. This shows the advantage of the 4-step correction method, which isn't based on the assumption that the particles of the study have the same losses as ammonium sulphate particles. Figure 4 also shows that the electrostatic state of the walls appears to be the most important factor determining particle losses to the walls.

The coefficients $\alpha+\beta_{ke}^{mean}$ and $k_{BC}$ both represent the rate at which particles are lost due to leakage and wall deposition, and should in theory be correlated. To investigate this, $\alpha+\beta_{ke}^{mean}$ coefficients of "neutralized walls" experiments (e.g. for which [BC] follows a simple exponential decay giving a $k_{BC}$ coefficient) are plotted in Figure 5 as a function of $k_{BC}$ or $k_{PM}^{amm\ sul}$.

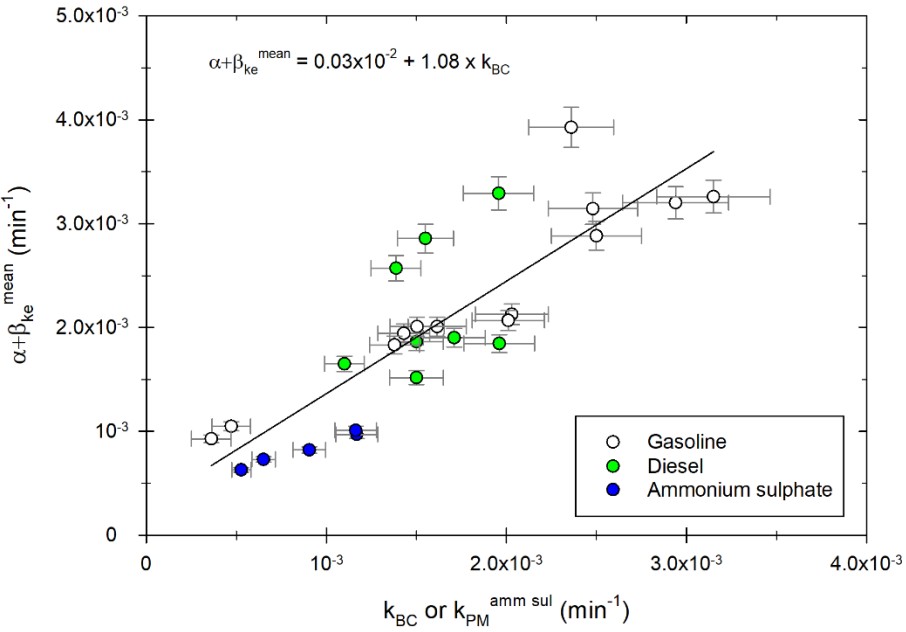

**Figure 5 : Mean loss coefficient $\alpha+\beta_{ke}^{mean}$ as a function of the decay coefficient $k_{BC}$ or $k_{PM}^{amm\ sul}$ obtained for the "neutralized walls" chamber experiments performed with particles from diesel (green) and gasoline (white) engines or ammonium sulphate particles (blue). A linear fit of the data is shown (black line).**

Figure 5 shows that there is a good correlation between both loss rates $\alpha+\beta_{ke}^{mean}$ and $k_{BC}$. The correlation is also good for ammonium sulphate experiments, between $\alpha+\beta_{ke}^{mean}$ and $k_{PM}^{amm\ sul}$. The average ratio of these 2 coefficients for ammonium sulphate experiments is 1.04. A linear regression is performed over all the exhaust and ammonium sulphate experiments. It gives a slope of $1.08 \pm 0.15$. The fact that values of $\alpha+\beta_{ke}^{mean}$ are slightly higher than values of $k_{BC}$ could be due to the size-dependence of wall losses, which results in different total loss coefficient depending on the size distribution. Overall, the correlation shows that the optimized value $\alpha+\beta_{ke}^{mean}$ gives a good representation of total losses due to leakage and wall deposition. This result indicates that the

complementarity of the [PM] correction using [BC], and the size-dependent correction, is relevant. Results of Figure 5 also show that loss rates of ammonium sulphate particles are generally lower than those from diesel or gasoline experiments, as found previously (Figure 4). Even though wall charge appears as the dominant factor for particle wall losses, the nature of the particles also seems to have an impact. This shows the interest of using the 4-step correction method described above.

### 3.2. Emission factors of particles, BC and IVOCs from diesel vehicles

Vehicle emissions are quantified in order to discuss the impact of each vehicle type and driving condition on initial chamber concentration and associated physical evolutions. Emission factors of PN, PM and BC are estimated from initial concentrations in the chamber by applying the dilution ratio. Initial concentrations are taken right after the chamber is well mixed. At that moment, no corrections for leakage or wall losses have yet been applied. This means that the concentrations taken for the computation of emission factors are those directly measured by the instruments. Moreover, as described above, some experiments occurred while the walls were highly electrostatically charged, meaning that particles can deposit onto the walls during the injection phase. This can therefore impact the initial concentrations of PN, PM and BC, and thus the computed emission factors. It is difficult to estimate the associated losses. PN emission factors found in literature for Euro 5 diesel vehicles in UC and MW conditions (Louis et al., 2016) are on the same range as those of this study. This indicates that the possible effects of wall charge are negligible. No specific correction was therefore applied to the initial concentrations. The concerned cycles are clearly identified on Figure 6.

Moreover, since condensation and nucleation partly depend on the concentration of organic material, emissions of IVOCs and certain SVOCs are quantified. They are measured directly from exhaust to estimate the quantity of available organic material in the chamber. Finally, NMHC emissions are measured from the CVS, to discuss the share of IVOCs which is identified. Six diesel vehicles (Euro 3 to Euro 6) are tested in both UC and MW conditions. Results are shown in Figure 6: EFs of PN (a), PM (b), BC (c), IVOCs (d), NMHCs (e), and the ratio (%) of the IVOC EFs over the NMHC EFs.

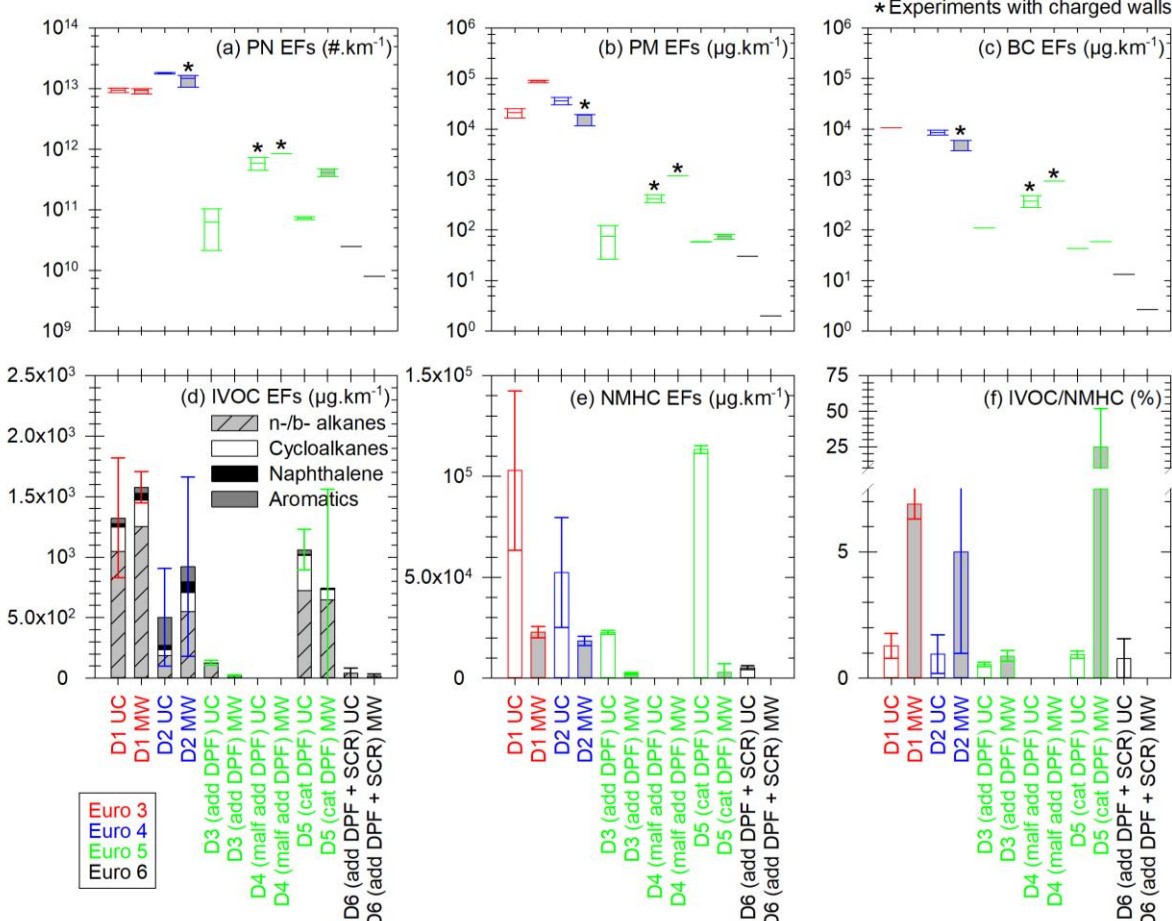

**Figure 6: Emission factors of PN (a), PM (b), BC (c), IVOCs (d) and NMHCs (e), and IVOC/NMHC ratio (f) for the 6 tested Euro 3-6 diesel vehicles UC (white) and MW (light grey) conditions. IVOC EFs (d) are the sum of EFs of linear and branched (n- and b- respectively) alkanes (hatched grey), cycloalkanes (white), naphthalene (black) and aromatics (dark grey), with retention times in the range corresponding to C12-C22 alkanes. EFs are given in µg.km$^{-1}$, and the IVOC/NMHC ratio in %. Vehicles are sorted by Euro norm: Euro 3 in red; Euro 4 in blue; Euro 5 with add or cat DPF in green and Euro 6 in black. The Euro 5 vehicle showing signs of malfunctioning DPF is referred to as "D4 (malf add DPF)". The boxes represent the 25$^{th}$ and 75$^{th}$ percentile. The solid line in the center represents the median. The whiskers are the 10$^{th}$ and 90$^{th}$ percentiles. Error bars represent 1 standard variation on figures (d) to (f). Figure (f) has a broken axis to better show the smaller values. The box plots with the sign \* represent the cycles for which chamber initial concentrations can be impacted by electrostatic charge on the walls. It could only impact PN, PM and BC, since IVOCs and NMHCs were sampled directly from emissions.**

Figure 6 shows that PM and BC EFs have a good correlation for all vehicles and driving conditions, with slightly higher emissions for PM than BC. BC accounts on average for (62 ± 34) % of PM, which is consistent with particle composition. The Euro 3 (D1) and Euro 4 (D2) vehicles emit more PN, PM and BC than the Euro 5 (D3, D4, D5) and Euro 6 (D6) ones, with factors 10–70000 (PN), 10–50000 (PM) and 4–4000 (BC). In urban conditions the Euro 5 (D3 to D5) vehicles EFs of BC, PN and PM cover a large range of values. This can be explained by the diversity and proper functioning of aftertreatment technologies for this category of vehicles (cat and add DPFs). DPFs are more effective at a certain exhaust temperature, usually not reached during a cold start urban cycle. This can lead to variations in the particle emissions. The Euro 5 vehicle with cat DPF (D5) shows PN emissions much higher in MW than in UC conditions (factor 5.7). For those cycles, the PN mode is below 19 nm, which could indicate a DPF regeneration.

IVOC and NMHC EFs are the highest for the Euro 3 (D1), Euro 4 (D2), and Euro 5 with cat DPF (D5) vehicles. They are in the range 501–1576 µg.km$^{-1}$ and (3–113)×10$^3$ µg.km$^{-1}$ for IVOCs and NMHCs respectively. IVOC

emissions from the Euro 5 vehicle with cat DPF (D5) are 5.3 times and 26.8 times higher than those from the vehicles with add DPFs (the Euro 5 D3 and the Euro 6 D6, respectively). The additive and catalyzed DPFs seem to have different impacts on IVOC and NMHC emissions. In UC conditions, the Euro 3 vehicle (D1) emits more IVOCs than the add DPF Euro 5 vehicle (D3) and the cat DPF vehicle (D5), with respective factors 10.7 and 1.2. This is close to what was reported by Zhao et al. (2015), with a ratio 7 in creep conditions between cat DPF-equipped and nonaftertreatment vehicles. Moreover, the driving conditions also impact IVOC emissions. For the DPF-equipped vehicles, IVOC EFs are 1.4 to 6.2 times higher in UC conditions than in MW conditions. This is due to more complete combustion at high temperature operations. This follows the same trend as what was reported by Zhao et al. (2015) for diesel vehicles, with ratios from 6 to 23 between creep/idle and high speed conditions. For the Euro 3 (D1) and the Euro 4 (D2) vehicles, IVOC emissions are respectively 1.2 and 1.8 times higher in UC conditions. This could be due to the cold start.

Moreover, for non-DPF vehicles (D1 and D2), IVOC emissions are dominated by n- and b- alkanes, which account for 64 %. Naphthalene accounts for 6 % of IVOC emissions. Cycloalkanes and aromatics account respectively for 14 and 16 % of IVOC emissions. These results are consistent with those obtained by Lu et al. (2018) and Zhao et al. (2015). They found IVOC emissions of non-DPF diesel vehicles to be dominated by cyclic compounds and alkanes (mainly unspeciated), with also a fraction of aromatics. For the cat DPF vehicle (D5), alkanes represent 78 % of IVOCs, followed by cycloalkanes (19 %) and then aromatics and naphthalene (2 and 1 % respectively). For the vehicles with add DPFs (D3 and D6), only alkanes are identified in the IVOC retention time range.

The IVOC/NMHC ratios range from $(0.5 \pm 0.1)$ % to $(6.9 \pm 0.6)$ % for all vehicles, except for the cat DPF Euro 5 (D5) one in MW conditions, with a ratio of $(24.7 \pm 27.1)$ %. The possible regeneration observed from PN emissions for this vehicle in MW conditions could explain high IVOC emissions, and therefore a high IVOC/NMHC ratio. The average IVOC/NMHC ratio is $(3.5 \pm 2.9)$ % for the vehicles without DPF (D1 and D2), and $(0.8 \pm 0.2)$ % for the DPF-equipped vehicles (D3, D5, D6), without the possible regeneration value. This is substantially lower than the values of $(60 \pm 10)$ % and $(150 \pm 80)$ % found by Zhao et al. (2015), respectively for nonaftertreatment and DPF-equipped diesel vehicles. However, respectively only about $(8.1 \pm 2.3)$ % and $(5.6 \pm 3.1)$ % of the total IVOC mass given by Zhao et al. (2015) was identified. This brings the IVOC/NMHC ratios to $(4.9 \pm 2.2)$ % (nonaftertreatment) and $(8.4 \pm 9.1)$ % (DPF-equipped), if considering only the identified portion of IVOCs. This is higher than what is found in this study, but remains in the same range if uncertainties are accounted for.

Certain n- and b- alkanes corresponding to SVOCs are also sampled on the sorbent tubes. Their emissions are higher for the Euro 3 (D1), Euro 4 (D2) and cat DPF Euro 5 (D5) vehicles, with average EFs of 1093 and 1520 µg.km⁻¹ in UC and MW conditions respectively. For the Euro 5 (D3) and Euro 6 (D6) vehicles, both equipped with add DPFs, the average SVOC EFs are 263 and 247 µg.km⁻¹ in UC and MW conditions respectively. However, their quantification is performed with response factors from compounds with different carbon numbers (C20 for vehicles D1, D5 and C16 for vehicles D2, D3, D6). This can induce important uncertainties in their EFs.

Considering the French fleet (André M. et al., 2014) and EFs of Figure 6, the contribution of each Euro norm to particle and IVOC emissions from diesel passenger cars is estimated. Since no vehicle of norms pre-Euro, Euro 1 and Euro 2 was tested, their emissions are assumed to be in the same range as those from Euro 3 vehicles. They are grouped in a category named Euro pre-1-2-3. Figure 7a gives the French passenger car fleets from 2015 to 2030 (André M. et al., 2014). Figure 7b and Figure 7c give the evolution of total PM and IVOC emissions,

computed as the product of emission factors and fleet composition (assuming a constant number of vehicles). Figure 7d and Figure 7e show the contribution of each Euro norm to total diesel passenger car emissions of PM and IVOCs.

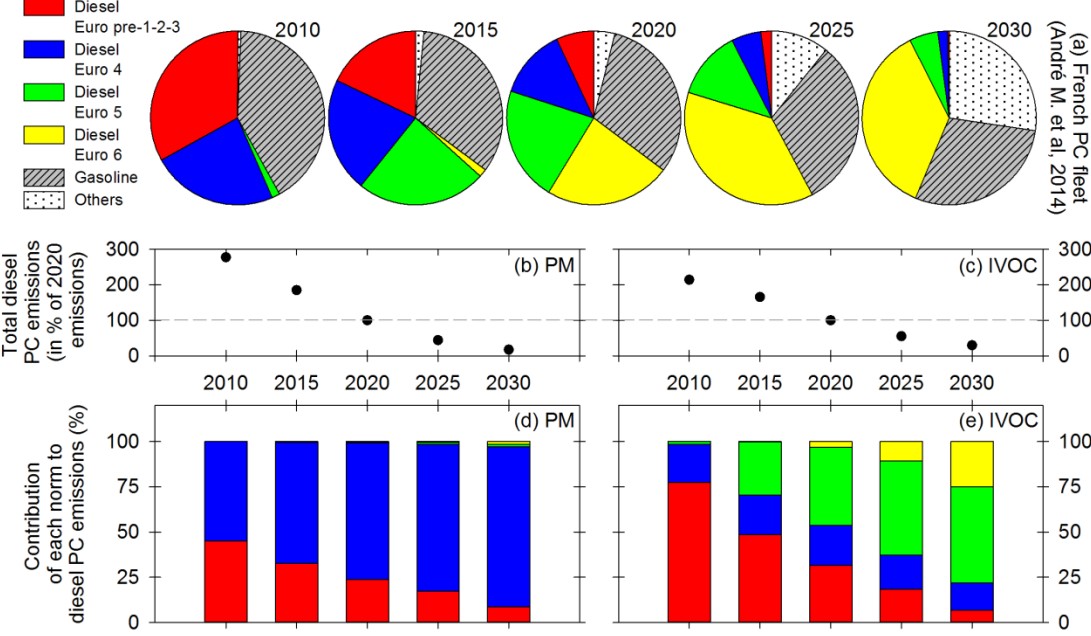

**Figure 7. Evolution of the French passenger car fleet (a) given by André M. et al. (2014), for diesel vehicles of norm Euro pre-1-2-3 (red), Euro 4 (blue), Euro 5 (green) and Euro 6 (yellow), as well as gasoline vehicles (hatched grey) and vehicles with other motorization like hybrid or electrical (dotted white). Total emissions of PM (b) and IVOCs (c) by diesel PCs, given as % of the 2020 emissions. Contribution of each Euro norm to diesel PC emissions of PM (d) and**
595 **IVOCs (e). Results are given for the years 2010, 2015, 2020, 2025 and 2030.**

Results show that total diesel PC emissions decrease by a factor 15.8 for PM and 7.2 for IVOCs between 2010 and 2030, assuming the total number of vehicles remains constant. This is due to evolution in the PC fleet composition, with diesel vehicles representing 58 % of the PC fleet in 2010, 65 % in 2020 and 44 % in 2030. Moreover, the share of more polluting diesel PCs (Euro pre-1-2-3 and Euro 4) evolves, representing 56 % of the fleet in 2010,
20 % in 2020 and 2 % in 2030. However, due to their much higher PM emissions (compared to Euro 5 and Euro 6 diesel vehicles), their share in total PM emissions remains dominant: almost 100 % in 2010, 99 % in 2020 and still 97 % in 2030.

For IVOCs, emissions are also due to DPF-equipped vehicles (Euro 5 and Euro 6). The share of those vehicles in IVOC emissions by diesel PCs is less than 1 % in 2010, 46 % in 2020 and 78 % in 2030. More modern vehicles
will be the main contributor to IVOC emissions in 2030, with effects on particle physical and photochemical evolutions. More investigations should therefore be conducted to estimate their emissions and evolutions.

### 3.3. Evolution of particles from diesel exhaust

In this part, the physical processes having effects on particle number, mass and size are investigated, for the 6 diesel vehicles (Euro 3 to Euro 6) in both UC and MW conditions. Data are corrected for leakage and wall losses,
using the new 4-step correction method.

Figure 8a gives the hourly increase of [PM], for the vehicles classified by Euro norm. It shows that the Euro 3 and Euro 4 vehicles have the higher [PM] hourly increases. The increase of [PM] observed in this section is not likely to be an artifact induced by the correction method, as indicated by the observation of measured (e.g. uncorrected)

data. This is discussed in Appendix F. Increases are at least of 5 %.h$^{-1}$ during more than 500 min, and up to 17 %.h$^{-1}$ during 100 minutes for the Euro 3. The Euro 4 vehicle has hourly [PM] increases in the same range, and undergoes an increase during an average of 400 minutes. This increase in [PM] could in part be explained by condensation of organic material onto preexisting particles (IVOCs and SVOCs). The partitioning of IVOCs, and potential role in [PM] evolution is discussed in Appendix G. The emissions of IVOCs observed in Figure 6 for the Euro 3 and Euro 4 vehicles seem to confirm this explanation. The IVOC emissions from the Euro 5 vehicles could explain the slight increase in particle mass during evolution, presented in Figure 8a. This increase for the Euro 5 vehicles can go up to 5 %.h$^{-1}$ during almost 5 hours. The Euro 6 vehicle however, does not undergo any [PM] increase, which is consistent with the very low emissions of precursors. The observed [PM] increase trends can be explained by different processes. First of all, condensation of organics (SVOCs and IVOCs) onto preexisting particles is the most obvious explanation. However, it is not likely to occur during several hours without the presence of a source. One hypothesis could be to consider the walls as a source. Indeed, organics deposit onto Teflon walls, and this process in known to be reversible (Matsunaga and Ziemann ‡, 2010). In addition to being a sink for the gas-phase, the walls could also represent a source of pollutants (Kaltsonoudis et al., 2019). Even though deposition is generally estimated to lower SOA formation, the impact of vapor deposition of organics onto the walls is not well described yet, and many parameters remain uncertain (Pratap et al., 2020; Yeh and Ziemann, 2015). Namely, Zhang et al. (2015) observed evaporation from the walls when the temperature increases from 25 to 45 °C. In the Euro 3 and Euro 4 experiments, the temperature increases slowly during evolution time (due to instrumentation in the laboratory), of 5 °C on average. This is a slight increase, and even though it is to this day impossible to quantify the mass of organics which could evaporate, the walls of the chamber may in some cases be a source of organic material. Moreover, vapor wall deposition is a competing process with condensation onto particles. At high particle concentrations, vapor wall deposition becomes less significant (Zhang et al., 2015, 2014). This could explain why [PM] increase is higher when initial particle concentrations are high. Since for low particle concentrations the share of organics depositing onto the walls is more important, it has more impact on [PM] evolution. Overall, the walls could play the role of source of organic material, which might partly explain the increase of [PM] during several hours. Finally, when concentrations of organics are high enough, nucleation may occur, leading to nucleation mode particles. If those particles are too small to be detected by the SMPS, they can grow due to coagulation or condensation and at some point, be detected by the SMPS. This would result in an increase of particle number and particle mass. This phenomenon is discussed in Appendix H. It can be part of the interpretation of [PM] increase. However, considering the size of such particles, and their low relative mass (compared to bigger particles), this phenomenon is not likely to be responsible for much of the total [PM] increase.

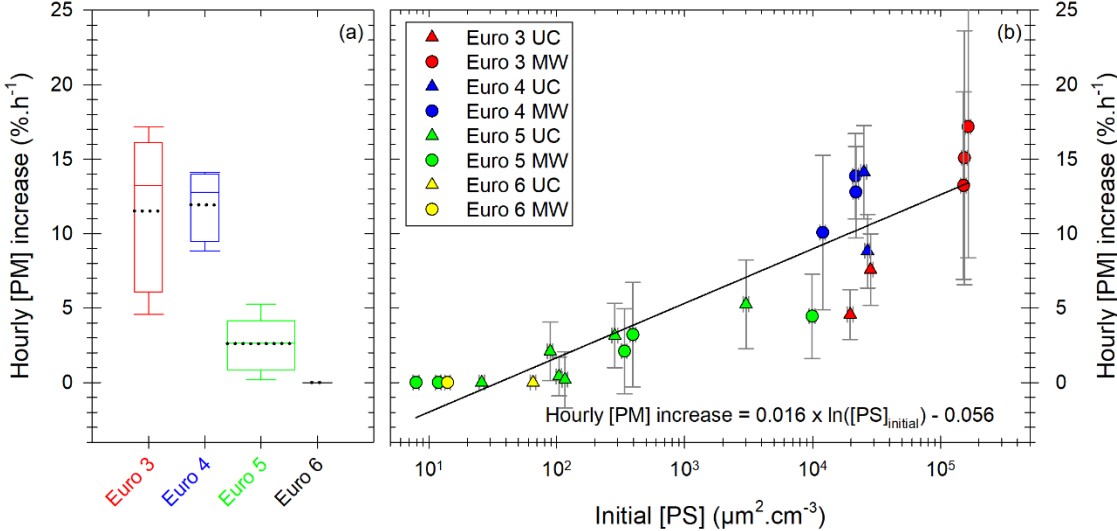

**Figure 8: Particle mass hourly increase in % of initial [PM] per hour. Results in (a) are given for all the diesel vehicles sorted by Euro norm. The boxes represent the 25th and 75th percentile. The solid line in the center represents the median; the dashed line represents the mean. The whiskers are the 10th and 90th percentiles. The width of the boxes is proportional to the data set of the box. Results in (b) are given as a function of initial particle surface [PS] for all Euro norms (Euro 3 in red; Euro 4 in blue; Euro 5 in green; Euro 6 in yellow or black) and in both UC (triangles) and MW (circles) conditions. The x-axis is presented with a logarithmic scale. A logarithmic fit of the data is performed (black line) with equation written in figure (b). Error bars on figure (b) represent 1 standard variation.**

Figure 8b shows the [PM] hourly increase as a function of initial particle surface [PS]. A logarithmic correlation appears, indicating that conditions with high initial particle surface are more likely to lead to an increase in [PM]. The high hourly [PM] increases (5-17 %.h$^{-1}$) are mainly due to Euro 3 and Euro 4 vehicles, whereas Euro 5 and Euro 6 vehicle experiments results in lower [PM] increases (0-5 %.h$^{-1}$). The correlation can be explained by the fact that high initial [PS] results in high probability for organic material to find available surface for condensation. The logarithmic shape indicates that at some point, [PM] increase is limited by other factors. This regime is reached when [PS] is above $\sim 10^4$ µm$^2$.cm$^{-3}$. A limiting factor could be the availability of organic material. Similar trends were observed by Charan et al.(2020), with initial surface areas above $\sim$ 1800 µm$^2$.cm$^{-3}$ becoming insignificant for SOA yields (the numerical difference is not surprising considering the fact that the experimental protocol is quite different from that of this study). This could explain why the 3 points with higher initial [PS] (1 order of magnitude above the preceding ones) have hourly [PM] increase in the same order of magnitude as the preceding points.

In addition to the hourly [PM] increase, the time during which condensation occurs is an important piece of information. Results giving the time needed to reach the maximum of PM concentration are displayed in Figure 9a. It is plotted as a function of the initial particle surface, and the labels give the hourly [PM] increase.

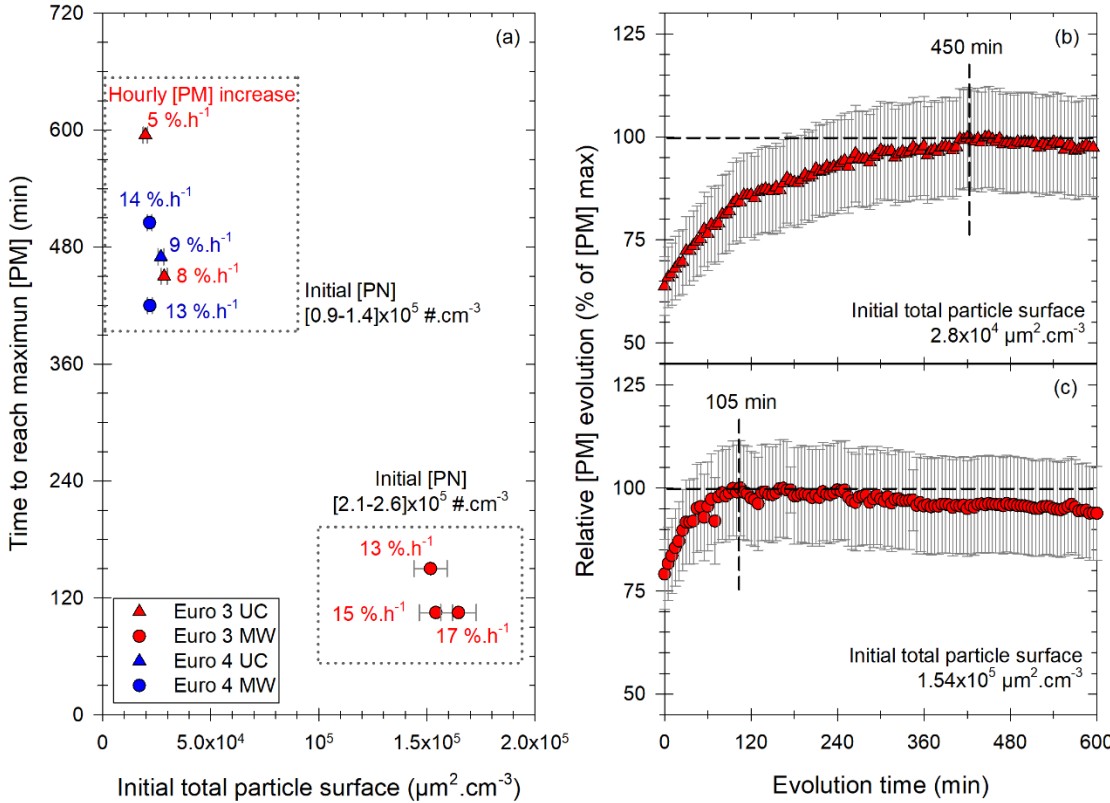

**Figure 9: Results of the study of the [PM] evolution in the chamber for experiments showing a significant increase with the Euro 3 (red) and the Euro 4 (blue) vehicles, in UC (triangles) or MW (circles) conditions. Figure (a) gives the time to reach the maximum of [PM] as a function of initial total particle surface. Each point has a label giving the hourly [PM] increase in % of initial [PM] per hour. Two groups are found: the first one (top) for initial [PN] between 0.9 and 1.4×10⁵ #.cm⁻³; the second one (bottom) for initial [PN] between 2.1 and 2.6×10⁵ #.cm⁻³. Figures (b) and (c) give 2 examples of those [PM] evolutions with time, plotted as relative evolution of the maximum [PM]. Examples are taken: (b) from the first group of figure (a) with initial surface of 2.8×10⁻⁴ μm².cm⁻³ and a maximum [PM] reached in 450 min; and (b) from the second group of figure (a) with initial surface of 1.54×10⁵ μm².cm⁻³ and a maximum [PM] reached in 105 min.**

Figure 9 clearly shows that when the available particle surface is high (> $10^5$ µm$^2$.cm$^{-3}$), the time to reach the maximum [PM] is quite short, and doesn't exceed 2.5 hours. However, for initial surfaces below $4\times10^4$ µm$^2$.cm$^{-3}$, [PM] increase seems to be a very slow process. Even though uncertainties become large after 6 hours of evolution, it seems that [PM] increase can occur slowly during 7 to 10 hours. This difference between fast and slow [PM] increases is shown in Figure 9b and Figure 9c, for 2 examples of [PM] evolution with time. [PM] evolution is given relatively, as a percentage of [PM] max, for 2 experiments with low (Figure 9b) and high (Figure 9c) initial particle surface. The first curve (Figure 9b) shows slow [PM] increase, with the maximum reached after 450 min. The second curve (Figure 9c) however shows a fast evolution, with an increase occurring during 105 minutes. This fast [PM] increase seems to be mostly determined by the high particle surface available. By taking the average initial particle surface and the average time to reach maximum [PM] for both groups (fast and slow [PM] increases), it appears that [PM] increase is 4 times faster when the initial total particle surface is multiplied by 6.5. Other than condensation, nucleation and coagulation are 2 processes that can have important effects on particle concentrations and distributions. Both of those processes are investigated, and results are shown in Figure 10.

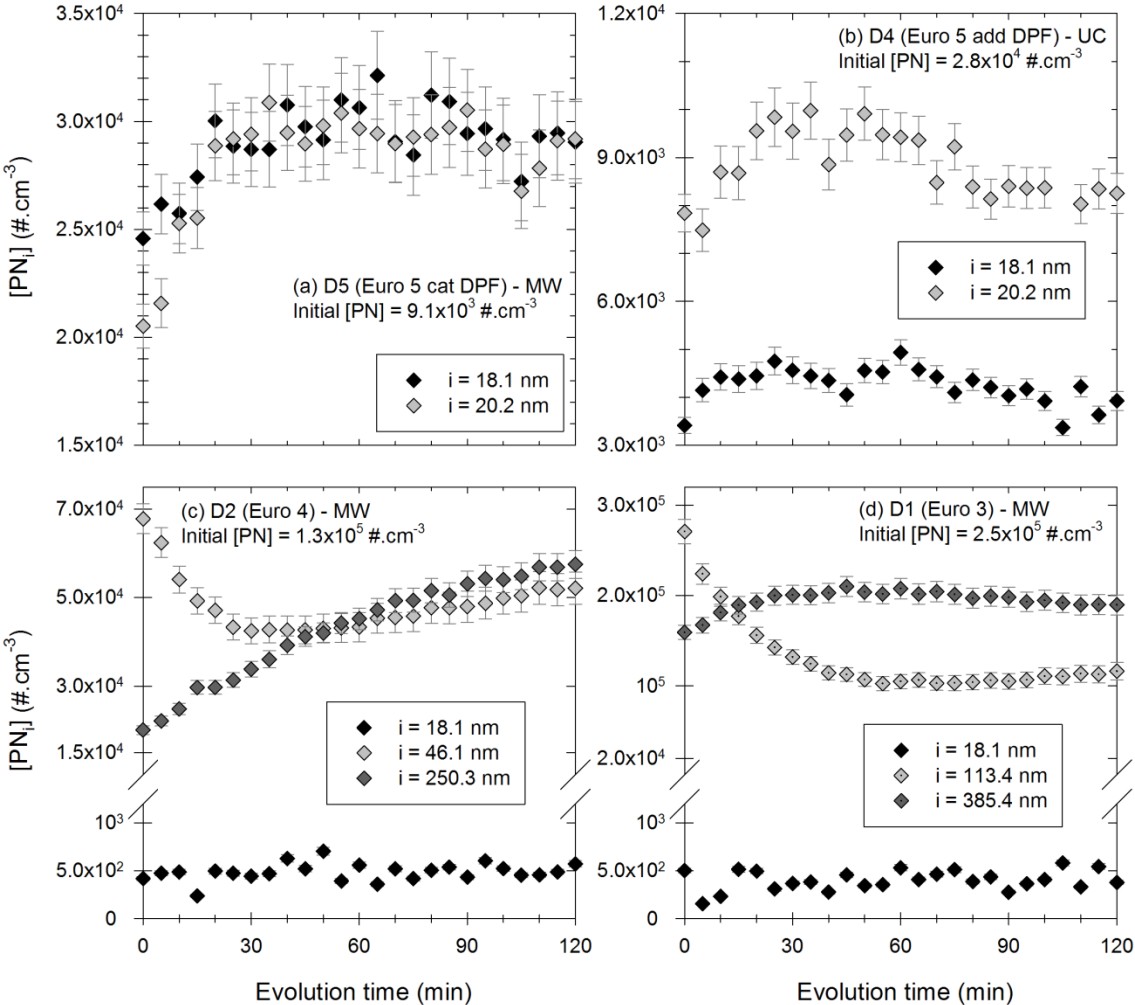

**Figure 10: Evolution with time of the particles [PN$_i$] in certain diameter bins during 2 hours. Figures (a) and (b) show particles in the bins 18.1 nm (black) and 20.2 nm (light grey), for experiments with the vehicle D5 in MW conditions with an initial [PN] of 9.1×10³ #.cm⁻³ and with the vehicle D4 in UC conditions with an initial [PN] of 2.8×10⁴ #.cm⁻³, respectively. Figure (c) shows particles in the bins 18.1 nm (black), 46.1 nm (light grey) and 250.3 nm (dark grey) with the vehicle D2 in MW conditions with an initial [PN] of 1.3×10⁵ #.cm⁻³. Figure (d) shows particles in the bins 18.1 nm (black), 113.4 nm (light grey) and 385.4 nm (dark grey) with the vehicle D1 in MW conditions with an initial [PN] of 2.5×10⁵ #.cm⁻³.**

Figure 10a and Figure 10b show the evolution of small particles (in the bins centered at 18.1 and 20.2 nm), for 2 experiments with respective initial [PN] of 9.1×10³ #.cm⁻³ and 2.8×10⁴ #.cm⁻³. In both cases, concentrations increase during the first 30 to 60 minutes of the evolution. The increase of particles with such small diameters could be due to growth of smaller particles, which are initially too small to be detectable by the SMPS. Such particles could be nucleation particles, formed shortly after injection. Their growth due to coagulation or condensation could explain the increase of 18-20 nm particles of Figure 10a and Figure 10b. These figures seem to indicate the presence of nucleation mode particles formed shortly after injection. This is discussed below with results of Figure 11.

Figure 10c and Figure 10d give the evolution of the particles in 3 diameter bins, for experiments with initial [PN] of 1.3×10⁵ #.cm⁻³ and 2.5×10⁵ #.cm⁻³ respectively. In Figure 10c, the particles around 18.1 nm have a constant concentration. This could mean that they coagulate, and that this coagulation is compensated by the growth of smaller particles, initially undetected by the SMPS. This means that there might also be nucleation mode particles in those conditions. This is consistent with particle size distributions presented in Appendix H. It would however

have limited impact on particle number evolution. A small nucleation mode in those high concentration conditions is also consistent with the [PM] increase. Indeed, at high particle concentrations, organic material is more likely to condense rather than nucleate. Moreover, the concentration of the particles in the bin 46.1 nm decrease rapidly during 20 to 30 minutes. Simultaneously, the concentration of larger particles (in the bin 250.3 nm) increases. This seems to indicate that particles around 46.1 nm coagulate to form larger particles. The same trends are observed in Figure 10d, with coagulation of larger particles (around 113.4 nm) forming particles with diameters around 385.4 nm. In both cases (Figure 10c and Figure 10d), there might be nucleation mode particles formed at the beginning of the evolution, without a significant impact on particle number evolution. Coagulation of particles is however an important process.

As those 2 processes (coagulation of the particles in the SMPS range and growth of nucleation mode particles) impact the total number of particles, the conditions in which they occur are investigated with regards to total [PN] evolutions. Figure 11 shows the [PN] evolution (at the beginning of the evolution) as a function of initial [PN]. It also illustrates the different profiles that [PN] evolutions can have for several initial PN concentrations.

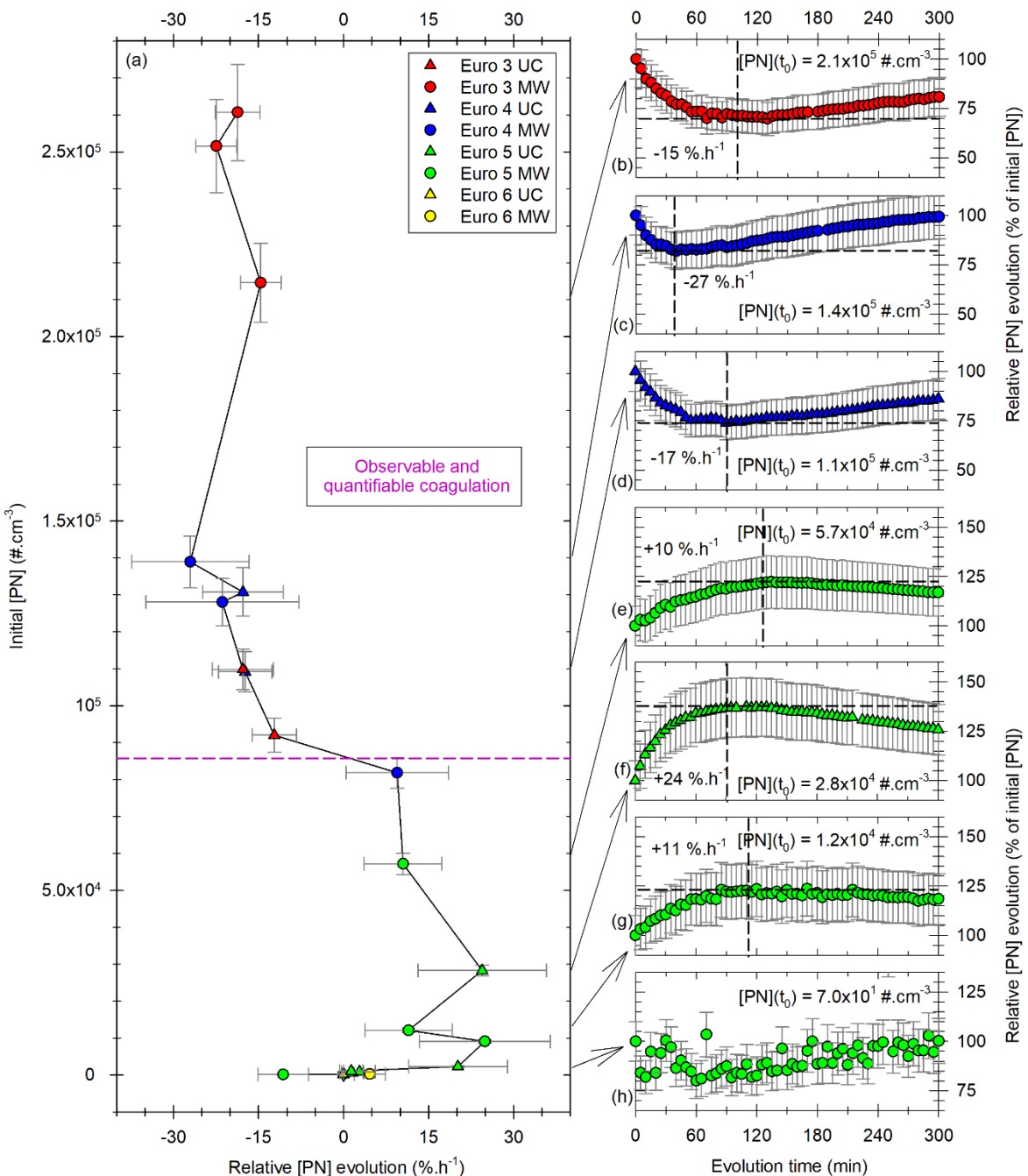

**Figure 11: Evolutions of total [PN] for the vehicles of all Euro norms (Euro 3 in red; Euro 4 in blue; Euro 5 in green; Euro 6 in yellow) and in both UC (triangles) and MW (circles) driving. Figure (a) on the y-axis the initial particle concentration, plotted versus the [PN] evolution, expressed in % of the initial concentration per hour. The horizontal pink dashed line shows the concentration above which it is estimated that coagulation becomes observable and**
730 **quantifiable with this chamber. Figures (b) to (h) show examples of such [PN] evolutions with time, expressed relatively as a percentage of initial [PN], for several initial concentrations ranging between $7.0 \times 10^1$ and $2.1 \times 10^5$ #.cm$^{-3}$. The vertical dashed lines indicate the time at which the [PN] evolution stops or changes, with the percentage of initial [PN] reached at this time shown by the horizontal dashed line, and with the average hourly evolution during this period.**

Figure 11a shows on the x-axis the relative [PN] evolution, in % of the initial concentration per hour, for different
initial particle concentrations on the y-axis. It shows that at low initial PN concentrations (up to ~$10^3$ #.cm$^{-3}$), [PN] evolution is not significant, and is either positive or negative, but remains low. Figure 11h shows an evolution with low initial concentration. [PN] remains quite steady during the 5 hours of evolutions. When initial [PN] increase, in the range $10^3$ up to $[8-9] \times 10^4$ #.cm$^{-3}$, PN concentrations increase as well, up to 24 %.h$^{-1}$. Figure 11e, Figure 11f

and Figure 11g show relative [PN] evolution with initial [PN] in that range ($10^3$ to [8-9]$\times10^4$ #.cm$^{-3}$). They show medium to high particle increase. This occurs between 90 and 120 min. This increase could be explained by the growth of nucleation mode particles (formed at the beginning of the evolution) which are initially not detectable by the SMPS (as mentioned above). Ning and Sioutas (2010) showed that organic vapors nucleate more easily when particle concentrations are low, and condense more easily when particle concentrations are high. Imhof et al. (2006) showed that nucleation would occur more easily under low traffic density (e.g. with lower initial particle concentrations). Moreover, they found that nucleation mode particles increase until the surface area of soot particles reaches a threshold of $0.5\times10^4$ $\mu m^2$.cm$^{-3}$. In this study, nucleation particles seem to be observed and significant for initial [PN] below [8-9]$\times10^4$ #.cm$^{-3}$. The equivalent surface concentration is [1.2-2]$\times10^4$ $\mu m^2$.cm$^{-3}$. This is in the same order of magnitude (factor 2-4) as the threshold found experimentally by Imhof et al. (2006). This confirms that a nucleation mode can be present in our experimental conditions. This nucleation mode would grow due to coagulation or condensation, and become detectable by the SMPS. It would explain the increase of [PN] in those conditions, during 90 to 120 min. Guo et al. (2020) observed growth of nucleation particles during about 2 hours. This seems to confirm the interpretation of [PN] increase due to the growth of nucleation mode particles becoming detectable by the SMPS. Therefore, the observed [PN] increase in this work is not a physical creation of particles during several hours, since nucleation would occur rapidly after injection. Once the nucleation mode is formed, the actual trend of total particle evolution would be a decrease due to coagulation. The increase of [PN] which is observed is therefore an artifact due to the measurement range of the SMPS. This artifact is predominant for initial concentrations below [8-9]$\times10^4$ #.cm$^{-3}$. In those conditions, the process of coagulation cannot be quantified with this chamber and the experimental set up described in this study.

When initial particle concentration is over [8-9]$\times10^4$ #.cm$^{-3}$, the main evolution becomes negative. It remains negative during 40 to 120 minutes. It can reach -27 %.h$^{-1}$, clearly showing that coagulation is occurring. Figure 11b, Figure 11c and Figure 11d gives examples of the [PN] decrease for those high PN initial concentration conditions. In those conditions of initial concentrations, growth of nucleation particles doesn't seem to be significant, as seen in Figure 10. The evolution of [PN] is therefore mainly explained by the process of coagulation of the particles in the SMPS range. In those conditions, this process can be observed and quantified with this chamber and experimental set up. A slight increase can also be observed in the second half of the experiments. It could in part be explained by growth of nucleation mode particles. As discussed for Figure 10c and d, the nucleation mode would be relatively small in those conditions, with limited effects. It would therefore coagulate slowly, thus explaining why the increase is observed at the end of the experiments. Also, considering the error bars, which are more important when the [PN] becomes lower (e.g. when time increases), the slight increase doesn't seem to be significant.

Moreover, the characteristic timescale for coagulation (time necessary for reduction of initial [PN] by a factor 2) was computed using Eq. (13.67) from Seinfeld and Pandis (2016), assuming monodisperse distributions, and considering initial particle concentration. For the experiments with initial concentrations below [8-9]$\times10^4$ #.cm$^{-3}$, the range of characteristic timescale indicates that the time needed to reach 50 % of the initial concentration is between 11 and 40 000 hours. This could explain why coagulation is not observed for low initial concentrations in Figure 11a. For the experiments with initial concentrations above [8-9]$\times10^4$ #.cm$^{-3}$, the characteristic timescale ranges between 3.4 and 9.7 hours. This means that significant [PN] decrease due to coagulation can occur in a few hours. It confirms that coagulation is observable in our experimental conditions with high initial concentrations,

as shown in Figure 11a. Coagulation therefore seems to be more significant for initial concentrations above
[8-9]$\times 10^4$ #.cm$^{-3}$. With the chamber and experimental set up of this study, coagulation is observable and
quantifiable only for initial concentrations above this value.

## 4.        Conclusion

This study presents the characterization of a new 8 m$^3$ environmental chamber with Teflon walls, meant to study
the physical evolution of primary pollutants emitted by road traffic. A new size-dependent method to correct
particle losses due to leakage and wall deposition was developed and applied. It accounts for experiment-to-
experiment variations. It consists in 4 steps, using the [BC] decay and wall loss coefficients from the theory of
Crump and Seinfeld (1981). These 2 complementary parts are in good agreement. The total loss coefficients
$\alpha+\beta_{ke}^{mean}$ of "charged walls" experiments are in the range [4.19-11.05]$\times 10^{-3}$ min$^{-1}$ for diesel and
[2.04-5.01]$\times 10^{-3}$ min$^{-1}$ for gasoline. For "neutral walls" experiments, they are in the range [1.25-4.09]$\times 10^{-3}$ min$^{-1}$
for diesel, [0.93-3.93]$\times 10^{-3}$ min$^{-1}$ for gasoline and [0.63-1.01]$\times 10^{-3}$ min$^{-1}$ for ammonium sulphate. The wall
charges appear to be the most important factor affecting particle wall losses. It is responsible for 76 to 93 % of
total losses for exhaust experiments, and for 55 % in the case of ammonium sulphate experiments. Results of wall
losses obtained from ammonium sulphate particle experiments show similar trends as those found in the literature
for a chamber of comparable size.

EFs of PN, PM, BC and IVOCs for Euro 3 to Euro 6 diesel vehicles were studied in detail in order to understand
their role on particle evolution in dark conditions. For non-DPF vehicles (D1 and D2), IVOC emissions are
dominated by n- and b- alkanes, (64 %), followed by cycloalkanes (14 %) and aromatics (16 %). For the cat DPF
vehicle (D5), alkanes represent 78 % of IVOCs, followed by cycloalkanes (19 %) and then aromatics (2 %). For
the vehicles with add DPFs (D3 and D6), only alkanes were identified in the IVOCs.

Moreover, this study presents results of evolution in the dark of particles emitted by diesel passenger cars. It is
shown that [PM] increase can reach 17 %.h$^{-1}$, without the effects of photochemistry. Uncorrected measurements
confirm that this is not an artifact induced by the correction method. It could in part be explained by the walls
playing the role of source of organic material. [PM] increase appears to have a logarithmic correlation with initial
[PS]. The [PM] increase of older diesel vehicles (Euro 3 and Euro 4) is enhanced in comparison to modern vehicles.
This is due to their higher emissions of BC, PN and PM, compared to DPF-equipped vehicles, as well as their
emissions of IVOCs. In several cases with high initial [PM], a fraction of IVOCs (0.8 to 34.0 %) and SVOCs (85.5
to 99.9 %) can be found in the particle phase, and participate in the physical evolution of particles. The increase
of [PM] is also found to be 4 times faster when the available surface is multiplied by 6.5. However, when initial
[PS] is above ~ 10$^4$ $\mu$m$^2$.cm$^{-3}$, [PM] increase seems to be limited. The quantity of organic material available could
be a limiting factor. Finally, this study shows that nucleation mode particles are likely to be present for initial
concentrations below [8-9]$\times 10^4$ #.cm$^{-3}$. The growth of those particles results in an increase of measured [PN]. This
is an artifact of the experimental set up, due to the measurement range of the SMPS, and not an actual creation of
particles during several hours. This artifact prevents the study of the process of coagulation in this chamber when
initial concentrations are below [8-9]$\times 10^4$ #.cm$^{-3}$. However, above this value, coagulation is the dominant process
for [PN] evolutions. It leads to [PN] decrease up to -27 %.h$^{-1}$. In those conditions, this chamber and experimental
set up are well suited for the observation and quantification of the coagulation process.

Results found in this study under laboratory conditions are in good agreement with tunnel observations described in the literature. They can help better understand the conditions under which physical processes are more likely to occur. They can be applied to several conditions in the dark, such as winter rush hours or tunnel evolutions.

*Data availability*. All data from this study are available from the authors upon request.

*Supplement*.

*Author contribution*. YL, BD, CF, KS, CL and BV designed the research. YL, CL and BV performed and analyzed the characterization experiments. YL and BV performed and analyzed the emission and evolution experiments. CF, LF and BV performed the ATD-GC-MS analysis. EK, BTR, BD contributed to the experimental set up and the experimental procedure. PT and PP drove the cars and helped with experimental set up. BV synthesized all the data and wrote the paper with contributions from CL, VC, BD, EK, BTR, KS, CF and YL.

*Competing interests*. The authors declare that they have no conflict of interest.

*Acknowledgements*.

*Financial support*. This work was supported by the ADEME CORTEA program with the project CAPVEREA and MAESTRO and the ANR program with the project POLEMICS.

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

## Appendix A

Below are shown the speed profiles of the Artemis Urban and Artemis Motorway cycles.

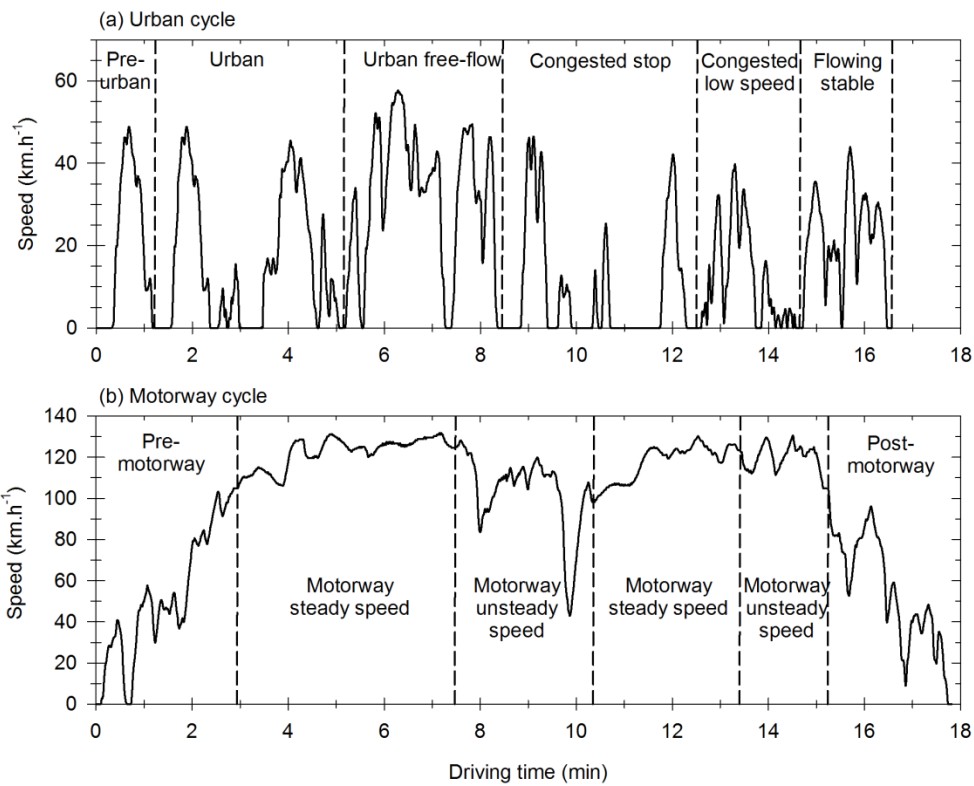

**Figure A1: Speed in km.h$^{-1}$ during the Artemis Urban cycle (a) and the Artemis Motorway cycle (b), with the different types of driving found in those conditions.**

## Appendix B

Below are listed the constants taken for the calculation of $\beta(D_p)$:

- $k_B = 1.381 \times 10^{-23}$ J.K$^{-1}$ the Boltzmann constant

- T (K) the temperature in the chamber for each experiment

- $\mu = 1.8 \times 10^{-5}$ kg.m$^{-1}$.s$^{-1}$ the viscosity of air

- $\rho_p = 1.2$ g.cm$^{-3}$ the particle density

- g=9.81 m.s$^{-2}$ the g-force

- λ=6.51 nm the mean-free path of air

**Appendix C**

**Table C1: Emission factors in mg.km$^{-1}$ of: linear (n-) and branched (b-) alkanes with carbon number below between 12 and 22 (C12-C22), of naphthalene, cycloalkanes and aromatics measured with the ATD-GC-MS method. The sum of those EFs gives the IVOC EFs. Emission factors of n- and b- alkanes with carbon number above 22. Emission factors of NMHCs measured by flame ionization detection. Emissions below quantification limit are indicated as "< QL". Ratio of IVOC/NMHC expressed as % of NMHCs. Results are shown for vehicles D1 (Euro 3), D2 (Euro 4), D3 (Euro 5 add DPF), D5 (Euro 5 cat DPF) and D6 (Euro 6 add DPF and SCR), in both UC and MW conditions.**

| Vehicle | D1 | | D2 | | D3 (add DPF) | | D5 (cat DPF) | | D6 (add DPF and SCR) | |
|---|---|---|---|---|---|---|---|---|---|---|
| Cycle | UC | MW | UC | MW | UC | MW | UC | MW | UC | MW |
| n-/b- alkanes C12-C22 (mg.km$^{-1}$) | 1.05 ± 0.31 | 1.25 ± 0.08 | 0.19 ± 0.03 | 0.55 ± 0.48 | 0.12 ± 0.02 | 0.02 ± 0.01 | 0.72 ± 0.10 | 0.65 ± 0.68 | 0.04 ± 0.04 | 0.02 ± 0.02 |
| Naphthalene (mg.km$^{-1}$) | 0.03 ± 0.03 | 0.06 ± 0.02 | 0.04 ± 0.01 | 0.09 ± 0.05 | <QL | <QL | 0.01 ± 0.01 | <QL | <QL | <QL |
| Cycloalkanes (mg.km$^{-1}$) | 0.2 ± 0.10 | 0.22 ± 0.01 | 0.05 ± 0.04 | 0.16 ± 0.16 | <QL | <QL | 0.29 ± 0.06 | 0.08 ± 0.12 | <QL | <QL |
| Aromatics (mg.km$^{-1}$) | 0.04 ± 0.49 | 0.05 ± 0.13 | 0.23 ± 0.40 | 0.12 ± 0.74 | <QL | <QL | 0.04 ± 0.17 | 0.01 ± 0.82 | <QL | <QL |
| IVOCs (mg.km$^{-1}$) | 1.32 ± 0.93 | 1.58 ± 0.24 | 0.5 ± 0.48 | 0.92 ± 1.42 | 0.12 ± 0.04 | 0.02 ± 0.01 | 1.06 ± 0.34 | 0.74 ± 1.62 | 0.04 ± 0.08 | 0.02 ± 0.04 |
| n-/b- alkanes C>22 (mg.km$^{-1}$) | 1.56 ± 1.26 | 1.27 ± 1.35 | 0.26 ± 0.30 | 0.66 ± 0.01 | 0.35 ± 0.06 | 0.29 ± 0.43 | 1.46 ± 1.47 | 2.63 ± 3.17 | 0.17 ± 0.12 | 0.2 ± 0.24 |
| NMHCs (mg.km$^{-1}$) | 102.93 ± 39.43 | 22.82 ± 2.79 | 52.31 ± 27.29 | 18.38 ± 2.29 | 22.68 ± 0.89 | 2.25 ± 0.53 | 113.37 ± 2.00 | 3.01 ± 4.14 | 5.18 ± 0.96 | No data |
| IVOC/NMHC (%) | 1.28 ± 0.48 | 6.91 ± 0.59 | 0.96 ± 0.77 | 5.00 ± 4.02 | 0.54 ± 0.09 | 0.88 ± 0.21 | 0.93 ± 0.15 | 24.73 ± 27.10 | 0.79 ± 0.78 | No data |

* QL = Quantification Limit

**Appendix D**

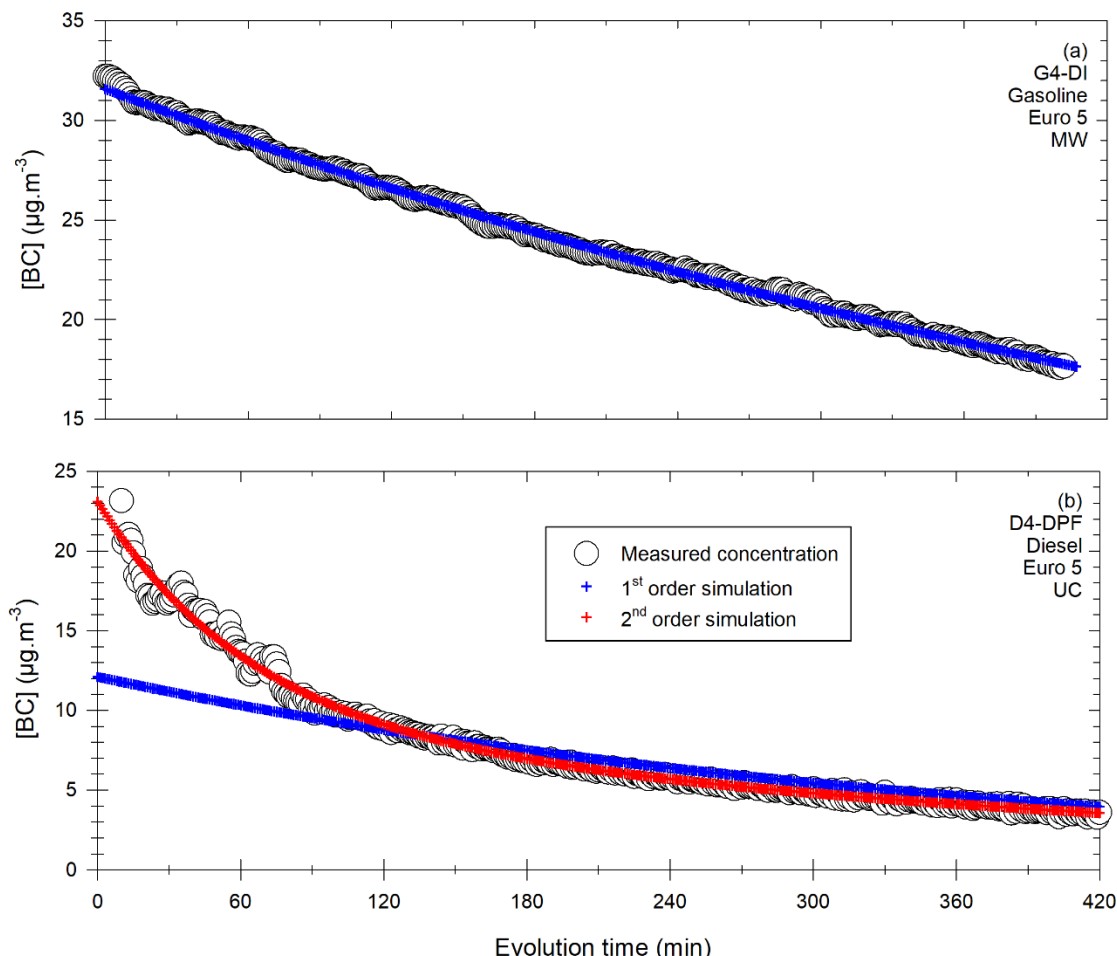

**Figure D1: Evolution of the measured BC concentration (white cercles) for the G4-DI vehicle in MW conditions (a) and the D4-DPF vehicle in UC conditions (b). The curve is fitted with a 1st order exponential decay (blue) in both cases, and with a 2nd order exponential decay (red) on figure (b).**

Figure D1a and Figure D1b show 2 examples of [BC] evolutions and decay fits. Figure D1a is an example of an evolution well fitted by a 1st order exponential decay. Figure D1b shows an evolution not well fitted by a 1st order exponential decay. For instance, the fitted initial concentration is 12.1 µg.m$^{-3}$ whereas the actual initial concentration is 23.2 µg.m$^{-3}$. This means that this fit would induce an error of 48 % on the initial concentration. However, the 2nd order exponential decay matches very well the measured values.

**Appendix E**

To verify how well [PM] corrections from steps 1 and 2 agree, the average relative difference between $[PM]_{corrected}^{step1}(t)$ and $[PM]_{corrected}^{step1}(t)$ is computed as given in Eq. (E1).

$$\frac{1}{n} \times \left( \sum_{i=0}^{n} \frac{\left| [PM]_{corrected}^{step\ 1}(t_i) - [PM]_{corrected}^{step\ 2}(t_i) \right|}{[PM]_{corrected}^{step\ 1}(t_i)} \times 100 \right) \tag{E1}$$

      **Appendix F**

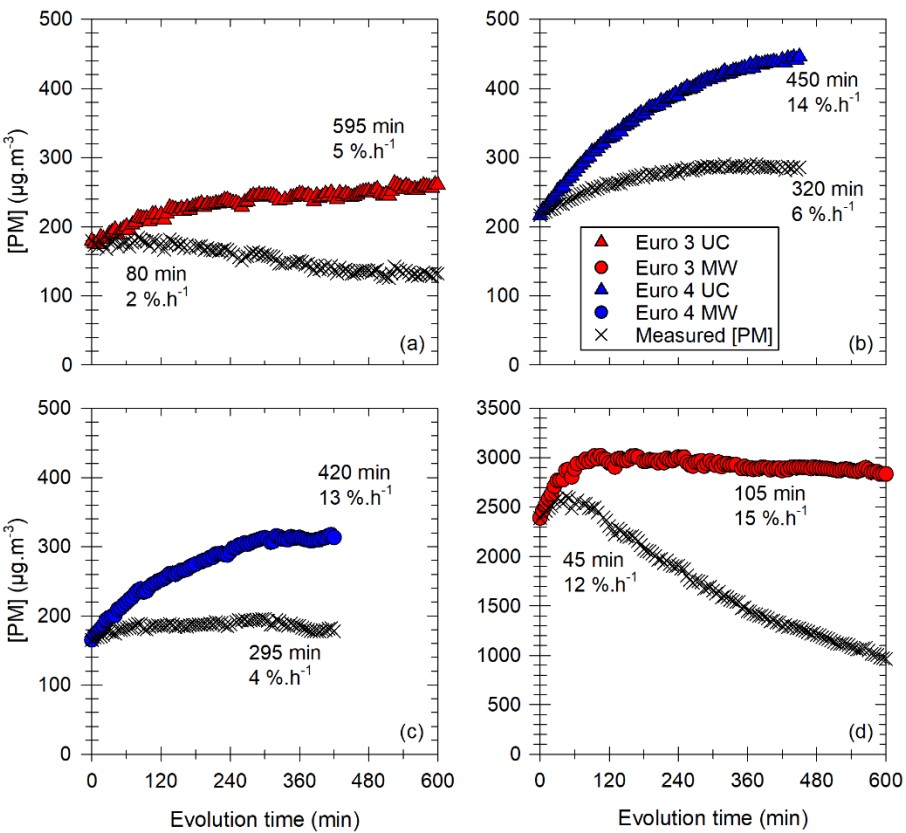

**Figure F1: Evolution of the corrected PM concentrations, for the Euro 3 vehicle (red) and Euro 4 vehicle (blue) in both UC (triangles) and MW (circles) conditions. Figures (a), (b) and (c) represent long increases, whereas Figure (d) represents a short increase. The corrected concentrations are compared to the measured (e.g. uncorrected) PM concentrations (black crosses). Labels give for each evolution (measured or corrected) the time at which [PM] reaches its maximum, as well as the associated hourly increase.**

Figure F1 shows uncorrected [PM] measurements (black crosses) and corrected [PM] for the Euro 3 (red) and Euro 4 (blue) vehicles, in UC (triangles) and MW (circles) conditions. Figure (a), (b) and (c) show "long" evolutions for which corrected [PM] reaches its maximum after several hours (> 400 min). Those figures correspond to the points on the upper left side of Figure 9a, with lower initial particle surface. Figure (d) shows a "rapid" [PM] evolution, for which the corrected concentration reaches its maximum in tan 105 minutes. It corresponds to one of the points on the lower right side of Figure 9a, with higher initial particle surface. For the "long" evolution of Figure (a), the measured concentration increases during 80 min. After that, it slightly decreases, with a decrease rate smaller than that of BC (hence the continuous increase in corrected [PM]). Figures (b) and (c) show that measured [PM] steadily increase during 295 to 320 minutes. They then remain pretty constant. Considering the fact that all of these measured concentrations (Figures (a) to (c)) are subject to leakage and wall deposition, the fact that they increase during up to 6 hours and then mostly remain constant shows that continuous increase of [PM] is possible during several hours. Therefore, the continuous increases of corrected [PM] observed on Figures 8 and 9 are not likely to be an artifact induced by the correction algorithm. These increases are obviously more important and longer on corrected concentrations, but this is normal as leakage and wall deposition, which both lower [PM], are corrected. Moreover, the "rapid" evolution of Figure (d) lasts 80 min for measured [PM], with an increase rate of 12 %.h$^{-1}$. These values indicate that there is indeed a rapid and significant increase in [PM] during the beginning of the evolution. Then, concentrations start slowly decreasing, before reaching a steady

decrease rate (similar to that of BC). This is consistent with the evolution of the corrected concentration, which

increases significantly, then increases more slightly, before it reaches a maximum and remains constant. Here again, it seems that the significant and rapid increase observed for corrected [PM] of Figure (d) is not an artifact induced by the correction method.

**Appendix G**

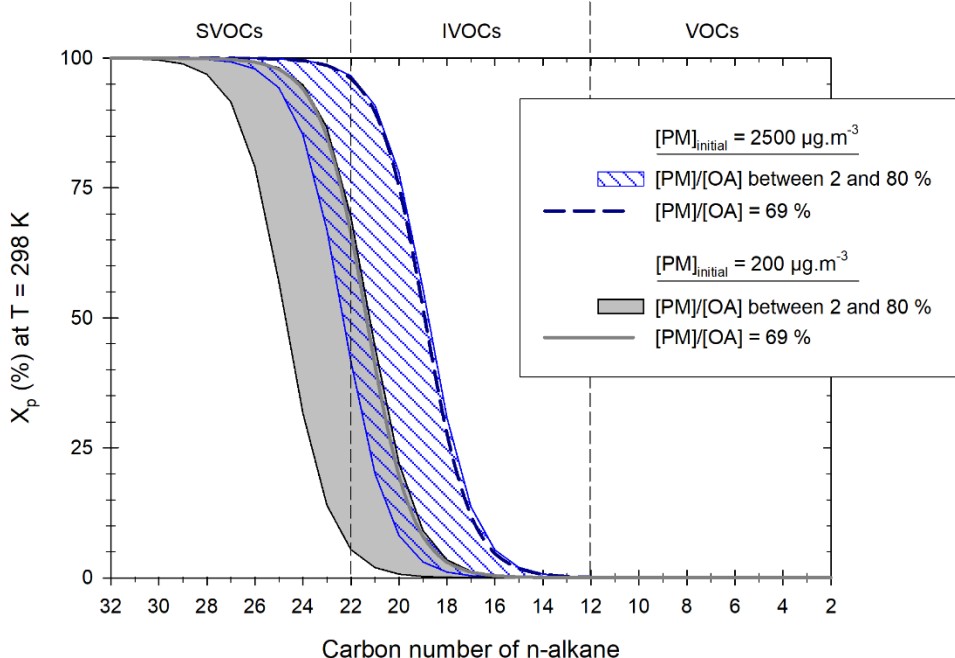

**Figure G1: Particle fraction at T = 298 K of n-alkanes (VOCs, IVOCs and SVOCs), in %, with carbon number between 2 and 32. The repartition is given assuming 2 initial PM concentrations of 200 µg.m$^{-3}$ (grey) and 2500 µg.m$^{-3}$ (blue), typical of significant [PM] increase. X$_p$ is computed with different estimates of organic aerosol concentrations [OA], ranging from 2 to 80 % of initial [PM] (grey area and blue hatched area respectively). It is also computed assuming that [OA] = 69 % of initial [PM], with initial [PM] of 200 µg.m-3 (continuous grey line) and 2500 µg.m-3 (dashed blue line).**

The effective saturation concentrations of n-alkanes up to C32 is obtained with the method described in Lu et al. (2018), for VOCs (C>12), IVOCs (C12-C22) and SVOCs (C23-C32). Using Equation (14.43) from Seinfeld and Pandis (2016), the fraction in particle phase is computed, for 2 conditions of initial [PM] typical of significant [PM] increase observed on Figure 9 (200 µg.m$^{-3}$ and 2500 µg.m$^{-3}$). The organic aerosol concentration [OA] is estimated to range roughly between 2 and 80 % of [PM]. This range is based on studies giving the ratio [PM]/[OA]

or the ratio EC/OC, which cover a wide range of values (Kostenidou et al., 2021; May et al., 2014). The wide range is caused by several parameters, such as measurement technique, experimental conditions, driving conditions, vehicle type (motorization and aftertreatment technologies). Results are given on Figure G1, as areas delimited by the lower (2 %) and upper (80 %) estimates of [PM]/[OA]. The grey area is for initial [PM] = 200 µg.m$^{-3}$ (with [OA] ranging from 4 to 160 µg.m$^{-3}$) and the hatched blue area is for initial

[PM] = 2500 µg.m$^{-3}$ (with [OA] ranging from 50 to 160 µg.m$^{-3}$). Also, in this study, the Euro 3 and Euro 4 vehicles have differences between initial concentrations of PM and BC of about 69.6 % of initial [PM]. This indicates that [OA] could represent up to 69 % of [PM]. This value is used for another estimation of [OA]. The associated fractions are shown on Figure G1 for initial [PM] = 200 µg.m$^{-3}$ (grey line, associated [OA] = 138 µg.m$^{-3}$) and for initial [PM] = 2500 µg.m$^{-3}$ (dashed blue line, associated [OA] = 1725 µg.m$^{-3}$).

For the condition with initial [PM] = 200 µg.m$^{-3}$, the percentage of n-alkanes IVOCs present in the particle phase ranges from 0.8 % to 13.7 %, for [OA] estimated as 2 % and 80 % of [PM] respectively. The percentage of n-alkanes SVOCs in the particle phase ranges from 85.5 % to 98.7 % respectively. For the estimation with [OA] = 69 % of [PM], 12.7 % of n-alkanes IVOCs and 98.5 % of n-alkanes SVOCs are in the particle phase.

For the condition with initial [PM] = 2500 µg.m$^{-3}$, the percentage of n-alkanes IVOCs present in the particle phase
ranges from 6.7 % to 34.0 %, for [OA] estimated as 2 % and 80 % of [PM] respectively. The percentage of n-alkanes SVOCs in the particle phase ranges from 96.5 % to 99.9 % respectively. For the estimation with [OA] = 69 % of [PM], 32.7 % of n-alkanes IVOCs and 99.9 % of n-alkanes SVOCs are in the particle phase.

These results show that at such high particle concentrations, significant fractions of n-alkanes IVOCs and n-alkanes SVOCs could be present in the particle phase. Overall, it indicates that IVOCs can participate in the [PM]
evolutions observed on Figure 8 and Figure 9, due to high PM concentrations and presence of IVOCs (Figure 6d) for the Euro 3 and Euro 4 vehicles.

**Appendix H**

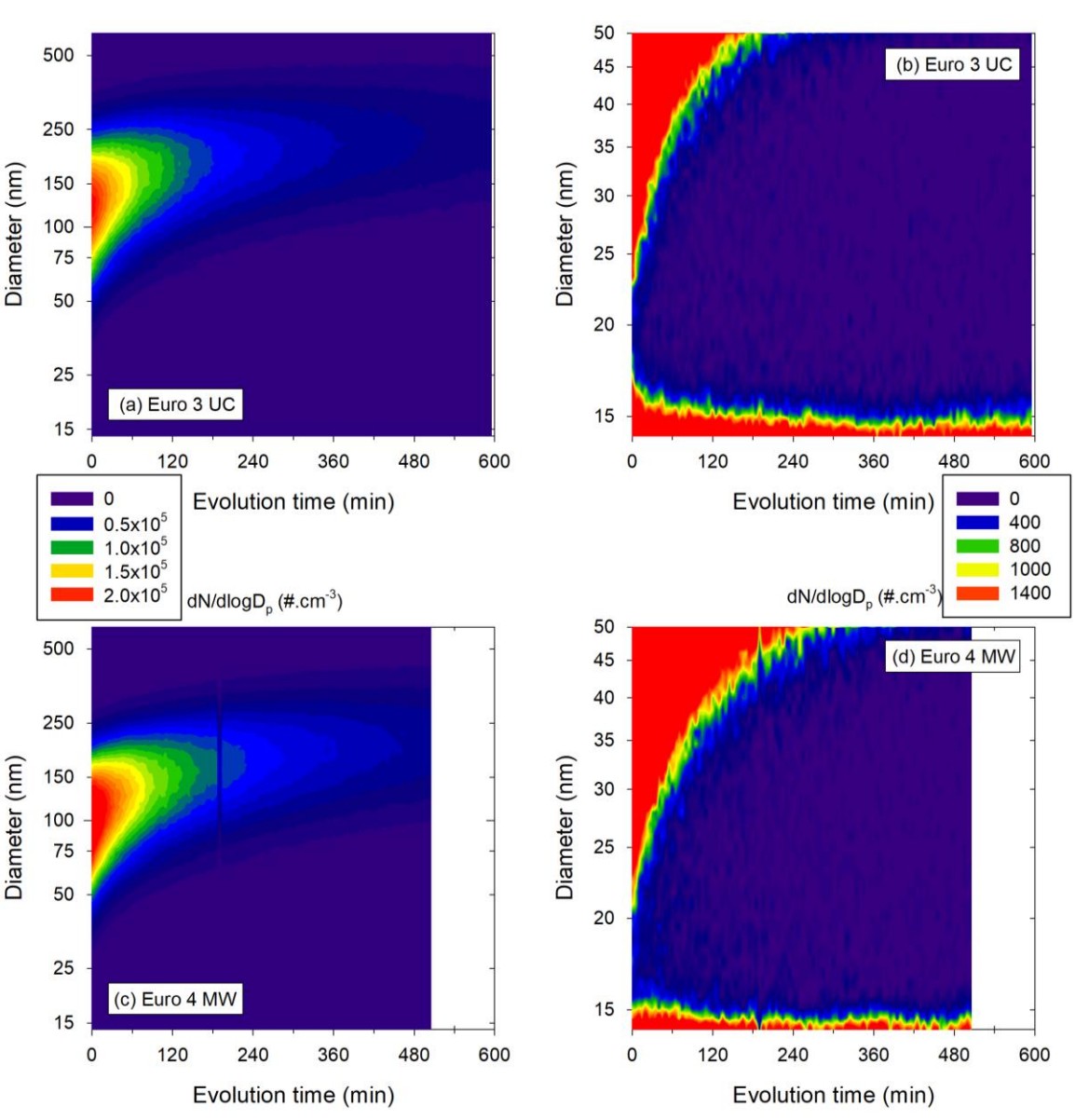

**Figure H1: Measured PN distribution evolutions for the Euro 3 UC ((a) and (b)) and the Euro 4 MW ((c) and (d)) conditions. The x-axis gives the evolution time in min. The y-axis gives the diameter in nm. The color scale gives the lognormal concentration. Figures (a) and (c) give the PN distributions for the whole SMPS measurement range. Figures (b) and (d) show the same cycles for the range 14 to 50 nm. The color scales are changed to see the smallest particles.**

Figure H1 shows measured lognormal [PN] evolutions (e.g. not corrected) for 2 cycles concerned by continuous [PM] increase during several hours, as seen on Figure 9. The cycles are from the 2 concerned vehicles: Euro 3 UC (Figures (a) and (b)) and Euro 4 MW (Figures (c) and (d)) conditions. On the left are presented the measured [PN] evolutions over the whole SMPS measurement range. On the right are presented the same data, but with a focus on smaller diameters (up to 50 nm), and with a more adapted scale. It shows that in both conditions there is the presence of small particles (below 20 nm). Their concentrations show a slight decrease over the course of the experiments, but they are present until the end. They could come from growth (due to coagulation or condensation) of nucleation mode particles. Their concentrations remaining quite constant could be because they are at the same time increasing (from growth of nucleation mode particles) and decreasing (from their own coagulation). This phenomenon could partly explain the continuous [PM] increase observed on Figure 8 and Figure 9 for the Euro 3 and Euro 4 vehicles.