# Peer review of "Evolution under dark conditions of particles from old and modern diesel vehicles, in a new environmental chamber characterized with fresh exhaust emissions"

_Atmospheric Measurement Techniques, 2021_

## Author Comment (AC1)

Dear reviewer,

Thank you for your detailed comments. Your comments are in the grey lines. Responses are given below each of them, in the white lines.

This paper discusses use of an environmental chamber to characterize particles in primary exhaust emissions, and discusses a new method to correct for particle loss on chamber walls. This method is applied to measurements of total particle number, mass, and VOC levels in various volatility ranges from representatives of various types of gasoline and diesel vehicles used in Europe, using two different driving cycles. The exhausts are injected into the chamber whose contents are monitored in the dark for several hours.

I have several concerns and questions about this study and I think more information needs to be given in the manuscript before it is suitable for publication. Since the primary objective of the paper seems to be describing the method to correct for particle wall losses, more information is needed concerning how well the data are fit by the conceptual model used, and also the magnitudes of the corrections on the reported results.

**Response:**

I do agree that information about validity and effects of correction method should be presented. We added several points in appendices, as well as some modifications in the main text, to assess your undermentioned concerns.

Regarding the magnitudes of the corrections, they were computed for each cycle, using the average ratio of corrected PM divided by measured PM. For the neutralized wall experiments, corrected PM is on average $(1.5 \pm 0.4)$ times higher than measured PM. For the charged wall experiments, corrected PM is on average $(2.8 \pm 1.5)$ times higher than measured PM. For the ammonium sulphate experiments, corrected PM is on average $(1.3 \pm 0.2)$ times higher than measured PM. Moreover, Platt et al. (2013) found particle half-life between 3.3 and 4 hours. This is equivalent to having BC decay coefficients of $3.5 \times 10^{-3}$ and $2.9 \times 10^{-3}$ $min^{-1}$ respectively. Over a 10-hour long experiment, this would give corrected PM 3.4 and 2.7 times higher than measured PM (respectively). The corrections applied in our study therefore appear to be in a reasonable range. Some discussion was added in the text to address your comment.

The method used to estimate particle loss rates is based on several assumptions that are not validated by the data that they present, or are not applicable to all experiments. It is assumed that the BC loss rate can be fit by a unimolecular decay, but it is stated that there are some experiments where the BC data are not fit by this model. This is attributed to the walls being charged in some experiments, which is a reasonable explanation. No data are shown concerning how well or poorly the BC decay are fit by a unimolecular loss curve for representative experiments, nor is there any discussion of the implications of the non-unimolecular decay in some experiments on the validity or possible biases of the corrections.

**Response:**

Regarding the lack of information on the PM correction using BC, two graphs were added in Appendix D, showing 2 evolutions of BC. The first one is well represented by a $1^{st}$ order exponential decay. It shows that the fit is in very good agreement with measurement data. The second one is not well fitted by a $1^{st}$ order exponential decay. The exponential decay clearly doesn't match the measured values, and induces for instance a 46 % error on the initial concentration. However, the simulation using a $2^{nd}$ order decay is in very good agreement with measurements. Those graphs show the importance of using in some cases a $2^{nd}$ order decay. The use of a $2^{nd}$ order decay has no effect on the $1^{st}$ step of the correction. In both cases, $[PM]_{measured}$ is multiplied by $[BC]_{t0}/[BC]_t$. The only difference is that this term can be written as $\exp(k_{BC} \times t)$ (in Eq. (2)) when the BC evolution is well fitted by a $1^{st}$ order exponential. It can however induce some bias during steps 2 and 3 (as described below).

The size correction (steps 2 and 3) are based on the assumption that the size distributions of the BC particles are the same as the other PM from the exhaust, but no information or argument is presented to support this assumption. One might think that BC is physically different from condensed low-volatility organics that form most of the other PM so it is not unreasonable to expect that their size distributions might be quite different. Finally, no figures or data are presented to show how well the "step 3" optimization worked for various types of experiments.

**Response:**

Regarding the assumption that the size distribution of the BC particles is the same as those of the other PM, we relied on several studies which also assume that aerosols are internally mixed. Even though this assumption can induce uncertainties when the loss rates are size-dependent (Wang et al., 2018), it is used in many chamber studies (Grieshop et al., 2009; Hennigan et al., 2011). Moreover, it was used in a chamber study focusing on particles from vehicle exhausts (Platt et al., 2013). We therefore considered this assumption to be reliable enough to be used in the new correction method. Some discussion about this point was added in the updated manuscript.

How close could they get the step 1 and step 2 corrected PM to agree?

**Response:**

To verify how well PM corrections from steps 1 and 2 agree, the relative difference between both corrections was computed. For cycles with BC correctly fitted by a $1^{st}$ order decay, both PM corrections agree with an average of $(97.2 \pm 2.5)$ %. For cycles with BC correctly fitted by a $2^{nd}$ order decay, both PM corrections agree with an average of $(94.7 \pm 2.6)$ %. This shows that for the cycles with a $2^{nd}$ order decay, the bias induced during step 3 in slightly higher than that of cycles with a $1^{st}$ order decay. However, in both cases, the associated error remains acceptable (5.3 % and 2.8 % for $2^{nd}$ order and $1^{st}$ order respectively). The calculation equation was added in Appendix E, and some results were added in the main text

In Section 3.1.3 they state that the magnitudes of the optimized values of the eddy diffusion coefficient, k(e), they obtained for their experiments ranged over 4 orders of magnitude from ~10^-3 to ~30 sec-1 for the different experiments. This wide variability in diffusion and mixing in experiments in the same chamber and comparable operating procedures gives me concerns about the credibility and validity of the correction method. Shouldn't the experiment with the anomalously low k(3) value of 10^-3 sec-1 have been rejected?

**Response:**

Over all the chamber experiments, we obtained $k_e$ values ranging over 4 orders of magnitude. Among those experiments, there were the ones using ammonium sulphate particles instead of exhaust particles. These are the experiments for which $k_e$ were of the order of $10^{-3}$ to $10^{-2}$ sec$^{-1}$. We agree with the reviewer that these values are particularly low. This could be attributed to the nature of the particles. These ammonium sulphate experiments have be done to be compared with the results found in literature (Nah et al., 2017). Moreover, the values of the order of magnitude $10^1$ sec$^{-1}$ were all found for experiments associated to highly electrostatically charged walls (e.g. with high wall losses). We therefore considered that is was reasonable to obtain extreme values of $k_e$ for those experiments. Over all, without considering those specific conditions (ammonium sulphate and highly charged wall experiments), the values of $k_e$ range over 2 orders of magnitude. In the simulations of particle wall deposition made by Charan et al. (2018), the values of $k_e$ which were used in the model also range over 2 orders of magnitude (from 0.015 to 8.06 sec$^{-1}$). The values we found in standard conditions are in a similar range (from 0.04 to 3.23 sec$^{-1}$). Some discussion was added in the text.

It is stated that about a third of the experiments cannot be fitted by exponential decays, and this is attributed to electrostatic charge on the walls. But is it appropriate to use Equation (3) to predict how wall loss rates depend on size under electrostatic charge conditions? I would think the loss rates would be less size dependent if it were dominated by electrostatic forces, and that maybe only using Step 1 would be more appropriate.

**Response:**

Thank you for this comment which rises a very interesting question. It seems reasonable to assume that under charged wall conditions, the electrostatic forces should dominate the other forces making the wall deposition size-independent. However, since larger particles have higher kinetic energy, they shouldn't deviate from their path to deposit onto the electrostatically charged walls as easily as smaller particles. This interpretation correlates with a mathematical analysis of the problem. Indeed, high electrostatic charges induce greater turbulence near the walls. This results in much higher difference between deposition coefficients of small and large particles, as shown in Figure 3. Therefore, it appears that under charged conditions, it is particularly important to use the size-dependent deposition coefficient of Eq. (3). This is confirmed by Nah et al. (2017), who explains that particle wall loss rates are enhanced if the chamber walls are charged.

Are data from runs with "charged walls" excluded from the averages on Figure 6? If so, this should be clearly stated. If not, different symbols or bars should be used for data obtained from such experiments, or data should be presented that there are statistically the same.

**Response:**

Thank you for underlying this aspect, as it is true that when the walls are charged, particles start depositing on them during the injection phase. It can therefore have an impact on initial particle mass, and thus on the emission factors computed using initial concentrations. The charged walls affected all cycles of the vehicle D4, as well as the MW cycle of the vehicle D2. None of the box plots of Figure 6 was covering experiments with both neutral and charged walls conditions. Therefore, the box plots representing charged walls experiments were clearly specified on Figure 6, and the legend was adapted. Moreover, comparison with particle number emission factors from Louis et al. (2016) with similar Euro 5 diesel vehicles show very similar results, thus indicating that potential wall effects are negligible. Some discussion was added in the main text to clarify this aspect.

While the loss of particles to the wall in an environmental chamber can significantly affect results of environmental chamber experiments where the objective is to study the evolution of particles over time in well-mixed air masses, it is less clear whether such elaborate corrections are needed when characterizing primary particle emissions from vehicles. Wouldn't just using the initial measurements after the chamber is well mixed, maybe with extrapolating back to time=0, be sufficient for characterizing primary emissions? Would it give similar results? There is no indication of the magnitude of the wall loss corrections in affecting the primary emissions results summarized in Figure 6.

**Response:**

The characterization of primary particle emissions is performed right after the chamber is well mixed. At that moment, no corrections for dilution and wall losses have yet been applied to the concentrations. It is directly the measured concentrations which are taken for the estimation of the emission factors. As your comment indicates, this is not clearly described in the text. The manuscript was modified so that this aspect appears clearly.

The range of values for the loss rated due to dilution (alphas) should be presented so we can compare them in magnitude with the loss rates due to wall deposition (betas), and show that the dilution rates in all the runs are in the expected range. One way to do this would be to separate "whisker" plots for alpha as part of, or on conjunction with, Figure 4. Are the dilution rates similar in the NH4SO4 experiments, or are they a factor in the lower alpha+beta values shown for those experiments in Figure 4? Is particle loss to the walls important compared to dilution in the NH4SO4 experiments?

**Response:**

The dilution rates are almost all similar (vary with a factor 3, between $11.25\times10^{-4}$ and $3.75\times10^{-4}$ min$^{-1}$), and represent only 7 to 24 % of the total loss rate (alpha+beta) for exhaust experiments. For the ammonium sulphate experiments, they represent on average 45 % of the total loss rate. I agree that it is important to explain this in the manuscript, to emphasize the importance of the wall deposition process depending on the conditions. Figure 4 was changed in the updated manuscript to address your comment, and highlight the predominance of the wall losses over leakage by separating the box plots to show vales of alpha, beta, and alpha+beta separately.

Regarding your remarks on ammonium sulphate experiments, they do have similar dilution rates as exhaust experiments. However, since wall loss rates are smaller for those experiments, the share of dilution is higher than for exhaust experiments. It represents on average 45 % of the total losses. This was added in the updated manuscript.

Figure 5 shows that, except for two gasoline exhaust runs that are very different from all the others, the kPM values from the NH4SO4 experiments are quite a bit lower than the kBC values from the exhaust experiments, and also the slope of the k vs alpha+beta line is lower. Since BC is also chemically different from exhaust particles, couldn't it also have different wall loss rates or different effects of rates on size? Were any of the NH4SO4 experiments carried out with electrostatic charged walls?

**Response:**

It is true that for ammonium sulphate particle the slope of $k_{PM}$ vs alpha+beta is a little bit lower than the slope of the fit showed on Figure 5. However, when taking the ratios of $k_{PM}$/(alpha+beta) for ammonium sulphate, the average over the 5 experiments is 1.04. This indicates that the average difference is about 4 % between both coefficients for ammonium sulphate particle experiments. This is consistent with the general trend observed on Figure 5. However, as the graphic representation doesn't really represent this result, the main text was modified to clearly make it appear.

Regarding BC, it enters in the composition of exhaust particles as one of the main species (Kostenidou et al., 2021). Therefore, when particles deposit onto the walls, BC does too. The wall loss rates can be different if BC is not homogeneously distributed in all particle sizes. However, the particle internal mixing assumption has been made in many studies, considering BC to be a good tracer for exhaust particles (Grieshop et al., 2009; Hennigan et al., 2011; Platt et al., 2013).

Finally, the ammonium sulphate experiments were all carried out when the walls were assumed to be neutralized. The goal of these experiment was to be carried out without organic material, and in conditions as similar as possible to our standard experimental conditions (e.g. without high electrostatic charge on the wall). The parameter that was changed between each ammonium sulphate experiment was the initial concentration, as it can cover a wide range of values during exhaust particle experiments.

The increase in particle mass with time during most of the experiments are explained by low-volatility gases condensing onto existing particles. Equilibrium partitioning theory predicts

that the equilibrium fraction in the particle phase increases with the total particle mass, and is not dependent on particle number. Likewise, the condensation rate would depend on particle surface area, which I think should correlate somewhat better with mass than number. Nevertheless, Figure 8b shows a plot of data related to particle mass increases against particle number, not particle mass or surface area. Is the correlation not as good if plots are against particle mass or surface area instead? If this is the case, it should be pointed out and attempts to explain this should be offered (though I can't think of any explanation if this indeed were the case.) If number, mass and area are highly correlated then the plots would look the same, but in that case plots against particle mass would be more appropriate since it corresponds more directly to the explanation you are giving and existing theories.

**Response:**

Thank you for this comment on the fact that plotting PM increase versus initial mass PM or initial surface PS would be more adequate than versus initial PN. We decided to plot the PM increase against initial particle surface PS instead, since it makes more sense with regards to the theory (as explained in your comment). The general trend is similar to that obtained when plotted against initial PN. The best fit obtained here is logarithmic. It reflects the fact that PM increase is limited, and will at some point stop increasing even though initial PS increases. We interpreted it as the fact that above a certain threshold ($\sim 10^4 \, \mu m^2.cm^{-3}$), PM increase is limited by certain factors. A limiting factor could be the concentration of available organic material. This result is interesting, and is in good agreement with other studies. Namely, Charan et al. (2020) found that above $\sim 1800 \, \mu m^2.cm^{-3}$, initial seed surface area becomes insignificant in terms of SOA yield, partly due to initial precursor concentrations. The different value of the threshold between this study and ours could be explained by the differences in chamber size, experimental conditions, particle nature, and composition of the gas phase. Figure 8b was changed to have initial particle surface PS on the x-axis, and this discussion was added in the main text and in the conclusion.

In conclusion, I think the paper needs to give more data and information about the validity and performance of the correction method, and the effects of these uncertainties on the corrections to the data that they present, before it is accepted for publication.

**References:**

Charan, S.M., Buenconsejo, R.S., Seinfeld, J.H., 2020. Secondary Organic Aerosol Yields from the Oxidation of Benzyl Alcohol (preprint). Aerosols/Laboratory Studies/Troposphere/Chemistry (chemical composition and reactions). https://doi.org/10.5194/acp-2020-492

Charan, S.M., Kong, W., Flagan, R.C., Seinfeld, J.H., 2018. Effect of particle charge on aerosol dynamics in Teflon environmental chambers. Aerosol Sci. Technol. 52, 854–871. https://doi.org/10.1080/02786826.2018.1474167

Grieshop, A.P., Logue, J.M., Donahue, N.M., Robinson, A.L., 2009. Laboratory investigation of photochemical oxidation of organic aerosol from wood fires 1: measurement and simulation of organic aerosol evolution. Atmospheric Chem. Phys. 9, 1263–1277. https://doi.org/10.5194/acp-9-1263-2009

Hennigan, C.J., Miracolo, M.A., Engelhart, G.J., May, A.A., Presto, A.A., Lee, T., Sullivan, A.P., McMeeking, G.R., Coe, H., Wold, C.E., Hao, W.-M., Gilman, J.B., Kuster, W.C., de Gouw, J., Schichtel, B.A., Collett, J.L., Kreidenweis, S.M., Robinson, A.L., 2011. Chemical and physical transformations of organic aerosol from the photo-oxidation of open biomass burning emissions in an environmental chamber. Atmospheric Chem. Phys. 11, 7669–7686. https://doi.org/10.5194/acp-11-7669-2011

Kostenidou, E., Martinez-Valiente, A., R'Mili, B., Marques, B., Temime-Roussel, B., Durand, A., André, M., Liu, Y., Louis, C., Vansevenant, B., Ferry, D., Laffon, C., Parent, P., D'Anna, B., 2021. Technical note: Emission factors, chemical composition, and morphology of particles emitted from Euro 5 diesel and gasoline light-duty vehicles during transient cycles. Atmospheric Chem. Phys. 21, 4779–4796. https://doi.org/10.5194/acp-21-4779-2021

Louis, C., Liu, Y., Tassel, P., Perret, P., Chaumond, A., André, M., 2016. PAH, BTEX, carbonyl compound, black-carbon, NO 2 and ultrafine particle dynamometer bench emissions for Euro 4 and Euro 5 diesel and gasoline passenger cars. Atmos. Environ. 141, 80–95. https://doi.org/10.1016/j.atmosenv.2016.06.055

Nah, T., McVay, R.C., Pierce, J.R., Seinfeld, J.H., Ng, N.L., 2017. Constraining uncertainties in particle-wall deposition correction during SOA formation in chamber experiments. Atmospheric Chem. Phys. 17, 2297–2310. https://doi.org/10.5194/acp-17-2297-2017

Platt, S.M., El Haddad, I., Zardini, A.A., Clairotte, M., Astorga, C., Wolf, R., Slowik, J.G., Temime-Roussel, B., Marchand, N., Ježek, I., Drinovec, L., Močnik, G., Möhler, O., Richter, R., Barmet, P., Bianchi, F., Baltensperger, U., Prévôt, A.S.H., 2013. Secondary organic aerosol formation from gasoline vehicle emissions in a new mobile environmental reaction chamber. Atmospheric Chem. Phys. 13, 9141–9158. https://doi.org/10.5194/acp-13-9141-2013

Wang, N., Jorga, S., Pierce, J., Donahue, N., Pandis, S., 2018. Particle Wall-loss Correction Methods in Smog Chamber Experiments. Atmospheric Meas. Tech. Discuss. 1–29. https://doi.org/10.5194/amt-2018-175

---

## Author Comment (AC2)

Dear reviewer,

Thank you for your detailed comments. Your comments are in the grey lines. Responses are given below each of them, in the white lines.

The authors investigate the emissions of 6 diesel passenger cars from Euro 3 to Euro 6 under 2 different driving cycles. They determine emission factors of particle number, particle mass, black carbon, NMHC and IVOC. They present a method to correct the evolution of particle mass concentration for dilution and wall loss of particles in the chamber. They claim that under dark conditions particle mass concentration (PM) increases over time as a function of initial particle number or particle surface concentration. Below a particle number concentration of (8-9)E4 cm-3 the particle number concentration (PN) increases, while above it decreases.

The experiments and the data analysis are well described. However, there is much speculation regarding the interpretation of the data and a lack of proof of their claims.

The authors interpret their observation of a sustained increase of particle mass and number concentrations by condensation and nucleation, respectively. However, to have continuous condensation or nucleation a constant production of condensable or nucleating vapors need to occur. Otherwise, vapors will rapidly condense on particles and on the wall and condensation or nucleation stops. In their small chamber the lifetime of such vapors will be less than ten minutes. Therefore, the claim that PM increases over many hours could be due to condensation of IVOCs is not plausible. The authors may also check if the saturation vapor pressure of IVOCs is low enough to partition to the particle phase at the particle mass concentrations of their experiments.

**Response:**

The partition of IVOCs and SVOCs has been studied to check the potential role of IVOCs in particle evolution. Briefly, the effective saturation concentrations of n-alkanes up to C32 is obtained with the method described in Lu et al. (2018), for VOCs (C>12), IVOCs (C12-C22) and SVOCs (C23-C32). Using Equation (14.43) from Seinfeld and Pandis (2016), the fraction in particle phase is computed, for 2 conditions of initial [PM] typical of significant [PM] increase observed on Figure 9 (200 $\mu$g.m$^{-3}$ and 2500 $\mu$g.m$^{-3}$). The organic aerosol concentration [OA] is estimated to range roughly between 2 and 80 % of [PM], based on studies giving the ratio [PM]/[OA] or the ratio EC/OC (Kostenidou et al., 2021; May et al., 2014). The wide range is caused by several parameters, such as measurement technique, experimental conditions, driving conditions, vehicle type (motorization and aftertreatment technologies). In this study, the [OA] fraction of [PM] is estimated to be in the range 2-80 %. This is based on the difference between [PM] and [BC], which indicates that the share of organic aerosol could go up to 69 % for the Euro 3 and Euro 4 vehicles.

For the condition with initial [PM] = 200 $\mu$g.m$^{-3}$, the percentage of n-alkanes IVOCs present in the particle phase ranges from 0.8 % to 13.7 %, for [OA] estimated as 2 % and 80 % of [PM] respectively. The percentage of n-alkanes SVOCs in the particle phase ranges from 85.5 % to 98.7 % respectively. For the estimation with [OA] = 69 % of [PM], 12.7 % of n-alkanes IVOCs and 98.5 % of n-alkanes SVOCs are in the particle phase.

For the condition with initial [PM] = 2500 µg.m$^{-3}$, the percentage of n-alkanes IVOCs present in the particle phase ranges from 6.7 % to 34.0 %, for [OA] estimated as 2 % and 80 % of [PM] respectively. The percentage of n-alkanes SVOCs in the particle phase ranges from 96.5 % to 99.9 % respectively. For the estimation with [OA] = 69 % of [PM], 32.7 % of n-alkanes IVOCs and 99.9 % of n-alkanes SVOCs are in the particle phase.

These results indicate that at such high particle concentrations, significant fractions of IVOCs and SVOCs could be present in the particle phase. It indicates that IVOCs can participate in the [PM] evolutions observed on Figure 8 and Figure 9, due to high PM concentrations and presence of IVOCs (Figure 6d) for the Euro 3 and Euro 4 vehicles.

This part was added and discussed in Appendix G.

I agree that the growth over several hours cannot be explained solely by the presence of organic material, which would deposit onto the walls or partition in the particle phase. Other interpretations must be discussed. One of them is the possibility that the walls would play the role of a source of organic material. This hypothesis relies on several studies, which showed that vapor wall deposition was a reversible process (Matsunaga and Ziemann ‡, 2010; Yeh and Ziemann, 2015; Zhang et al., 2015). Walls could therefore play a role of sink as well as a role of source of gas-phase pollutants during chamber experiments (Kaltsonoudis et al., 2019). Zhang et al. (2015) observed evaporation of organics from the walls when the temperature went from 25 to 45 °C. Even though the temperature increase in our experiments is not as high (5 °C on average, due to heat created by instrumentation), the role of walls as a source of organic material might be part of the explanation of PM increase. This is only a hypothesis, and it is not likely to explain alone the PM increase. It might still play a role in the observed evolutions by inducing continuous condensation. Some discussion was added in the updated manuscript.

There are also some issues regarding the PN increase and the particle number concentration threshold. This observation depends on the available instrumentation. The smallest particles they can measure is 14 nm. They do not present the particle size distributions of their different experiments and therefore it is not clear in which cases a nucleation mode is formed. This would happen very fast during injection and there is not a steady nucleation going on as explained above. If an increase of PN occurs, this happens because of coagulation of nucleation mode particles, which produces particles of larger size that become measurable in their SMPS. This is not an increase of PN but the fact, that they did not measure the particles below 14 nm. Thus, the threshold and the time of increase depend on this measurement parameter. The fact, that a growing nucleation mode influences the PN at larger sizes is not a new finding. One should always be aware of it when analyzing such data.

**Response:**

I agree that continuous nucleation is not a possible explanation for continuous PN increase. Coagulation of nucleation mode particles, initially undetected by the SMPS, is a reasonable explanation. Nucleation mode particles are most likely to be present and have significant impacts when initial concentrations are low (below [8-9]×10$^4$ #.cm$^{-3}$). Above that concentration, they might still be present but without observable impacts on PN evolution. Some discussion was added with Figure 10 to discuss the cases for which nucleation mode particles are likely to be present.

Due to this, the interpretation of Figure 11 was thoroughly modified, and the figure was adapted. The PN increase is now clearly presented as an artifact due to the SMPS measurement range. It is not a physical creation of particles, but growth of nucleation mode particles. This enables to discuss the presence of a nucleation mode even though it is not measurable with the current experimental set up. Above the value of $[8-9] \times 10^4$ #.cm$^{-3}$, coagulation of nucleation mode particles might still occur, but with negligible impacts on PN evolution. Coagulation of the particles in the SMPS range is in those conditions the predominant process explaining the decrease of PN. Therefore, the chamber is well adapted to observation and quantification of the process of coagulation for initial concentrations above this value. As your comment says, the threshold and the time of increase don't represent indicators of physical processes. The time of increase was removed since it has no physical interpretation and has no use for the characterization of the chamber. The threshold value is now described as the value below which nucleation mode particles are likely to have significant effects on particle number evolution. It is also given as the value above which the chamber and experimental set up are suitable for quantification and observation of the coagulation process.

We decided to show these results in the text with the modified Figure 11, because one of the main goals of this paper is also to characterize the chamber. Therefore, detailing the most suitable conditions for the study of a physical process (coagulation) appears to be relevant.

At high PN concentrations when "coagulation prevails" the Figures 11c,d,e show first the expected decrease of PN due to coagulation, which is however after 1-2-h followed by an increase of PN. Do the authors have an explanation for this observation? Are these wall loss corrected PN data? If yes, this would mean the newly developed algorithm could eventually not correctly compensate for the losses.

**Response:**

The slight increase in the second half of the evolutions could in part be explained by the growth of nucleation mode particles. As discussed on Figure 10c and d, nucleation mode particles could be present in those conditions (high initial PN). It would be relatively small compared to total PN, and would grow slowly by coagulation and/or condensation. This slow growth could explain the slight PN increase observed after a few hours. Also, as time increases, the correction becomes more important, with higher error bars. Taking the error bars into account makes the increase less significant.

The authors need to provide more evidence why PM should grow over hours. Condensable vapors are lost to the walls and it is not plausible how an oversaturation is maintained over hours without production. Thus, the wall-loss correction method could induce such an artifact.

**Response:**

Regarding the possible artifact induced by the correction method, we made some verifications to make sure that the observed PM growth was a real phenomenon. We specifically looked at the cycles with the highest increase rates (3 last points on Figure 8, from Euro 3 MW conditions) and the longest increase durations (points on the upper left corner of Figure 9a, from Euro 3 UC and Euro 4 UC and MW conditions). We performed the verifications using the measured (e.g. uncorrected) PM concentrations. It is presented on Appendix F of the updated manuscript. Briefly, for the "rapid" PM increase experiments, measured data also show high PM increase rates. The increase durations are smaller than for corrected PM. This difference is normal considering the fact that leakage and wall deposition would at some point overtake PM growth even though the latter is still occurring. This corresponds to a slight decrease of uncorrected PM, and an increase of corrected PM. It hereby extends the PM increase duration obtained for corrected concentrations. Moreover, for the "long" PM increase experiments, measured PM concentrations show increase durations going up to 355 minutes (Euro 4 MW conditions). The shorter increases are followed by slight PM decreases, with decrease rates smaller than that of BC (by a factor 2.2 on average), until the end of the experiment. This indicates that processes leading to PM increase are present at all times of the experiments. Moreover, for the longer increases, the increase phases are followed by extremely small decreases or constant concentrations. This indicates that PM increasing processes are present during the whole time of the experiment. Overall, the PM increase rates and durations observed for corrected concentrations appear to be due to actual physical processes, and not artifacts induced by the correction method. The figures and discussion have been added in Appendix F of the updated manuscript and in the main text, before Figure 8, to clarify this aspect in section 3.3

More interpretation is needed to explain the observed PM increase. The walls being a source of organic material is not likely to be the main reason. Another possible explanation is the growth of small particles, undetected by the SMPS (as seen for the PN evolutions). This is discussed with the measured PN distribution of 2 cycles with long increases (Euro 3 UC and Euro 4 MW). They show that during the whole experiments, small particles (below 20 nm) are present in significant and almost constant concentrations. They could come from the growth of nucleation mode particles (by coagulation or condensation), undetected by the SMPS before their growth. The reason why the concentration of those particles remains quite steady could be that they increase (by growth of nucleation mode particles) and decrease at the same time (by their own coagulation). This would have the effect of bringing more material inside the detection range of the SMPS, therefore increasing total particle mass during the whole course of the experiments. This explanation would mean that the observed PM increase is partially an artifact (due to SMPS measurement range), and that it might induce an overestimation. Some discussion about this was added in the main text.

Overall, the continuous PM increase observed for the Euro 3 and Euro 4 vehicles can be explained by several complementary phenomena (growth of initially undetected particles, condensation of emitted organics, condensation of organics released by the walls). Three figures and some discussions were added in the updated manuscript (Appendix F, Appendix H, main text) in order to more accurately describe the observed PM evolutions.

Although Figure 5 shows a good correlation between the mean loss coefficients and $k_{BC}$ it is also obvious, but not mentioned, that there is a large off-set. Why does this occur and how

does this affect the PM correction? The authors need to show that the PM growth is a real phenomenon and not an artifact from their correction method.

**Response:**

The off-set of Figure 5 represents the fact that for some cycles the average loss coefficient obtained with the optimized value of $k_e$, is slightly higher than $k_{BC}$. It reflects the fact that small particles have a higher loss coefficient than the larger ones, and that it impacts the average loss coefficient $\alpha+\beta_{ke}^{mean}$. In spite of that, the optimized value of $k_e$ that was taken is that for which corrected PM from steps 1 and 2 match the better. Another value of $k_e$ could have be chosen, in order to have $\alpha+\beta_{ke}^{mean}$ coefficients and $k_{BC}$ coefficients as close as possible. Some tests were conducted on 3 experiments to establish the impact of matching the loss coefficients $\alpha+\beta_{ke}^{mean}$ and $k_{BC}$ instead of matching the corrected PM from steps 1 and 2. The tests were conducted on the 2 experiments for which the coefficients $\alpha+\beta_{ke}^{mean}$ and $k_{BC}$ have the worse match (the 2 gasoline on the lower left corner of Figure 5). It results in a PM difference of less than 10 % on average (8.9 and 9.7 % for each test). A test was also conducted on a cycle with higher loss coefficients (the gasoline experiment with the highest value of $\alpha+\beta_{ke}^{mean}$). For this cycle, the impact on corrected PM is 9.8 %. These impacts are not negligible, but don't affect the trends observed in section 3.2. Also, due to the size-dependence of the wall losses, the match of corrected PM from steps 1 and 2 seems to be a better indicator than the match of average loss coefficients $\alpha+\beta_{ke}^{mean}$ and $k_{BC}$. Some discussion about this was added in the main text, below Figure 5.

By calculating corrected PM do they use the average loss rate or the size dependent loss rates?

**Response:**

Corrected PM is obtained using the size-dependent loss rate. The "average loss rate" $\alpha+\beta_{ke}^{mean}$ is never actually applied. It is computed using the size-dependent loss coefficients of each diameter range, which are applied to each particle size bin.

Line 27: "Condensation is 4 times faster when the available particle surface if multiplied by 3". How did the authors calculate this? Is this for a certain particle size?

**Response:**

This result is obtained using Figure 9a. The group with high initial particle surface has an average initial concentration of $1.57\times10^5$ $\mu m^2.cm^{-3}$, with an average time to reach maximum PM of 12 minutes. The other group has an average initial concentration of $2.40\times10^4$ $\mu m^2.cm^{-3}$, with an average time to reach maximum PM of 482 minutes. This means that there is a ratio 6.5 between initial surfaces of both groups (instead of 3 as written in the preprint), and a ratio ¼ in the time needed to reach maximum PM. The value was modified, and the explanation of the computation was added in the main text of the updated manuscript.

Overall, I do not see much scientific progress in this paper.

**References**

Kaltsonoudis, C., Jorga, S.D., Louvaris, E., Florou, K., Pandis, S.N., 2019. A portable dual-smog-chamber system for atmospheric aerosol field studies. Atmospheric Meas. Tech. 12, 2733–2743. https://doi.org/10.5194/amt-12-2733-2019

Kostenidou, E., Martinez-Valiente, A., R'Mili, B., Marques, B., Temime-Roussel, B., Durand, A., André, M., Liu, Y., Louis, C., Vansevenant, B., Ferry, D., Laffon, C., Parent, P., D'Anna, B., 2021. Technical note: Emission factors, chemical composition, and morphology of particles emitted from Euro 5 diesel and gasoline light-duty vehicles during transient cycles. Atmospheric Chem. Phys. 21, 4779–4796. https://doi.org/10.5194/acp-21-4779-2021

Lu, Q., Zhao, Y., Robinson, A.L., 2018. Comprehensive organic emission profiles for gasoline, diesel, and gas-turbine engines including intermediate and semi-volatile organic compound emissions. Atmospheric Chem. Phys. 18, 17637–17654. https://doi.org/10.5194/acp-18-17637-2018

Matsunaga, A., Ziemann ‡, P.J., 2010. Gas-Wall Partitioning of Organic Compounds in a Teflon Film Chamber and Potential Effects on Reaction Product and Aerosol Yield Measurements. Aerosol Sci. Technol. 44, 881–892. https://doi.org/10.1080/02786826.2010.501044

May, A.A., Nguyen, N.T., Presto, A.A., Gordon, T.D., Lipsky, E.M., Karve, M., Gutierrez, A., Robertson, W.H., Zhang, M., Brandow, C., Chang, O., Chen, S., Cicero-Fernandez, P., Dinkins, L., Fuentes, M., Huang, S.-M., Ling, R., Long, J., Maddox, C., Massetti, J., McCauley, E., Miguel, A., Na, K., Ong, R., Pang, Y., Rieger, P., Sax, T., Truong, T., Vo, T., Chattopadhyay, S., Maldonado, H., Maricq, M.M., Robinson, A.L., 2014. Gas- and particle-phase primary emissions from in-use, on-road gasoline and diesel vehicles. Atmos. Environ. 88, 247–260. https://doi.org/10.1016/j.atmosenv.2014.01.046

Seinfeld, J.H., Pandis, S.N., 2016. Atmospheric chemistry and physics: from air pollution to climate change.

Yeh, G.K., Ziemann, P.J., 2015. Gas-Wall Partitioning of Oxygenated Organic Compounds: Measurements, Structure–Activity Relationships, and Correlation with Gas Chromatographic Retention Factor. Aerosol Sci. Technol. 49, 727–738. https://doi.org/10.1080/02786826.2015.1068427

Zhang, X., Schwantes, R.H., McVay, R.C., Lignell, H., Coggon, M.M., Flagan, R.C., Seinfeld, J.H., 2015. Vapor wall deposition in Teflon chambers. Atmospheric Chem. Phys. 15, 4197–4214. https://doi.org/10.5194/acp-15-4197-2015

---

## Author Response (AR1)

Author's response (amt-2021-43)

**Anonymous Referee #1** (https://doi.org/10.5194/amt-2021-43-RC1, 2021)

Dear reviewer,

Thank you for your detailed comments. Your comments are in the grey lines (1). Responses are given below each of them, in the white lines (2). They are followed by the changes which have been made in the manuscript (3).

**(1) Comment from referee**

This paper discusses use of an environmental chamber to characterize particles in primary exhaust emissions, and discusses a new method to correct for particle loss on chamber walls. This method is applied to measurements of total particle number, mass, and VOC levels in various volatility ranges from representatives of various types of gasoline and diesel vehicles used in Europe, using two different driving cycles. The exhausts are injected into the chamber whose contents are monitored in the dark for several hours.

**(1) Comment from referee**

I have several concerns and questions about this study and I think more information needs to be given in the manuscript before it is suitable for publication. Since the primary objective of the paper seems to be describing the method to correct for particle wall losses, more information is needed concerning how well the data are fit by the conceptual model used, and also the magnitudes of the corrections on the reported results.

**(2) Author's response**

I do agree that information about validity and effects of correction method should be presented. We added several points in appendices, as well as some modifications in the main text, to assess your undermentioned concerns.

Regarding the magnitudes of the corrections, they were computed for each cycle, using the average ratio of corrected PM divided by measured PM. For the neutralized wall experiments, corrected PM is on average $(1.5 \pm 0.4)$ times higher than measured PM. For the charged wall experiments, corrected PM is on average $(2.8 \pm 1.5)$ times higher than measured PM. For the ammonium sulphate experiments, corrected PM is on average $(1.3 \pm 0.2)$ times higher than measured PM. Moreover, Platt et al. (2013) found particle half-life between 3.3 and 4 hours. This is equivalent to having BC decay coefficients of $3.5 \times 10^{-3}$ and $2.9 \times 10^{-3}$ min$^{-1}$ respectively. Over a 10-hour long experiment, this would give corrected PM 3.4 and 2.7 times higher than measured PM (respectively). The corrections applied in our study therefore

appear to be in a reasonable range. Some discussion was added in the text to address your comment.

**(3) Changes in the manuscript**

| | | |
|---|---|---|
| The discussion on the correction magnitude has been added in the text. | P17 | L475-483 |

**(1) Comment from referee**

The method used to estimate particle loss rates is based on several assumptions that are not validated by the data that they present, or are not applicable to all experiments. It is assumed that the BC loss rate can be fit by a unimolecular decay, but it is stated that there are some experiments where the BC data are not fit by this model. This is attributed to the walls being charged in some experiments, which is a reasonable explanation. No data are shown concerning how well or poorly the BC decay are fit by a unimolecular loss curve for representative experiments, nor is there any discussion of the implications of the non-unimolecular decay in some experiments on the validity or possible biases of the corrections.

**(2) Author's response**

Regarding the lack of information on the PM correction using BC, two graphs were added in Appendix D, showing 2 evolutions of BC. The first one is well represented by a $1^{st}$ order exponential decay. It shows that the fit is in very good agreement with measurement data. The second one is not well fitted by a $1^{st}$ order exponential decay. The exponential decay clearly doesn't match the measured values, and induces for instance a 46 % error on the initial concentration. However, the simulation using a $2^{nd}$ order decay is in very good agreement with measurements. Those graphs show the importance of using in some cases a $2^{nd}$ order decay. The use of a $2^{nd}$ order decay has no effect on the $1^{st}$ step of the correction. In both cases, $[PM]_{measured}$ is multiplied by $[BC]_{t0}/[BC]_t$. The only difference is that this term can be written as $\exp(k_{BC} \times t)$ (in Eq. (2)) when the BC evolution is well fitted by a $1^{st}$ order exponential. It can however induce some bias during steps 2 and 3 (as described below).

**(3) Changes in the manuscript**

| | | |
|---|---|---|
| The discussion has been added in the text. | P9 | L275-277 |
| Appendix D has been added to discuss how well exponential decays fit the BC evolutions. | P40-41 | L1132-1141 |

**(1) Comment from referee**

The size correction (steps 2 and 3) are based on the assumption that the size distributions of the BC particles are the same as the other PM from the exhaust, but no information or argument is presented to support this assumption. One might think that BC is physically different from condensed low-volatility organics that form most of the other PM so it is not unreasonable to expect that their size distributions might be quite different. Finally, no figures or data are presented to show how well the "step 3" optimization worked for various types of experiments.

**(2) Author's response**

Regarding the assumption that the size distribution of the BC particles is the same as those of the other PM, we relied on several studies which also assume that aerosols are internally mixed. Even though this assumption can induce uncertainties when the loss rates are size-dependent (Wang et al., 2018), it is used in many chamber studies (Grieshop et al., 2009; Hennigan et al., 2011). Moreover, it was used in a chamber study focusing on particles from vehicle exhausts (Platt et al., 2013). We therefore considered this assumption to be reliable enough to be used in the new correction method. Some discussion about this point was added in the updated manuscript.

**(3) Changes in the manuscript**

| | | |
|---|---|---|
| The internal mixing assumption discussion has been added in the text. | P9 | L269-272 |

**(1) Comment from referee**

How close could they get the step 1 and step 2 corrected PM to agree?

**(2) Author's response**

To verify how well PM corrections from steps 1 and 2 agree, the relative difference between both corrections was computed. For cycles with BC correctly fitted by a $1^{st}$ order decay, both PM corrections agree with an average of $(97.2 \pm 2.5)$ %. For cycles with BC correctly fitted by a $2^{nd}$ order decay, both PM corrections agree with an average of $(94.7 \pm 2.6)$ %. This shows that for the cycles with a $2^{nd}$ order decay, the bias induced during step 3 in slightly higher than that of cycles with a $1^{st}$ order decay. However, in both cases, the associated error remains acceptable (5.3 % and 2.8 % for $2^{nd}$ order and $1^{st}$ order respectively). The calculation equation was added in Appendix E, and some results were added in the main text

**(3) Changes in the manuscript**

| | | |
|---|---|---|
| The discussion has been added in the text. | P15 | L413-418 |
| Appendix E has been added to discuss corrections between steps 1 and 2. | P41 | L1142-1144 |

**(1) Comment from referee**

In Section 3.1.3 they state that the magnitudes of the optimized values of the eddy diffusion coefficient, k(e), they obtained for their experiments ranged over 4 orders of magnitude from ~$10^{-3}$ to ~30 sec-1 for the different experiments. This wide variability in diffusion and mixing in experiments in the same chamber and comparable operating procedures gives me concerns about the credibility and validity of the correction method. Shouldn't the experiment with the anomalously low k(3) value of $10^{-3}$ sec-1 have been rejected?

**(2) Author's response**

Over all the chamber experiments, we obtained $k_e$ values ranging over 4 orders of magnitude. Among those experiments, there were the ones using ammonium sulphate particles instead of exhaust particles. These are the experiments for which $k_e$ were of the order of $10^{-3}$ to $10^{-2}$ sec$^{-1}$. We agree with the reviewer that these values are particularly low. This could be attributed to the nature of the particles. These ammonium sulphate experiments have be done to be compared with the results found in literature (Nah et al., 2017). Moreover, the values of the order of magnitude $10^1$ sec$^{-1}$ were all found for experiments associated to highly electrostatically charged walls (e.g. with high wall losses). We therefore considered that is was reasonable to obtain extreme values of $k_e$ for those experiments. Over all, without considering those specific conditions (ammonium sulphate and highly charged wall experiments), the values of $k_e$ range over 2 orders of magnitude. In the simulations of particle wall deposition made by Charan et al. (2018), the values of $k_e$ which were used in the model also range over 2 orders of magnitude (from 0.015 to 8.06 sec$^{-1}$). The values we found in standard conditions are in a similar range (from 0.04 to 3.23 sec$^{-1}$). Some discussion was added in the text.

**(3) Changes in the manuscript**

| | | |
|---|---|---|
| The discussion on the orders of magnitude of $k_e$ has been added in the text. | P15 | L435-443 |

**(1) Comment from referee**

It is stated that about a third of the experiments cannot be fitted by exponential decays, and this is attributed to electrostatic charge on the walls. But is it appropriate to use Equation (3) to predict how wall loss rates depend on size under electrostatic charge conditions? I would think the loss rates would be less size dependent if it were dominated by electrostatic forces, and that maybe only using Step 1 would be more appropriate.

**(2) Author's response**

Thank you for this comment which rises a very interesting question. It seems reasonable to assume that under charged wall conditions, the electrostatic forces should dominate the other forces making the wall deposition size-independent. However, since larger particles have higher kinetic energy, they shouldn't deviate from their path to deposit onto the electrostatically charged walls as easily as smaller particles. This interpretation correlates with a mathematical analysis of the problem. Indeed, high electrostatic charges induce greater turbulence near the walls. This results in much higher difference between deposition coefficients of small and large particles, as shown in Figure 3. Therefore, it appears that under charged conditions, it is particularly important to use the size-dependent deposition coefficient of Eq. (3). This is confirmed by Nah et al. (2017), who explains that particle wall loss rates are enhanced if the chamber walls are charged.

**(3) Changes in the manuscript**

| | | |
|---|---|---|
| A sentence has been added in the text. | P15 | L412-413 |

**(1) Comment from referee**

Are data from runs with "charged walls" excluded from the averages on Figure 6? If so, this should be clearly stated. If not, different symbols or bars should be used for data obtained from such experiments, or data should be presented that there are statistically the same.

**(2) Author's response**

Thank you for underlying this aspect, as it is true that when the walls are charged, particles start depositing on them during the injection phase. It can therefore have an impact on initial particle mass, and thus on the emission factors computed using initial concentrations. The charged walls affected all cycles of the vehicle D4, as well as the MW cycle of the vehicle D2. None of the box plots of Figure 6 was covering experiments with both neutral and charged walls conditions. Therefore, the box plots representing charged walls experiments were clearly specified on Figure 6, and the legend was adapted. Moreover, comparison with particle number emission factors from Louis et al. (2016) with similar Euro 5 diesel vehicles show very similar results, thus indicating that potential wall effects are negligible. Some discussion was added in the main text to clarify this aspect.

**(3) Changes in the manuscript**

| | | |
|---|---|---|
| A discussion about potential effects of wall charge on particle emission factors has been added in the text. | P19 | L516-522 |
| Figure 6 has been modified to clearly show which cycles are concerned by highly charged walls. | P20-21 | L530 |
| The description of Figure 6 has been adapted to the modifications. | P21 | L539-541 |

**(1) Comment from referee**

While the loss of particles to the wall in an environmental chamber can significantly affect results of environmental chamber experiments where the objective is to study the evolution of particles over time in well-mixed air masses, it is less clear whether such elaborate corrections are needed when characterizing primary particle emissions from vehicles. Wouldn't just using the initial measurements after the chamber is well mixed, maybe with extrapolating back to time=0, be sufficient for characterizing primary emissions? Would it give similar results? There is no indication of the magnitude of the wall loss corrections in affecting the primary emissions results summarized in Figure 6.

**(2) Author's response**

The characterization of primary particle emissions is performed right after the chamber is well mixed. At that moment, no corrections for dilution and wall losses have yet been applied to the concentrations. It is directly the measured concentrations which are taken for the estimation of the emission factors. As your comment indicates, this is not clearly described in the text. The manuscript was modified so that this aspect appears clearly.

**(3) Changes in the manuscript**

| The clarification has been added in the text. | P19 | L513-516 |
* * *
**(1) Comment from referee**

The range of values for the loss rated due to dilution (alphas) should be presented so we can compare them in magnitude with the loss rates due to wall deposition (betas), and show that the dilution rates in all the runs are in the expected range. One way to do this would be to separate "whisker" plots for alpha as part of, or on conjunction with, Figure 4. Are the dilution rates similar in the NH4SO4 experiments, or are they a factor in the lower alpha+beta values shown for those experiments in Figure 4? Is particle loss to the walls important compared to dilution in the NH4SO4 experiments?

**(2) Author's response**

The dilution rates are almost all similar (vary with a factor 3, between $11.25 \times 10^{-4}$ and $3.75 \times 10^{-4}$ min$^{-1}$), and represent only 7 to 24 % of the total loss rate (alpha+beta) for exhaust experiments. For the ammonium sulphate experiments, they represent on average 45 % of the total loss rate. I agree that it is important to explain this in the manuscript, to emphasize the importance of the wall deposition process depending on the conditions. Figure 4 was changed in the updated manuscript to address your comment, and highlight the predominance of the wall losses over leakage by separating the box plots to show vales of alpha, beta, and alpha+beta separately.

Regarding your remarks on ammonium sulphate experiments, they do have similar dilution rates as exhaust experiments. However, since wall loss rates are smaller for those experiments, the share of dilution is higher than for exhaust experiments. It represents on average 45 % of the total losses. This was added in the updated manuscript.

**(3) Changes in the manuscript**

| The dilution rates have been added in the text, in addition to total loss coefficients. | P14-15 | L407-410 |
| | P15 | L423-431 |
| Figure 4 has been modified to clearly show the dilution rates, wall loss coefficients, and total loss coefficients. | P16 | L445-454 |
| Description of Figure 4 has been adapted to the modified figure. | P16-17 | L455-463 |
| The total loss coefficients have been more clearly mentioned. | P17 | L471 |
| The share of wall deposition in total losses has been added in the conclusion. | P33 | L839-840 |
* * *
**(1) Comment from referee**

Figure 5 shows that, except for two gasoline exhaust runs that are very different from all the others, the kPM values from the NH4SO4 experiments are quite a bit lower than the kBC values from the exhaust experiments, and also the slope of the k vs alpha+beta line is lower. Since BC is also chemically different from exhaust particles, couldn't it also have different wall loss rates or different effects of rates on size? Were any of the NH4SO4 experiments carried out with electrostatic charged walls?

**(2) Author's response**

It is true that for ammonium sulphate particle the slope of $k_{PM}$ vs alpha+beta is a little bit lower than the slope of the fit showed on Figure 5. However, when taking the ratios of $k_{PM}$/(alpha+beta) for ammonium sulphate, the average over the 5 experiments is 1.04. This indicates that the average difference is about 4 % between both coefficients for ammonium sulphate particle experiments. This is consistent with the general trend observed on Figure 5. However, as the graphic representation doesn't really represent this result, the main text was modified to clearly make it appear.

Regarding BC, it enters in the composition of exhaust particles as one of the main species (Kostenidou et al., 2021). Therefore, when particles deposit onto the walls, BC does too. The wall loss rates can be different if BC is not homogeneously distributed in all particle sizes. However, the particle internal mixing assumption has been made in many studies, considering BC to be a good tracer for exhaust particles (Grieshop et al., 2009; Hennigan et al., 2011; Platt et al., 2013).

Finally, the ammonium sulphate experiments were all carried out when the walls were assumed to be neutralized. The goal of these experiment was to be carried out without organic material, and in conditions as similar as possible to our standard experimental conditions (e.g. without high electrostatic charge on the wall). The parameter that was changed between each ammonium sulphate experiment was the initial concentration, as it can cover a wide range of values during exhaust particle experiments.

**(3) Changes in the manuscript**

| The discussion on the correlation for ammonium sulphate particles has been added in the text. | P18 | L498-500 |
|---|---|---|

**(1) Comment from referee**

The increase in particle mass with time during most of the experiments are explained by low-volatility gases condensing onto existing particles. Equilibrium partitioning theory predicts that the equilibrium fraction in the particle phase increases with the total particle mass, and is not dependent on particle number. Likewise, the condensation rate would depend on particle surface area, which I think should correlate somewhat better with mass than number. Nevertheless, Figure 8b shows a plot of data related to particle mass increases against particle number, not particle mass or surface area. Is the correlation not as good if plots are against particle mass or surface area instead? If this is the case, it should be pointed out and attempts to explain this should be offered (though I can't think of any explanation if this indeed were the case.) If number, mass and area are highly correlated then the plots would look the same, but in that case plots against particle mass would be more appropriate since it corresponds more directly to the explanation you are giving and existing theories.

**(2) Author's response**

Thank you for this comment on the fact that plotting PM increase versus initial mass PM or initial surface PS would be more adequate than versus initial PN. We decided to plot the PM

increase against initial particle surface PS instead, since it makes more sense with regards to the theory (as explained in your comment). The general trend is similar to that obtained when plotted against initial PN. The best fit obtained here is logarithmic. It reflects the fact that PM increase is limited, and will at some point stop increasing even though initial PS increases. We interpreted it as the fact that above a certain threshold ($\sim 10^4$ $\mu m^2.cm^{-3}$), PM increase is limited by certain factors. A limiting factor could be the concentration of available organic material. This result is interesting, and is in good agreement with other studies. Namely, Charan et al. (2020) found that above $\sim 1800$ $\mu m^2.cm^{-3}$, initial seed surface area becomes insignificant in terms of SOA yield, partly due to initial precursor concentrations. The different value of the threshold between this study and ours could be explained by the differences in chamber size, experimental conditions, particle nature, and composition of the gas phase. Figure 8b was changed to have initial particle surface PS on the x-axis, and this discussion was added in the main text and in the conclusion.

| **(3) Changes in the manuscript** | | |
|---|---|---|
| Figure 8 has been modified as suggested to plot PM increase versus initial particle surface instead of initial particle number. | P25 | L654-660 |
| Interpretation of Figure 8 has been adapted to the modified figure. | P25-26 | L662-676 |
| Since Figure 8 shows PS in $\mu m^2.cm^{-3}$, Figure 9 has been adapted to use the same unit. | P27-28 | L680-693 |
| The correlation between PM increase and particle surface instead of particle number has been modified in the conclusion. | P34 | L851-852

L858-859 |

In conclusion, I think the paper needs to give more data and information about the validity and performance of the correction method, and the effects of these uncertainties on the corrections to the data that they present, before it is accepted for publication.

**Anonymous Referee #2** (https://doi.org/10.5194/amt-2021-43-RC2, 2021)

Dear reviewer,

Thank you for your detailed comments. Your comments are in the grey lines (1). Responses are given below each of them, in the white lines (2). They are followed by the changes which have been made in the manuscript (3).

The authors investigate the emissions of 6 diesel passenger cars from Euro 3 to Euro 6 under 2 different driving cycles. They determine emission factors of particle number, particle mass, black carbon, NMHC and IVOC. They present a method to correct the evolution of particle mass concentration for dilution and wall loss of particles in the chamber. They claim that under dark conditions particle mass concentration (PM) increases over time as a function of initial particle number or particle surface concentration. Below a particle number concentration of $(8\text{-}9)\text{E4 cm}^{-3}$ the particle number concentration (PN) increases, while above it decreases.

The experiments and the data analysis are well described. However, there is much speculation regarding the interpretation of the data and a lack of proof of their claims.

**(1) Comment from referee**

The authors interpret their observation of a sustained increase of particle mass and number concentrations by condensation and nucleation, respectively. However, to have continuous condensation or nucleation a constant production of condensable or nucleating vapors need to occur. Otherwise, vapors will rapidly condense on particles and on the wall and condensation or nucleation stops. In their small chamber the lifetime of such vapors will be less than ten minutes. Therefore, the claim that PM increases over many hours could be due to condensation of IVOCs is not plausible. The authors may also check if the saturation vapor pressure of IVOCs is low enough to partition to the particle phase at the particle mass concentrations of their experiments.

**(2) Author's response**

The partition of IVOCs and SVOCs has been studied to check the potential role of IVOCs in particle evolution. Briefly, the effective saturation concentrations of n-alkanes up to C32 is obtained with the method described in Lu et al. (2018), for VOCs (C>12), IVOCs (C12-C22) and SVOCs (C23-C32). Using Equation (14.43) from Seinfeld and Pandis (2016), the fraction in particle phase is computed, for 2 conditions of initial [PM] typical of significant [PM] increase observed on Figure 9 ($200\,\mu\text{g.m}^{-3}$ and $2500\,\mu\text{g.m}^{-3}$). The organic aerosol concentration [OA] is estimated to range roughly between 2 and 80 % of [PM], based on studies giving the ratio [PM]/[OA] or the ratio EC/OC (Kostenidou et al., 2021; May et al., 2014). The wide range is caused by several parameters, such as measurement technique, experimental conditions, driving conditions, vehicle type (motorization and aftertreatment technologies). In this study, the [OA] fraction of [PM] is estimated to be in the range 2-80 %.

This is based on the difference between [PM] and [BC], which indicates that the share of organic aerosol could go up to 69 % for the Euro 3 and Euro 4 vehicles.

For the condition with initial [PM] = 200 µg.m$^{-3}$, the percentage of n-alkanes IVOCs present in the particle phase ranges from 0.8 % to 13.7 %, for [OA] estimated as 2 % and 80 % of [PM] respectively. The percentage of n-alkanes SVOCs in the particle phase ranges from 85.5 % to 98.7 % respectively. For the estimation with [OA] = 69 % of [PM], 12.7 % of n-alkanes IVOCs and 98.5 % of n-alkanes SVOCs are in the particle phase.

For the condition with initial [PM] = 2500 µg.m$^{-3}$, the percentage of n-alkanes IVOCs present in the particle phase ranges from 6.7 % to 34.0 %, for [OA] estimated as 2 % and 80 % of [PM] respectively. The percentage of n-alkanes SVOCs in the particle phase ranges from 96.5 % to 99.9 % respectively. For the estimation with [OA] = 69 % of [PM], 32.7 % of n-alkanes IVOCs and 99.9 % of n-alkanes SVOCs are in the particle phase.

These results indicate that at such high particle concentrations, significant fractions of IVOCs and SVOCs could be present in the particle phase. It indicates that IVOCs can participate in the [PM] evolutions observed on Figure 8 and Figure 9, due to high PM concentrations and presence of IVOCs (Figure 6d) for the Euro 3 and Euro 4 vehicles.

This part was added and discussed in Appendix G.

I agree that the growth over several hours cannot be explained solely by the presence of organic material, which would deposit onto the walls or partition in the particle phase. Other interpretations must be discussed. One of them is the possibility that the walls would play the role of a source of organic material. This hypothesis relies on several studies, which showed that vapor wall deposition was a reversible process (Matsunaga and Ziemann ‡, 2010; Yeh and Ziemann, 2015; Zhang et al., 2015). Walls could therefore play a role of sink as well as a role of source of gas-phase pollutants during chamber experiments (Kaltsonoudis et al., 2019). Zhang et al. (2015) observed evaporation of organics from the walls when the temperature went from 25 to 45 °C. Even though the temperature increase in our experiments is not as high (5 °C on average, due to heat created by instrumentation), the role of walls as a source of organic material might be part of the explanation of PM increase. This is only a hypothesis, and it is not likely to explain alone the PM increase. It might still play a role in the observed evolutions by inducing continuous condensation. Some discussion was added in the updated manuscript.

| (3) Changes in the manuscript | | |
|---|---|---|
| Some discussion regarding IVOC/SVOC partitioning and their role on PM evolution has been added in the text. | P24 | L623-625 |
| The discussion regarding the potential role of the walls as a source of organics has been added in the text. | P24 | L629-645 |
| The discussion on IVOC/SVOC partitioning has been added in the conclusion. | P34 | L854-856 |
| Appendix G has been added to discuss the partitioning of IVOCs/SVOCs. | P42-43 | L1173-1206 |

**(1) Comment from referee**

There are also some issues regarding the PN increase and the particle number concentration threshold. This observation depends on the available instrumentation. The smallest particles they can measure is 14 nm. They do not present the particle size distributions of their different experiments and therefore it is not clear in which cases a nucleation mode is formed. This

would happen very fast during injection and there is not a steady nucleation going on as explained above. If an increase of PN occurs, this happens because of coagulation of nucleation mode particles, which produces particles of larger size that become measurable in their SMPS. This is not an increase of PN but the fact, that they did not measure the particles below 14 nm. Thus, the threshold and the time of increase depend on this measurement parameter. The fact, that a growing nucleation mode influences the PN at larger sizes is not a new finding. One should always be aware of it when analyzing such data.

**(2) Author's response**

I agree that continuous nucleation is not a possible explanation for continuous PN increase. Coagulation of nucleation mode particles, initially undetected by the SMPS, is a reasonable explanation. Nucleation mode particles are most likely to be present and have significant impacts when initial concentrations are low (below $[8-9]\times10^4$ #.cm$^{-3}$). Above that concentration, they might still be present but without observable impacts on PN evolution. Some discussion was added with Figure 10 to discuss the cases for which nucleation mode particles are likely to be present.

Due to this, the interpretation of Figure 11 was thoroughly modified, and the figure was adapted. The PN increase is now clearly presented as an artifact due to the SMPS measurement range. It is not a physical creation of particles, but growth of nucleation mode particles. This enables to discuss the presence of a nucleation mode even though it is not measurable with the current experimental set up. Above the value of $[8-9]\times10^4$ #.cm$^{-3}$, coagulation of nucleation mode particles might still occur, but with negligible impacts on PN evolution. Coagulation of the particles in the SMPS range is in those conditions the predominant process explaining the decrease of PN. Therefore, the chamber is well adapted to observation and quantification of the process of coagulation for initial concentrations above this value. As your comment says, the threshold and the time of increase don't represent indicators of physical processes. The time of increase was removed since it has no physical interpretation and has no use for the characterization of the chamber. The threshold value is now described as the value below which nucleation mode particles are likely to have significant effects on particle number evolution. It is also given as the value above which the chamber and experimental set up are suitable for quantification and observation of the coagulation process.

We decided to show these results in the text with the modified Figure 11, because one of the main goals of this paper is also to characterize the chamber. Therefore, detailing the most suitable conditions for the study of a physical process (coagulation) appears to be relevant.

| (3) Changes in the manuscript | | |
|---|---|---|
| The explanation by nucleation has been replaced by growth of nucleation mode particles. | P1 | L28-33 |
| Some discussion regarding the growth of nucleation mode particles has been added in the text. | P29 | L716-748 |
| Figure 11 has been modified to address the reviewer's comments, and the time of increase was removed. | P31 | L750-760 |
| The discussion on Figure 11 and PN evolution has been modified to address the reviewer's comments. | P31-33 | L761-795 |

| | | L801-829 |
| --- | --- | --- |
| The interpretations of PN evolution have been modified in the conclusion. | P34 | L859-871 |

**(1) Comment from referee**

At high PN concentrations when "coagulation prevails" the Figures 11c,d,e show first the expected decrease of PN due to coagulation, which is however after 1-2-h followed by an increase of PN. Do the authors have an explanation for this observation? Are these wall loss corrected PN data? If yes, this would mean the newly developed algorithm could eventually not correctly compensate for the losses.

**(2) Author's response**

The slight increase in the second half of the evolutions could in part be explained by the growth of nucleation mode particles. As discussed on Figure 10c and d, nucleation mode particles could be present in those conditions (high initial PN). It would be relatively small compared to total PN, and would grow slowly by coagulation and/or condensation. This slow growth could explain the slight PN increase observed after a few hours. Also, as time increases, the correction becomes more important, with higher error bars. Taking the error bars into account makes the increase less significant.

**(3) Changes in the manuscript**

| The discussion has been added in the text. | P32 | L795-800 |
| --- | --- | --- |

**(1) Comment from referee**

The authors need to provide more evidence why PM should grow over hours. Condensable vapors are lost to the walls and it is not plausible how an oversaturation is maintained over hours without production. Thus, the wall-loss correction method could induce such an artifact.

**(2) Author's response**

Regarding the possible artifact induced by the correction method, we made some verifications to make sure that the observed PM growth was a real phenomenon. We specifically looked at the cycles with the highest increase rates (3 last points on Figure 8, from Euro 3 MW conditions) and the longest increase durations (points on the upper left corner of Figure 9a, from Euro 3 UC and Euro 4 UC and MW conditions). We performed the verifications using the measured (e.g. uncorrected) PM concentrations. It is presented on Appendix F of the updated manuscript. Briefly, for the "rapid" PM increase experiments, measured data also show high PM increase rates. The increase durations are smaller than for corrected PM. This difference is normal considering the fact that leakage and wall deposition would at some point overtake PM growth even though the latter is still occurring. This corresponds to a slight decrease of uncorrected PM, and an increase of corrected PM. It hereby extends the PM increase duration obtained for corrected concentrations. Moreover, for the "long" PM increase experiments, measured PM concentrations show increase

durations going up to 355 minutes (Euro 4 MW conditions). The shorter increases are followed by slight PM decreases, with decrease rates smaller than that of BC (by a factor 2.2 on average), until the end of the experiment. This indicates that processes leading to PM increase are present at all times of the experiments. Moreover, for the longer increases, the increase phases are followed by extremely small decreases or constant concentrations. This indicates that PM increasing processes are present during the whole time of the experiment. Overall, the PM increase rates and durations observed for corrected concentrations appear to be due to actual physical processes, and not artifacts induced by the correction method. The figures and discussion have been added in Appendix F of the updated manuscript and in the main text, before Figure 8, to clarify this aspect in section 3.3

More interpretation is needed to explain the observed PM increase. The walls being a source of organic material is not likely to be the main reason. Another possible explanation is the growth of small particles, undetected by the SMPS (as seen for the PN evolutions). This is discussed with the measured PN distribution of 2 cycles with long increases (Euro 3 UC and Euro 4 MW). They show that during the whole experiments, small particles (below 20 nm) are present in significant and almost constant concentrations. They could come from the growth of nucleation mode particles (by coagulation or condensation), undetected by the SMPS before their growth. The reason why the concentration of those particles remains quite steady could be that they increase (by growth of nucleation mode particles) and decrease at the same time (by their own coagulation). This would have the effect of bringing more material inside the detection range of the SMPS, therefore increasing total particle mass during the whole course of the experiments. This explanation would mean that the observed PM increase is partially an artifact (due to SMPS measurement range), and that it might induce an overestimation. Some discussion about this was added in the main text.

Overall, the continuous PM increase observed for the Euro 3 and Euro 4 vehicles can be explained by several complementary phenomena (growth of initially undetected particles, condensation of emitted organics, condensation of organics released by the walls). Three figures and some discussions were added in the updated manuscript (Appendix F, Appendix H, main text) in order to more accurately describe the observed PM evolutions.

| (3) Changes in the manuscript | | |
|---|---|---|
| The word "condensation" has been replaced by "PM increase" in several cases. | P1 | L24 |
| | P28 | L693-694-699 |
| The discussion on the potential artifact induced by the correction method on PM increase has been added in the text. | P23-24 | L619-621 |
| The discussion on PM increase due to growth of nucleation mode particles has been added in the text. | P24 | L645-651 |
| The interpretations on PM increase have been modified in the conclusion. | P34 | L849-851 |
| Appendix F has been added to show measured PM evolution, and show that the increase is not an artifact induced by the correction method. | P41-42 | L1145-1172 |
| Appendix H has been added to discuss the role of growth of nucleation mode particles on PM evolution. | P44-45 | L1207-1223 |

**(1) Comment from referee**

Although Figure 5 shows a good correlation between the mean loss coefficients and $k_{BC}$ it is also obvious, but not mentioned, that there is a large off-set. Why does this occur and how does this affect the PM correction? The authors need to show that the PM growth is a real phenomenon and not an artifact from their correction method.

**(2) Author's response**

The off-set of Figure 5 represents the fact that for some cycles the average loss coefficient obtained with the optimized value of $k_e$, is slightly higher than $k_{BC}$. It reflects the fact that small particles have a higher loss coefficient than the larger ones, and that it impacts the average loss coefficient $\alpha+\beta_{ke}^{mean}$. In spite of that, the optimized value of $k_e$ that was taken is that for which corrected PM from steps 1 and 2 match the better. Another value of $k_e$ could have be chosen, in order to have $\alpha+\beta_{ke}^{mean}$ coefficients and $k_{BC}$ coefficients as close as possible. Some tests were conducted on 3 experiments to establish the impact of matching the loss coefficients $\alpha+\beta_{ke}^{mean}$ and $k_{BC}$ instead of matching the corrected PM from steps 1 and 2. The tests were conducted on the 2 experiments for which the coefficients $\alpha+\beta_{ke}^{mean}$ and $k_{BC}$ have the worse match (the 2 gasoline on the lower left corner of Figure 5). It results in a PM difference of less than 10 % on average (8.9 and 9.7 % for each test). A test was also conducted on a cycle with higher loss coefficients (the gasoline experiment with the highest value of $\alpha+\beta_{ke}^{mean}$). For this cycle, the impact on corrected PM is 9.8 %. These impacts are not negligible, but don't affect the trends observed in section 3.2. Also, due to the size-dependence of the wall losses, the match of corrected PM from steps 1 and 2 seems to be a better indicator than the match of average loss coefficients $\alpha+\beta_{ke}^{mean}$ and $k_{BC}$. Some discussion about this was added in the main text, below Figure 5.

**(3) Changes in the manuscript**

| Some discussion has been added in the text. | P18 | L501-503 |
|---|---|---|

**(1) Comment from referee**

By calculating corrected PM do they use the average loss rate or the size dependent loss rates?

**(2) Author's response**

Corrected PM is obtained using the size-dependent loss rate. The "average loss rate" $\alpha+\beta_{ke}^{mean}$ is never actually applied. It is computed using the size-dependent loss coefficients of each diameter range, which are applied to each particle size bin.

**(3) Changes in the manuscript**

| A clarification has been added in the text. | P11 | L331 |
|---|---|---|

**(1) Comment from referee**

Line 27: "Condensation is 4 times faster when the available particle surface if multiplied by 3". How did the authors calculate this? Is this for a certain particle size?

**(2) Author's response**

This result is obtained using Figure 9a. The group with high initial particle surface has an average initial concentration of $1.57 \times 10^5$ µm$^2$.cm$^{-3}$, with an average time to reach maximum PM of 12 minutes. The other group has an average initial concentration of $2.40 \times 10^4$ µm$^2$.cm$^{-3}$, with an average time to reach maximum PM of 482 minutes. This means that there is a ratio 6.5 between initial surfaces of both groups (instead of 3 as written in the preprint), and a ratio ¼ in the time needed to reach maximum PM. The value was modified, and the explanation of the computation was added in the main text of the updated manuscript.

**(3) Changes in the manuscript**

| | | |
|---|---|---|
| The value has been corrected. | P1 | L28 |
| The explanation of the calculation has been added in the text. | P28 | L700-702 |
| The value has been corrected in the conclusion. | P34 | L857-858 |

Overall, I do not see much scientific progress in this paper.

**Other changes made in the manuscript**

**(3) Changes in the manuscript**

| | | |
|---|---|---|
| Figure 5 has been modified because decimals were separated by commas instead of points. | P18 | L494 |

**References:**

[revised manuscript text omitted]

---

## Author Response (AR2)

Author's response (amt-2021-43)

**Anonymous Referee #1**

Dear reviewer,

Thank you for your comment. Your comment is in the grey lines (1). The response is given below in the white lines (2). It is followed by the changes which have been made in the manuscript (3).

**(1) Comment from referee**

One of my comments concerned the need for more information regarding how well different correction methods agreed. The revised paper said agreement was on the order of 95-97%, and gave an equation in a new Appendix E to calculate the average relative difference. However, according to Equation (E1) 0% would be perfect agreement and 95-97% would be almost a factor of 2 discrepancy. The text either needs to be revised to be consistent with Appendix E or this discussion needs to be clarified.

**(2) Author's response**

The equation of Appendix E was meant to compute the error between results of both steps. From this error, the average agreement was also computed, as 100 % - error, and given in the text. This computation was not explicitly mentioned and therefore unclear. To correct this confusion, the equation in Appendix E was modified to directly give the agreement which is mentioned in the text.

**(3) Changes in the manuscript**

| Modification of Equation E1 and its description in the text. | P38 | L1114-1116 |
|---|---|---|

Dear reviewer,

Thank you for your detailed comments. Your comments are in the grey lines (1). Responses are given below each of them, in the white lines (2). They are followed by the changes which have been made in the manuscript (3).
* * *
**(1) Comment from referee**

The authors performed more data analysis and provide further information on the experiments. The answers to the reviewers are often general and interpretation of data still remain fairly speculative. There are two main issues.
* * *
**(1) Comment from referee**

1) From the new Figure H1 it is obvious that there is a strong growth of the particle size distribution over time. However, their "new method" to determine size dependent wall loss requires a stable size distribution. Otherwise, the change of mass concentration in a size bin does not only depend on wall loss, but also heavily on growth. Thus, the eddy diffusivity coefficient and size dependent wall loss coefficient depend on both, wall loss and growth. The loss rate of BC is not affected by this. The reason that kBC and Beta(Dp) correlate is only due to the fact that Beta(Dp) is fit to kBC. The size dependent wall loss rate method is not independent of kBC and would also not work without the knowledge of kBC. The huge scatter of ke may be due to the inappropriate application. In conclusion, this method development does not provide what it is claimed for.

The authors also claim that the discrepancy in calculated ke between ammonium nitrate and vehicle emission experiments could be attributed to the nature of particles. What is the physics behind this statement? What factor in equation (3) should lead to such a pronounced effect?
* * *
**(2) Author's response**

Thank you for your comment, which underlines a lack of clarity in the manuscript. The method is, as you say, dependent on the knowledge of $k_{BC}$. Black carbon is used as a tracer for total losses (leakage + wall losses), as has been done in several studies (Grieshop et al., 2009; Hennigan et al., 2011; Platt et al., 2013). From that, $k_e$ is optimized until corrections of steps 1 and 2 match. This is equivalent to forcing the correlation between the coefficients $\alpha+\beta^{mean}$ and $k_{BC}$. Therefore, fitting $\alpha+\beta^{mean}$ to $k_{BC}$ guarantees that $\alpha+\beta^{mean}$ accurately represents total losses, such as $k_{BC}$ does. Once $\alpha+\beta^{mean}$ is fitted, the size-dependent coefficients $\alpha+\beta_i(D_p)$ can be applied to the distribution, with the optimized value of $k_e$. It allows to correct each diameter bin, accounting for the fact that particles have different loss coefficients depending on their size (Crump and Seinfeld, 1981).

Moreover, the distribution cannot be stable, because the concentration in a size bin is affected by both losses and growth (both size-dependent). However, since $\alpha+\beta^{mean}$ (average of $\alpha+\beta_i(D_p)$ of each diameter) is fitted to $k_{BC}$, which is not affected by growth, the distribution is only corrected for total losses. Therefore, potential effects of growth (coagulation or condensation for instance) are

preserved. This is the goal of this method, which is meant to correct total losses (leakage and wall losses), in order to better show the physical processes. Some discussion regarding those aspects has been included in the manuscript.

Regarding the discrepancy and scatter of $k_e$ values, a hypothesis explaining lower loss rates for ammonium sulphate particles could be the effect of charge. Particle charge and electric field can have a dominant effect on wall deposition (McMurry and Rader, 1985; Nah et al., 2016). This factor is not included in the theory of Crump and Seinfeld (1981), and doesn't appear in Equation (3). Perturbations near a chamber can easily induce a buildup of charges on the walls, thus impacting loss rates (Wang et al., 2018). Perturbations can be due to friction of the walls, motion near the chamber, touching the walls, air flows. The ammonium sulphate experiments took place outside of exhaust campaigns, with less motion/people/instruments near the chamber. Also, the injection and sampling flows were lower for ammonium sulphate experiments, inducing lower turbulence. These parameters can reasonably explain the lower values of $k_e$ for ammonium sulphate experiments. Even though ammonium sulphate experiments took place in different conditions as exhaust experiments, their characterization was important to compare results obtained with the method as those found in literature. Also, such results are useful as many experiments of photochemistry are conducted with preexisting ammonium sulphate seeds. Finally, the fact that perturbations are easily induced on a chamber can also explain the scatter of $k_e$ values found for exhaust experiments. Some discussion has been added in the manuscript to emphasize the impact of the experimental conditions instead of the nature of particles.

| (3) Changes in the manuscript | | |
|---|---|---|
| Discussion regarding the use of BC as a tracer. | P9 | L267-268 |
| Clarification on the difference between $\alpha+\beta_i(D_p)$ and $\alpha+\beta^{mean}$ and their application. | P11 | L330-336 |
| Discussion on the correction of particles in a size bin. | P12 | L339-341 |
| Mention of the specific wall losses for ammonium sulphate particles. | P16 | L442-443 L449-450 |
| Details on the specific wall losses for ammonium sulphate particles. | P17-18 | L473-481 |
| Details on the specific wall losses for ammonium sulphate particles. | P18 | L495-499 |
| Discussion on the comparison between $\alpha+\beta^{mean}$ and $k_{BC}$. | P18 | L501-503 |
| Clarification on the use of $\alpha+\beta_i(D_p)$. | P19 | L517-519 |
| Discussion on the specific wall losses for ammonium sulphate particles. | P19 | L520-524 |

**(1) Comment from referee**

2) The authors attribute the measured growth of PM to a prolonged condensation of IVOCs, as outlined in Appendix G. Indeed, IVOCs can partition into PM at given conditions. However, according to Figure 6, IVOC emissions are 10-20 times lower than PM emissions (Euro 3 and Euro 4). The increase in PM mass is 30-120% (Fig F1). Thus, there is no way enough IVOCs to contribute to this PM increase.

The paper does not provide a sound explanation of what the reason for the measured PM increase is. I believe this is an important issue as such a large effect, if real, would have far-reaching implications. For example, there are many studies on SOA formation of vehicle emissions. If primary emissions would show such a behavior, all/most of these studies could be heavily biased. Even emission factors, as determined in this paper, would be incorrect. One should take PM not after injection but at a much later time. This is not even addressed in the paper. I am not aware of

chamber studies reporting such an effect. Thus, there is still the option of an instrumental or procedural artefact. One possibility could be an evolving change of particle density with coagulation of primary particles due to a high BC fraction.

**(2) Author's response**

We agree that IVOCs alone cannot be responsible for the PM increase observed in our study. Their emission and contribution are discussed because they were successfully quantified at emission. This was one of the goals of this study, as many uncertainties exist on IVOC emissions. Also, our results suggest that they could play a significant role in PM increase in some cases, due to their fraction in particle phase (as outlined in Appendix G). This is therefore included in the discussion of PM evolution. However, SVOCs are likely to have a major role on PM evolution in the dark, as they can easily partition between the gas and particle phase (as seen in Appendix G). They are known to be a potential large source of SOA (Kroll and Seinfeld, 2008; Robinson et al., 2007; Zhao et al., 2014). They are likely to participate in the PM evolutions observed in our study, with similar processes as those discussed for IVOCs. This interpretation relies on the assumption that SVOCs were emitted. This assumption seems reasonable, especially for the Euro 3 and Euro 4 vehicles, based on data on SVOCs from diesel vehicles (Lu et al., 2018; Zhao et al., 2015). Also, some SVOCs were measured in our study with sorbent tubes, with emission factors in the same range as IVOCs. However, it is likely that only a small fraction of SVOCs was actually measured with the sorbent tubes (Lu et al., 2018). Their emissions could therefore be higher and explain in part the increase of PM. We agree that the emphasis on IVOC role was too important, with limited discussion on SVOC role. This was modified in the updated manuscript.

Regarding the potential implications on primary emission evolutions, we think that our study could help understand the evolutions in conditions with no processes of photooxidation. As you say, there are many studies on SOA formation with artificial light exposure, and our study could be a complement to describe evolutions without light exposure. Also, we think that emission factors shouldn't be measured after several hours, but right after injection, because the evolutions are due to the partitioning but also to potential interactions in the gas-phase. However, given the important question of dilution conditions when measuring particle emissions (Robinson et al., 2007), it is important to clearly indicate the dilution ratio associated to emission factors of particles. As you suggested, some discussion has been added in the updated manuscript.

Finally, according to Peng et al. (2016), it appears that particle evolution induces an increase of density up to 1.4 $g.cm^{-3}$, due to deposition of organic material. Applying this density at the end of the evolution in our study would induce a more significant PM increase. The PM increase with a final density of 1.4 $g.cm^{-3}$ would be about 17 % higher than one with the density in our study (1.2 $g.cm^{-3}$). As density was not measured, we chose to apply a constant value to avoid overestimating PM increase. The value of 1.2 $g.cm^{-3}$ was chosen to be in the range of what is found in several studies (Bahreini et al., 2005; Barone et al., 2011; Hallquist et al., 2009; Totton et al., 2010). Some discussion has been added in the updated manuscript.

**(3) Changes in the manuscript**

| | | |
|---|---|---|
| Discussion on the methodology to compute particle emission factors from the chamber. | P20 | L537-543 |
| Discussion on SVOC measurements. | P22 | L608-610 |
| Discussion on SVOC role in PM evolution. | P23 | L648-651 |
| New paragraph for the interpretation of PM evolution. | P23 | L654 |

| | | |
|---|---|---|
| Discussion on the potential role of a change of particle density on PM evolution. | P24 | L677-681 |
| Mention of SVOCs for PM evolution. | P30 | L844 |
| Mention of SVOCs for PM evolution. | P41 | L1177 |

**Other changes made in the manuscript**

| (3) Changes in the manuscript | | |
|---|---|---|
| Misspelling of a word. | P17 | L457 |
| Reference list updated according to Copernicus Publications standards. | P31-36 | L876-1080 |

**References**

Bahreini, R., Keywood, M. D., Ng, N. L., Varutbangkul, V., Gao, S., Flagan, R. C., Seinfeld, J. H., Worsnop, D. R., and Jimenez, J. L.: Measurements of Secondary Organic Aerosol from Oxidation of Cycloalkenes, Terpenes, and *m*-Xylene Using an Aerodyne Aerosol Mass Spectrometer, Environ. Sci. Technol., 39, 5674–5688, https://doi.org/10.1021/es048061a, 2005.

Barone, T. L., Lall, A. A., Storey, J. M. E., Mulholland, G. W., Prikhodko, V. Y., Frankland, J. H., Parks, J. E., and Zachariah, M. R.: Size-Resolved Density Measurements of Particle Emissions from an Advanced Combustion Diesel Engine: Effect of Aggregate Morphology, Energy Fuels, 25, 1978–1988, https://doi.org/10.1021/ef200084k, 2011.

Crump, J. G. and Seinfeld, J. H.: Turbulent deposition and gravitational sedimentation of an aerosol in a vessel of arbitrary shape, J. Aerosol Sci., 12, 405–415, https://doi.org/10.1016/0021-8502(81)90036-7, 1981.

Grieshop, A. P., Logue, J. M., Donahue, N. M., and Robinson, A. L.: Laboratory investigation of photochemical oxidation of organic aerosol from wood fires 1: measurement and simulation of organic aerosol evolution, Atmospheric Chem. Phys., 9, 1263–1277, https://doi.org/10.5194/acp-9-1263-2009, 2009.

Hallquist, M., Wenger, J. C., Baltensperger, U., Rudich, Y., Simpson, D., Claeys, M., Dommen, J., Donahue, N. M., George, C., Goldstein, A. H., Hamilton, J. F., Herrmann, H., Hoffmann, T., Iinuma, Y., Jang, M., Jenkin, M. E., Jimenez, J. L., Kiendler-Scharr, A., Maenhaut, W., McFiggans, G., Mentel, T. F., Monod, A., Prevot, A. S. H., Seinfeld, J. H., Surratt, J. D., Szmigielski, R., and Wildt, J.: The formation, properties and impact of secondary organic aerosol: current and emerging issues, Atmos Chem Phys, 82, 2009.

Hennigan, C. J., Miracolo, M. A., Engelhart, G. J., May, A. A., Presto, A. A., Lee, T., Sullivan, A. P., McMeeking, G. R., Coe, H., Wold, C. E., Hao, W.-M., Gilman, J. B., Kuster, W. C., de Gouw, J., Schichtel, B. A., Collett, J. L., Kreidenweis, S. M., and Robinson, A. L.: Chemical and physical transformations of organic aerosol from the photo-oxidation of open biomass burning emissions in an environmental chamber, Atmospheric Chem. Phys., 11, 7669–7686, https://doi.org/10.5194/acp-11-7669-2011, 2011.

Kroll, J. H. and Seinfeld, J. H.: Chemistry of secondary organic aerosol: Formation and evolution of low-volatility organics in the atmosphere, Atmos. Environ., 42, 3593–3624, https://doi.org/10.1016/j.atmosenv.2008.01.003, 2008.

Lu, Q., Zhao, Y., and Robinson, A. L.: Comprehensive organic emission profiles for gasoline, diesel, and gas-turbine engines including intermediate and semi-volatile organic compound emissions, Atmospheric Chem. Phys., 18, 17637–17654, https://doi.org/10.5194/acp-18-17637-2018, 2018.

McMurry, P. H. and Rader, D. J.: Aerosol Wall Losses in Electrically Charged Chambers, Aerosol Sci. Technol., 4, 249–268, https://doi.org/10.1080/02786828508959054, 1985.

Nah, T., McVay, R. C., Pierce, J. R., Seinfeld, J. H., and Ng, N. L.: Constraining uncertainties in particle wall-deposition correction during SOA formation in chamber experiments, Atmospheric Chem. Phys. Discuss., 1–35, https://doi.org/10.5194/acp-2016-820, 2016.

Peng, J., Hu, M., Guo, S., Du, Z., Zheng, J., Shang, D., Levy Zamora, M., Zeng, L., Shao, M., Wu, Y.-S., Zheng, J., Wang, Y., Glen, C. R., Collins, D. R., Molina, M. J., and Zhang, R.: Markedly enhanced absorption and direct radiative forcing of black carbon under polluted urban environments, Proc. Natl. Acad. Sci., 113, 4266–4271, https://doi.org/10.1073/pnas.1602310113, 2016.

Platt, S. M., El Haddad, I., Zardini, A. A., Clairotte, M., Astorga, C., Wolf, R., Slowik, J. G., Temime-Roussel, B., Marchand, N., Ježek, I., Drinovec, L., Močnik, G., Möhler, O., Richter, R., Barmet, P., Bianchi, F., Baltensperger, U., and Prévôt, A. S. H.: Secondary organic aerosol formation from gasoline vehicle emissions in a new mobile environmental reaction chamber, Atmospheric Chem. Phys., 13, 9141–9158, https://doi.org/10.5194/acp-13-9141-2013, 2013.

Robinson, A. L., Donahue, N. M., Shrivastava, M. K., Weitkamp, E. A., Sage, A. M., Grieshop, A. P., Lane, T. E., Pierce, J. R., and Pandis, S. N.: Rethinking Organic Aerosols: Semivolatile Emissions and Photochemical Aging, Science, 315, 1259–1262, https://doi.org/10.1126/science.1133061, 2007.

Totton, T. S., Chakrabarti, D., Misquitta, A. J., Sander, M., Wales, D. J., and Kraft, M.: Modelling the internal structure of nascent soot particles, Combust. Flame, 157, 909–914, https://doi.org/10.1016/j.combustflame.2009.11.013, 2010.

Wang, N., Jorga, S., Pierce, J., Donahue, N., and Pandis, S.: Particle Wall-loss Correction Methods in Smog Chamber Experiments, Atmospheric Meas. Tech. Discuss., 1–29, https://doi.org/10.5194/amt-2018-175, 2018.

Zhao, Y., Hennigan, C. J., May, A. A., Tkacik, D. S., de Gouw, J. A., Gilman, J. B., Kuster, W. C., Borbon, A., and Robinson, A. L.: Intermediate-Volatility Organic Compounds: A Large Source of Secondary Organic Aerosol, Environ. Sci. Technol., 48, 13743–13750, https://doi.org/10.1021/es5035188, 2014.

Zhao, Y., Nguyen, N. T., Presto, A. A., Hennigan, C. J., May, A. A., and Robinson, A. L.: Intermediate Volatility Organic Compound Emissions from On-Road Diesel Vehicles: Chemical Composition, Emission Factors, and Estimated Secondary Organic Aerosol Production, Environ. Sci. Technol., 49, 11516–11526, https://doi.org/10.1021/acs.est.5b02841, 2015.